# On Robustness to Missing Video for Audiovisual Speech Recognition

**Oscar Chang**                                               *oscarchang@google.com*
*Google LLC, USA*

**Otavio Braga**                                                *obraga@google.com*
*Google LLC, USA*

**Hank Liao**                                                  *hankliao@google.com*
*Google LLC, USA*

**Dmitriy Serdyuk**                                            *dserdyuk@google.com*
*Google LLC, USA*

**Olivier Siohan**                                               *siohan@google.com*
*Google LLC, USA*

**Reviewed on OpenReview:** *https://openreview.net/forum?id=fXorxxbDvO*

## Abstract

It has been shown that learning audiovisual features can lead to improved speech recognition performance over audio-only features, especially for noisy speech. However, in many common applications, the visual features are partially or entirely missing, e.g. the speaker might move off screen. Multi-modal models need to be robust: missing video frames should not degrade the performance of an audiovisual model to be worse than that of a single-modality audio-only model. While there have been many attempts at building robust models, there is little consensus on how robustness should be evaluated. To address this, we introduce a framework that allows claims about robustness to be evaluated in a precise and testable way. We also conduct a systematic empirical study of the robustness of common audiovisual speech recognition architectures on a range of acoustic noise conditions and test suites. Finally, we show that an architecture-agnostic solution based on cascades can consistently achieve robustness to missing video, even in settings where existing techniques for robustness like dropout fall short.

## 1 Introduction

Learning from multiple modalities using large-scale datasets has increasingly been shown to produce stronger representations over those learned from a single modality. Such approaches have led to state-of-the-art performance on numerous tasks in computer vision, natural language processing, and speech recognition (Radford et al., 2021; Ramesh et al., 2021; Yuan et al., 2021; Shi et al., 2022). As multi-modal learning becomes more popular, it is paramount that it should be developed and deployed in a trustworthy manner. This means that multi-modal systems have to be architected in ways that not just leverage features from additional modalities when they are present, but are also robust to missing features from these modalities when they are absent.

In this paper, we study the problem of building audiovisual automatic speech recognition (ASR) models that are robust to missing video. This problem is decidedly asymmetric: only robustness to missing video, and not audio, is desired. This is because state-of-the-art lip-reading models still are not performant enough for many practical ASR applications, which makes only the audio, and not video, indispensable (Serdyuk et al., 2021).

Table 1: Examples of prior work in AV ASR. The second column is a model trained and tested on AV, the third column trained on AV and tested on AO, and the fourth column trained on AO and tested on AO. A robust model should show ascending numbers from left to right. '−' means that this information was not provided by the prior work.

| Prior Work | $Metric(\mathcal{M}_{AV}, Test_{AV})$ | $Metric(\mathcal{M}_{AV}, Test_{AO})$ | $Metric(\mathcal{M}_{AO}, Test_{AO})$ | Robust |
|---|---|---|---|---|
| Chung et al. (2017) | 13.9 WER | 17.7 WER | − | − |
| Zhou et al. (2019) | 9.07 CER | − | 10.33 CER | − |
| Shi et al. (2022) | − | 1.3 WER | 1.5 WER | − |
| Makino et al. (2019) | 20.5 WER | 24.0 WER | 21.5 WER | ✗ |
| Zhang et al. (2019) | 26.2 PER | 71.8 PER | 35.8 PER | ✗ |
| Our Work | 26.12 WER | 31.08 WER | 33.54 WER | ✓ |

Audiovisual (AV) models have consistently achieved ASR performance superior to audio-only (AO) ones (Afouras et al., 2018; Petridis et al., 2018; Ma et al., 2021b), with especially dramatic gains for noisy or overlapping speech (Chung et al., 2017; Abdelaziz et al., 2017; Rose et al., 2021). But it is common for the video of the speaker to be partially or entirely missing in typical ASR applications like providing closed captions for online meetings. For example, the speaker might move off screen, the camera can be turned off, the speaker will occasionally be occluded by other on-screen objects or changes in lighting conditions, etc.

If missing modalities can degrade the performance of a multi-modal model to be worse than that of a single-modality model, then the whole raison d'être for multi-modal learning in the first place becomes questionable. This motivates our goal to build a robust audiovisual model that accords with the following intuition:

**Intuition 1.** *A model is robust to missing video if additional video information at either training or test time can only help, and not hurt, its performance.*

The missing modality problem has received significant attention in the multi-modal learning community. Specifically, in the domain of audiovisual learning, recent years have seen determined efforts to build robust models for tasks including but not limited to: speech recognition (Makino et al., 2019; Zhang et al., 2019; Zhou et al., 2019), expression recognition (Parthasarathy & Sundaram, 2020), event localization (Xuan et al., 2020), voice activity detection (Tao, 2018; Hou et al., 2021), video classification (Nagrani et al., 2021), and speech enhancement (Gogate et al., 2021). Other studies of robustness in multi-modal learning include Morgado et al. (2021) and Han et al. (2022).

Despite the intense interest in the missing modality problem in the audiovisual and ASR literature, there has surprisingly been no consensus on how Intuition 1 ought to be translated into concrete, testable claims. For example, Chung et al. (2017); Parthasarathy & Sundaram (2020); Xuan et al. (2020) show that when trained on AV data, their models do better when tested on AV compared to AO data. Afouras et al. (2018); Zhou et al. (2019); Xu et al. (2020); Ma et al. (2021b) show instead that their proposed methods yield better performance when trained and tested on AV data compared to when trained and tested on AO data. And Ngiam et al. (2011); Shi et al. (2022) show that training on AV instead of AO data yields better test performance on AO. All these different tests capture disjoint aspects of model robustness that are neither equivalent to nor a superset of one another. By testing only one of these aspects, existing work fails to ascertain if their models are truly robust. In fact, on the rare occasion that multiple aspects of robustness are tested, they are found to be robust with respect to one test, but not another (cf. rows 4 and 5 of Table 1). Moreover, it is unclear that even the combination of the above-mentioned tests suffices to ensure that a model is robust enough to be deployed in a real-world production environment.

## 1.1 Our Contributions

In response to this uncertainty, we make the following three salient contributions.

**a) Robustness Framework**    Existing robustness criteria in the literature are inadequate because they try to reduce robustness to a single numerical comparison. In practice however, there are distinct aspects of robustness that individual ad-hoc comparisons fail to capture. To address this, we propose a mathematical

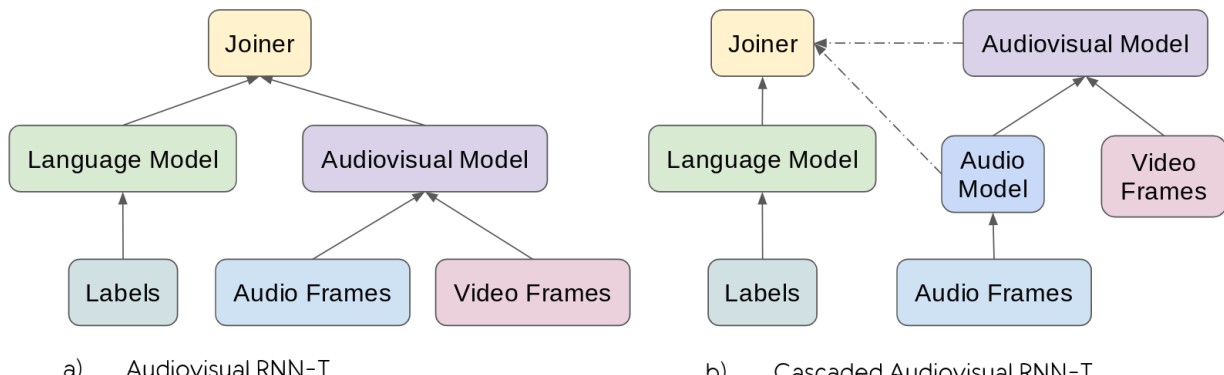

Figure 1: a) An audiovisual model trained with the RNN-T loss. b) A cascaded audiovisual model. The dashed lines indicate different routes the input can take: the AO path, which goes through the audio model, and the AV path, which goes through the audiovisual model.

framework based on order theory that defines what it means for a model to be robust to missing video under a wide variety of settings including missing video at training, testing, or even partially missing video. Our framework is motivated by the key observation that settings with varying amounts of video information can be put in a partial order. A robust model is one that respects this order: as the amount of video information increases, the performance of a robust model should increase as well. Even though there may be an exponential number of test conditions (since each video frame can either be present or absent), we provide useful simplifications that allow for practical simulations of distinct scenarios of missing video. This is critical because for the same quantity of missing video frames, the manner in which the frames are dropped can significantly affect the degradation caused by the missing video. For example, in the case of AV ASR, dropping every second video frame causes only a slight degradation compared to dropping half the video frames in a contiguous segment. By showing how claims about robustness can be made precise and explicitly testable, our framework is a contribution not just to the AV ASR literature, but to the audiovisual and multi-modal learning ones as well.

**b) Empirical Results for Existing Robustness Techniques on Different Architectures**    We conduct a comprehensive empirical study of the robustness of common audiovisual speech recognition architectures in the literature. Existing work recognizes that the solution to achieving robustness to missing video at test time is to expose the model to a similar condition at training time (to close the distribution gap between training and test), and advocates for a dropout-based approach (Chung et al., 2017; Makino et al., 2019; Zhang et al., 2019). Randomly dropping the video or an individual video frame at training time can be understood as an implicit ensemble method that allows a single model to sample from 2 or $2^{\text{number of frames}}$ possible missing video settings, and learn from all of them. However, the previous works of Makino et al. (2019) and Zhang et al. (2019) showed that training with dropout caused a big degradation on an AO test set for LSTMs and feedforward sequential memory networks respectively (cf. rows 4 and 5 of Table 1). We replicate Makino et al. (2019)'s findings for LSTMs on a noisy 0db AO test set: training an 8 layer LSTM on AO resulted in 46.38 *WER*, but on the same test set, training the same model on AV actually resulted in 51.53 *WER*, which is an 11% increase over training on AO. Using video dropout only partially rectifies the problem, yielding 48.13 *WER*, which is still a 3.8% increase over training on AO. These results violate Intuition 1, since the addition of video information at training time, whether dropout is used or not, actually resulted in worse test performance on AO. Surprisingly, we found that unlike prior architectures in the literature, conformers can attain robustness with the use of dropout. This is a fortuitous and notable empirical finding, because state-of-the-art architectures in both AO and AV ASR are based on conformers (Gulati et al., 2020; Ma et al., 2021a;b).

**c) Cascaded Audiovisual Models**    In general, a principled and architecture-agnostic approach to robustness is needed. It may not always be practical to use conformers due to their exorbitant memory requirements. Besides the choice of architecture, the success of dropout in producing a robust model also seems to hinge on ad-hoc empirical factors like noise conditions or the quality of features in the training and test datasets.

Fortunately, the asymmetry of our problem (only robustness to missing video, and not missing audio, is desired) presents a unique opportunity. When the video is absent, a simple way to guarantee the robustness of an AV model is to have it produce the same predictions as an AO model. But when the video is present, the model can improve its predictions by fusing the acoustic representations with the visual ones. This central insight motivates our proposal of a cascaded audiovisual model: stack an AV model on top of an AO model, and route inputs via the AO path or the AV path (cf. Figure 1b). There is a wealth of literature that points to the architecture-agnostic nature of cascades: LSTMs and Conformers for ASR in Narayanan et al. (2021), LSTMs and Transformers for ASR in Shi et al. (2020), and ViT, EfficientNet, ResNet, MobileNetV2, and X3D for computer vision in Wang et al. (2020). We confirm that this finding also applies to the robustness problem by conducting comprehensive experiments spanning the vast majority of architectures used in AV ASR: using LSTMs and Conformers (with Transformers omitted because they are roughly convolution-free conformers) for the encoder, as well as both concatenation-based and attention-based audiovisual fusion methods. The proposed cascaded model was found to be robust under a wide variety of architectural combinations (for example when the base encoder and cascaded encoder were of different architectures) on a range of test sets and noise conditions, even when existing techniques like dropout fail to achieve robustness. While cascades are not new, the proposal to use them as a solution for robustness is novel and significant given that prior to our work, variants of dropout have been the only proposed technique for robustness, even when it had already been noted in Makino et al. (2019) for example that dropout did not achieve robustness in some settings.

Besides being conceptually simple and architecture-independent, cascaded models also enjoy numerous other advantages: they can be trained in one pass, are applicable to streamable ASR models, and provide interpretable representations for the different modalities. The simplicity of cascades belies the complexity of the representation learning they do. We observed that jointly training both the AO and AV parts of the cascaded model yielded superior performance on both AO and AV test sets compared to first training the AO part, freezing its weights, and then training the cascaded AV part (even though the latter takes twice as much training time). Interestingly, we also saw that robustness was consistently achieved by cascading the entire video, but not cascading a partial collection of individual video frames, even though the test suites mostly contained partially, not entirely, missing video. Given the relative scarcity of high quality AV training data compared to AO, cascading an AV model over a frozen AO model is an embarrassingly simple recipe for turning an arbitrary pre-trained AO model into a robust AV model, thus opening up the wide gamut of existing AO ASR literature to the promise of audiovisual learning.

### 1.2 Organization for the Rest of the Paper

The remainder of the paper is organized as follows: Section 2 provides an overview of audiovisual speech recognition. Section 3 introduces dropout and cascades. Section 4 reviews prior claims to robustness in the literature, and defines a new mathematical framework for reasoning about robustness. Section 5 presents our experimental setup, and Section 6 presents the results of those experiments. Section 7 acknowledges the limitations of our work. Finally, Section 8 concludes the paper.

## 2 Audiovisual Speech Recognition Methodology

Audiovisual speech recognition is the task of transcribing an audiovisual clip of speech, also known as an utterance, into text. Formally, we define an audiovisual speech recognition model $\mathcal{M} \colon \mathbb{A}, \mathbb{V} \to \mathbb{Y}$ as a function that consumes an audio input of waveforms or spectrograms and a video input of mouth or face tracks, and produces a natural language transcript of what was said. The audio input $a \in \mathbb{A}$ is a sequence of $n_a$ acoustic frames of dimension $d_\mathbb{A}$ that is represented by a real-valued tensor in $\mathbb{R}^{n_a, d_\mathbb{A}}$. The video input $v \in \mathbb{V}$ is a sequence of $n_v$ visual frames of dimension $d_\mathbb{V}$ that is represented by a real-valued tensor in $\mathbb{R}^{n_v, d_\mathbb{V}}$. The text output $\hat{y} \in \mathbb{Y}$ is a sequence of $n_y$ one-hot vectors of dimension $d_\mathbb{Y}$ (vocabulary size) that is represented by a real-valued tensor in $\mathbb{R}^{n_y, d_\mathbb{Y}}$. Our formulation is general, because all existing AO ASR models can be considered as special cases of AV ASR models that do not use the video input.

For a given data point $(a, v)$, we can assume $n_a = n_v$, because synchronizing the audio and video features is a necessary pre-processing step for AV ASR (Chung et al., 2017; Afouras et al., 2018; Makino et al., 2019). The two modalities $a$ and $v$ are typically combined on a frame level to form input sequence $x$,

i.e. $\forall i \in [1, n_a] : x_i = Fuse(a_i, v_i)$, with concatenation (CAT) being the most common method of fusion in the literature and cross-modal attention (CM) being the alternative (Wei et al., 2020).

## 2.1 Training: RNN-T

The RNN-T loss has become a popular approach for training deep ASR models, because it is streamable unlike sequence-to-sequence losses, and allows the model to produce output tokens conditionally dependent on the history of previous tokens unlike CTC or the cross-entropy loss. It can be written as follows.

$$\mathcal{L}_{\text{RNN-T}} = \sum_{\hat{y} \in \mathcal{A}_{\text{RNN-T}}(x,y)} \prod_{i=1}^{T+U} P(\hat{y}_i | x_1 \ldots x_{t_i}, y_0 \ldots y_{u_{i-1}}), \tag{1}$$

where the alignments $\mathcal{A}_{\text{RNN-T}}(x, y)$ refer to the set of all possible sequences of $T$ blanks and $U$ labels.

An audiovisual speech model is generally factored into two separate components: a language model (also called the decoder) and an audiovisual model (also called the encoder). The RNN-T loss is then computed from these two components using a joiner (cf. Figure 1a). Given input sequence $x$, the probability of the transcript $y$ can be calculated as follows.

$$P(y|x) = Joiner(LM(y), Encoder(a, v)),$$
$$Encoder(a, v) = AVM(Fuse(a, v)). \tag{2}$$

We refer the interested reader to Graves (2012); He et al. (2019) for more information on the RNN-T loss.

For our study on robustness, we assume that we have a parallel corpus of AV data, i.e. both the audio and video are present in every training data point. If the model $\mathcal{M}$ is given full access to the training data, we refer to the trained model as $\mathcal{M}_{AV}$. If $\mathcal{M}$ is given access to only the audio portion of the data, we refer to the trained model as $\mathcal{M}_{AO}$, and call its test performance the *AO Baseline* of the model.

## 2.2 Testing: Word Error Rate

The most common metric for evaluating ASR performance is the word error rate. For a given labeled data point $(a, v, y)$ and model output $\hat{y} = \mathcal{M}(a, v)$, the word error rate of $\hat{y}$ is the ratio of substitutions, deletions, and insertions in $\hat{y}$ to the number of words in $y$: $WER(\hat{y}, y) := \frac{S+D+I}{W}$. The word error rate of a model $\mathcal{M}$ over a given test distribution *Test* is $WER(\mathcal{M}, Test) := \mathbb{E}_{(a,v,y) \sim Test}[WER(\mathcal{M}(a, v), y)]$.

# 3 Techniques for Robustness

The encoder in Equation (2) can be modified to expose the model to missing video during training time, so as to prepare it to handle missing video at test time.

## 3.1 Existing Work: Dropout

Makino et al. (2019) proposed to randomly drop the entire video utterance (Dropout Utt), while Zhang et al. (2019) proposed to randomly drop each video frame (Dropout Frame). Dropout can be implemented by sampling Bernoulli random variables and replacing $v$ in Equation (2) with $v'$ at training time. For Dropout Utt, a Bernoulli is sampled per utterance, while for Dropout Frame, a Bernoulli is sampled per frame.

$$\text{Utt: } v_i' := z v_i, \qquad\qquad z \sim Bernoulli(p). \tag{3}$$
$$\text{Frame: } v_i' := z_i v_i, \qquad\qquad z_i \sim Bernoulli(p). \tag{4}$$

There are also proposals to apply dropout simultaneously on both the audio and video features at the utterance level (AV Dropout Utt) (Chung et al., 2017; Shi et al., 2022).

$$(a_i', v_i') := \mathbb{1}_{z=0}(a_i, v_i) + \mathbb{1}_{z=1}(a_i, 0) + \mathbb{1}_{z=2}(0, v_i), \quad z \sim Multinomial(p,q,r). \tag{5}$$

While dropout can make a model robust, we observed in the literature and on our own experiments that it can be sensitive to the choice of architecture or the specific test sets used. This is because training on a mixture of AV and AO inputs does not guarantee that the architecture is able to automatically disentangle the AV and AO representations.

### 3.2 Proposed Method: Cascades

Instead of putting this burden on the model's architecture, we introduce an architecture-agnostic method that can benefit from the video information when it is there, and gracefully degrade to the audio-only case when it is not. The basic idea is to split the model explicitly into an acoustic model (AM) and an audiovisual model (AVM), and cascade the AVM on top of the AM (c.f. Figure 1b). For each frame of the input, the model routes it to the AV path if the video frame is present, and to the AO path if not.

$$Cascade(a,v)_i = \begin{cases} AM(a_i) & \text{if } v_i = 0, \\ AVM(Fuse(AM(a_i), v_i)) & \text{otherwise.} \end{cases} \tag{6}$$

Suppose that the cascaded model is trained in two passes (Two-Pass). On the first pass, no video information is made available, i.e. all inputs are routed to the AO path. On the second pass, the entire model except for the AVM is frozen, and all the video information is made available, i.e. inputs are routed to the AV path. Because the parts of the model that activate when it sees AO data are frozen, the cascaded model is guaranteed to never perform worse than its AO baseline regardless of its specific architecture. In fact, as we will show in our experiments, two passes are not needed, and we can train a robust model in one pass by stochastically routing the inputs between the AO and AV paths. Like dropout, the stochastic selection of routes is done by randomly dropping either the video (Cascade Utt) or the video frames (Cascade Frame). Unlike dropout, cascades use the AVM if and only if the video is present, thus explicitly disentangling the AV from the AO representations.

## 4 Robustness Framework

We propose the use of order theory (Davey & Priestley, 2002) as a suitable mathematical language for evaluating robustness in a rigorous way. By abstracting away absolute numbers, order theory allows claims about robustness to be formalized as statements about the relative ranking of a given model's performance under different conditions.

### 4.1 Technical Preliminaries

To begin, we state the definition of a poset, and show how we can make comparisons between pairs of real vectors, vector-valued random variables, and test distributions.

**Definition 4.1.** A poset is a set $\mathcal{P}$ equipped with a binary relation $\leq$ that is i) reflexive: $\forall p \in \mathcal{P}, p \leq p$, ii) anti-symmetric: $\forall p_1, p_2 \in \mathcal{P}, p_1 \leq p_2$ and $p_2 \leq p_1 \implies p_1 = p_2$, and iii) transitive: $\forall p_1, p_2, p_3 \in \mathcal{P}, p_1 \leq p_2$ and $p_2 \leq p_3 \implies p_1 \leq p_3$.

**Definition 4.2.** For real vectors $a$ and $b$, $a \leq b$ if all the elements in $a$ are less than or equals to all the elements in $b$, i.e. $\forall i \in [1, n], a_i \leq b_i$.

**Definition 4.3.** For vector-valued random variables $A$ and $B$, $A \leq B$ if $\mathbb{E}[A] \leq \mathbb{E}[B]$.

To model missing video frames, we can encode them using a (randomly distributed) binary mask $p$. The comparison between two test distributions can then be reduced to comparing the missing video probabilities for each frame.

**Definition 4.4.** For a test distribution $Test = (A, V, Y)$, and a pair of real vectors (or vector-valued random variables) $p_1, p_2$, $(A, p_1 \cdot V, Y) \leq (A, p_2 \cdot V, Y)$ if $p_1 \leq p_2$.

**Proposition 4.5.** *A poset of test distributions can be constructed from an underlying poset $\mathcal{P}$ as follows:* $C(\mathcal{P}) := \{(A, p \cdot V, Y)\}_{p \in \mathcal{P}}$.

### 4.2 Prior Work

After a careful survey of the AV (ASR) literature, we found that there are at least three distinct claims made about AV models, as exemplified by the first three rows of Table 1. We explain why they are not sufficient claims to robustness.

For a test distribution $(A, V, Y)$, let $Test_{AV} := (A, V, Y)$ denote testing in the presence of the entire video, and $Test_{AO} := (A, 0, Y)$ denote testing in its absence. The first claim is that $Metric(\mathcal{M}_{AV}, Test_{AV}) \leq Metric(\mathcal{M}_{AV}, Test_{AO})$, e.g. Chung et al. (2017); Parthasarathy & Sundaram (2020); Xuan et al. (2020). Notice that this can be true if only for the trivial reason that testing $\mathcal{M}_{AV}$ on out-of-distribution data $Test_{AO}$ leads to worse generalization. The second claim is that $Metric(\mathcal{M}_{AV}, Test_{AV}) \leq Metric(\mathcal{M}_{AO}, Test_{AO})$, e.g. Afouras et al. (2018); Zhou et al. (2019); Xu et al. (2020); Ma et al. (2021b). This is not a sufficient claim to robustness, because it can be true even if the model suffers a catastrophic degradation in performance when trained on AV but tested on AO, as is the case for example in rows 4 and 5 of Table 1. The third claim is that $Metric(\mathcal{M}_{AV}, Test_{AO}) \leq Metric(\mathcal{M}_{AO}, Test_{AO})$, e.g. Ngiam et al. (2011); Shi et al. (2022). This claim does not articulate the possibility of improved performance if the video modality was present at test time.

One way to formalize robustness might be to combine the above three claims into $Metric(\mathcal{M}_{AV}, Test_{AV}) \leq Metric(\mathcal{M}_{AV}, Test_{AO}) \leq Metric(\mathcal{M}_{AO}, Test_{AO})$. But this is also not an adequate definition that captures real-world settings where the visual modality is only partly missing, i.e. some video frames are dropped, but not all of them. In what follows, we present a comprehensive framework for evaluating robustness, and show that the combination of the three claims is a special case of robustness to $\mathcal{T}_{utt}$.

### 4.3 Definition of Robustness

Intuition 1 requires that an audiovisual model not do worse than an audio-only model, given additional visual information at training or test time. This motivates the following definition of robustness.

**Definition 4.6.** A model $\mathcal{M}$ is robust to a poset of test distributions $\mathcal{T}$ if it has both
**1) Train-Time Robustness:** $\forall T \in \mathcal{T}, WER(\mathcal{M}_{AV}, T) \leq WER(\mathcal{M}_{AO}, T)$, and
**2) Test-Time Robustness:** $\forall T_i, T_j \in \mathcal{T}, T_i \leq T_j \implies WER(\mathcal{M}_{AV}, T_i) \leq WER(\mathcal{M}_{AV}, T_j)$.

Because each of the two properties is defined with respect to some set of testing conditions, it follows that robustness is not a universal property of the model, but rather a statement about its performance on a given test suite. This key feature of our definition allows it to be reified into concrete, empirically testable claims on specific test sets.

An important detail of our definition is that there is an asymmetry: train-time robustness involves a coarse comparison between just $\mathcal{M}_{AV}$ and $\mathcal{M}_{AO}$, while test-time robustness allows for very fine-grained comparisons depending on the size of the poset. This reflects both theoretical and practical requirements. At training time, methods like dropout intentionally discard some video information for the purpose of simulating test time conditions, but at test time, it is imperative that all the available video information is used. Thus, it is not meaningful to distinguish between training methods that use different amounts of video information (e.g. dropout with $p = 0.5$ instead of $p = 0.25$), but absolutely essential to distinguish between the performance of a model tested with half of the video frames missing versus a quarter of the frames missing. Furthermore, training a model is significantly more computationally demanding than running inference, so it is necessary that the number of comparisons needed to evaluate train-time robustness be a lot smaller than what is needed to evaluate test-time robustness.

Our definition also implies the following corollary, which shows why the two properties are not merely necessary, but also jointly sufficient by spanning the entire space of training and test conditions.

**Corollary 4.7.** *Let $\mathcal{M}$ be a model robust to $\mathcal{T}$. Then, $\forall T_i, T_j \in \mathcal{T}, T_i \leq T_j \implies WER(\mathcal{M}_{AV}, T_i) \leq WER(\mathcal{M}_{AO}, T_j)$.*

### 4.4 Test Suites of Missing Video

We show how our proposed definition can be used to ascertain model robustness under distinct settings of missing video. Let $n$ be the number of video frames in a speech utterance. The naive approach is to test all combinations of the presence or absence of every video frame.

**Test Suite 4.8.** $\mathcal{T}_{frame} := C(\{0,1\}^n)$.

But this is expensive to run, due to the exponential number of test distributions. Luckily, we can dramatically reduce the number of tests by focusing on pragmatic scenarios common to AV ASR like randomly dropped video, video dropped in a contiguous way, and video dropped at a constant rate. We use a running example of transcribing an online meeting to motivate and illustrate the following test suites.

The most common cause of missing video in online meetings is when the user turns off their camera.

**Test Suite 4.9.** $\mathcal{T}_{utt} := C(\{0^n, 1^n\})$.

We can generalize $\mathcal{T}_{utt}$ to $\mathcal{T}_{BerUtt}$ to capture the scenario where the user turns their camera on with probability $r$.

**Test Suite 4.10.** *Let $BerUtt(r)$ be a vector-valued random variable parameterized by a scalar $r \in [0,1]$.*

$$
\begin{aligned}
P(BerUtt(r) = 1^n) &= r, \\
P(BerUtt(r) = 0^n) &= 1 - r.
\end{aligned}
\tag{7}
$$

$\mathcal{T}_{berUtt}(r) := C(\{BerUtt(r)\}_{r \in R})$. *In our experiments, $R = \{0, \frac{1}{4}, \frac{1}{2}, \frac{3}{4}, 1\}$.*

Another cause of missing video in online meetings is an unreliable internet connection. The more unreliable it is, the more likely that internet packets for individual video frames will be dropped. We can simulate this condition using i.i.d. Bernoulli random variables for each frame.

**Test Suite 4.11.** *Let $BerFrame(s)$ be a vector-valued random variable parameterized by a scalar $s \in [0,1]$.*

$$
\begin{aligned}
\forall i \in [1,n], P(BerFrame(s)_i = 1) &= s, \\
P(BerFrame(s)_i = 0) &= 1 - s.
\end{aligned}
\tag{8}
$$

$\mathcal{T}_{berFrame}(S) := C(\{BerFrame(s)\}_{s \in S})$. *In our experiments, $S = \{0, \frac{1}{4}, \frac{1}{2}, \frac{3}{4}, 1\}$.*

Video frames can also be dropped in a contiguous segment from the start, in the middle, or at the end, depending on when the user decides to move off screen and when they re-enter. We can simulate different amounts of missing video in each case with a deterministic binary mask.

**Test Suite 4.12.** *Let $\alpha_{a:b}$ denote a binary vector where the $i$th element is $0$ iff $i \in [a \cdot n + 1, b \cdot n]$. $\mathcal{T}_{con}(A, B) := C(\{\alpha_{a:b}\}_{(a,b) \in (A,B)})$. In our experiments, $\mathcal{T}_{start} := \mathcal{T}_{con}(\{(0,0), (0, \frac{1}{4}), (0, \frac{1}{2}), (0, \frac{3}{4}), (0,1)\})$, $\mathcal{T}_{mid} := \mathcal{T}_{con}(\{(\frac{1}{2}, \frac{1}{2}), (\frac{3}{8}, \frac{5}{8}), (\frac{1}{4}, \frac{3}{4}), (\frac{1}{8}, \frac{7}{8}), (0,1)\})$, $\mathcal{T}_{end} := \mathcal{T}_{con}(\{(0,1), (\frac{1}{4}, 1), (\frac{1}{2}, 1), (\frac{3}{4}, 1), (1,1)\})$.*

If the hardware has competing demands for its compute, it might decode the video at a smaller frame rate, which leads to video frames being dropped at a constant rate.

**Test Suite 4.13.** *Let $\beta_k$ denote a binary vector where the $i$th element is $0$ iff $i$ is a multiple of $\frac{1}{k}$. $\mathcal{T}_{rate}(K) := C(\{\beta_k\}_{k \in K})$. In our experiments, $K = \{0, \frac{1}{128}, \frac{1}{32}, \frac{1}{8}, \frac{1}{2}, 1\}$.*

Finally, we can unify all these test suites.

**Test Suite 4.14.** $\mathcal{T}_{all} := \mathcal{T}_{berUtt} \cup \mathcal{T}_{berFrame} \cup \mathcal{T}_{start} \cup \mathcal{T}_{mid} \cup \mathcal{T}_{end} \cup \mathcal{T}_{rate}$.

This list of test suites (visualized in Figure 2) is not exhaustive, but serves as a proof of concept for how extensible our framework is — by changing the poset of test distributions, we change the kind of robustness that we are testing. Our proposed test suites are not only qualitatively different, they also cover both easy and hard cases for ASR. For example, dropping every second frame makes for an easy test suite, while dropping a contiguous segment of half the frames makes for a challenging one. This is intuitive because for a given word, having partial visual context for it makes the ASR task significantly easier than having no visual context.

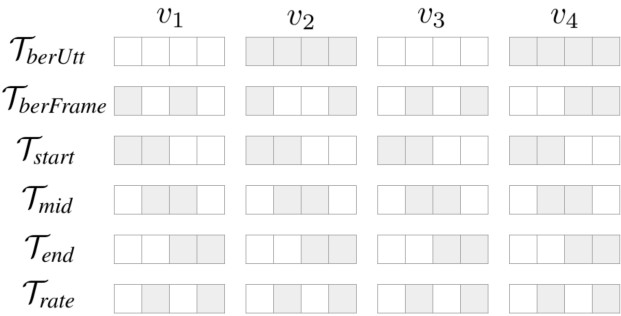

Figure 2: Dropping 50% of the frames under different test suites.

## 5 Experimental Setup

**Audiovisual Features**   The acoustic features are computed as log-scaled mel spectrograms sampled at 16kHz using a Hann window of size 25ms and step 10ms, with every three consecutive spectrograms stacked into one audio frame with dimension 240. At training time, we apply multi-style training for data augmentation (Cui et al., 2015). The visual features are computed by extracting mouth tracks from the video, re-sampling them at the frequency of the acoustic features (to ensure that both modalities have the same sequence length), cropping them to 32x32, and encoding them using a VGG into video frames of dimension 256. Because the visual features are extracted using an independent face tracker like in Afouras et al. (2018); Makino et al. (2019), the AV ASR model knows whether the video frame is missing or not. The VGG is not pre-trained or frozen, and its parameters are learned during the training process. Our models use a sequence length of up to $n = 512$, which corresponds to approximately 15 seconds of audiovisual speech.

**Training Data**   We closely adhere to the process outlined by Makino et al. (2019) to create a large-scale dataset containing $100,000$ hours of AV data from public YouTube videos. This is done by mining segments of videos where the force-aligned user uploaded transcript matches a production quality ASR system with high confidence. Then, SyncNet (Chung & Zisserman, 2016) is applied as a filtering step to ensure that the video track also matches the audio track with high confidence. Unlike Makino et al. (2019), we synchronize the frame rate of the video to match the audio, instead of vice versa, to ensure that we do not affect the AO baseline. As explained in Section 2.1, we train our models under two settings: $\mathcal{M}_{AO}$ and $\mathcal{M}_{AV}$.

**Test Data**   As discussed in Section 4.3, robustness should be treated not as a universal property of a model, but as an empirical claim about its performance on a particular test set. We benchmark our models on a separate test set of YouTube videos that contains 550 hours of professionally transcribed audiovisual clips ($27,353$ utterances and $342,507$ words), under varying amounts of artificially added babble noise (clean, 20db, 10db, 0db) from the NoiseX corpus (Varga & Steeneken, 1993). As explained in Section 4.4, we evaluate our models on six test suites: $\mathcal{T}_{berUtt}, \mathcal{T}_{berFrame}, \mathcal{T}_{start}, \mathcal{T}_{mid}, \mathcal{T}_{end}, \mathcal{T}_{rate}$. Both the training and test data were collected by following the Google AI principles (goo).

**Statistical Methodology**   Our study of robustness focuses on the relative ranking between the WER results of a given model under different test conditions, rather than the absolute WER numbers. That said, in practice, comparisons between any two results have to take into account statistical variance and uncertainty. To this end, we closely follow the prior AV ASR methodology of Makino et al. (2019) in assembling a large-scale audiovisual dataset for training and calculating confidence intervals for the test WER. If either of two results falls within the other's 95% confidence interval, we consider them equivalent, otherwise, we consider one result better than the other.

**Methods for Robustness**   We compare between the following methods: Audio Baseline, Vanilla, Vanilla (25L), Cascade Utt, Dropout Utt, Cascade Frame, Dropout Frame, AV Dropout Utt, and Two-Pass. While Section 3 contains a self-contained description for most of these methods, we re-iterate them here for the reader's convenience.

Audio Baseline: The model is trained on audio-only inputs.

Vanilla: The model is trained on audio-visual inputs.

Vanilla (25L): Same as the Vanilla model, but with 25 layers.

Cascade Utt: The model randomly drops the entire video. Audio-only inputs are routed to the audio encoder, while audio-visual inputs are routed to the audio-visual encoder.

Dropout Utt: The model randomly drops the entire video. Inputs with missing video are consumed by the model directly.

Cascade Frame: The model randomly drops each video frame. Audio-only inputs are routed to the audio encoder, while audio-visual inputs are routed to the audio-visual encoder.

Dropout Frame: The model randomly drops each video frame. Inputs with missing video are consumed by the model directly.

AV Dropout Utt: The model randomly drops the entire video. The model also randomly drops the entire audio. Inputs with missing audio or video are consumed by the model directly.

Two-Pass: Same architecture as the Cascade Utt and Cascade Frame models, but the model is trained in two passes. On the first pass, audio-only inputs are consumed, and all model weights are trainable. On the second pass, audio-visual inputs are consumed, but only the audio-visual encoder is trainable.

**Architectural Configurations**    All models use a two-layer bidirectional LSTM with hidden dimension 2048 for the decoder, and a one-layer MLP with hidden dimension 640 for the joiner. The decoder uses an English character-based vocabulary, with an embedding dimension of 128 and a beam width of size 8. The architectural differences between the different models reside in the encoder.

Conformer CAT: The encoder is a conformer that uses concatenation to fuse the audiovisual features. For cascaded models, both the base and the cascaded encoder are conformers.

Conformer CM: The encoder is a conformer that uses cross-modal attention, specifically the one used in Afouras et al. (2018), to fuse the audiovisual features. For cascaded models, both the base and the cascaded encoder are conformers.

Con-LSTM CAT: This is only used in the context of the cascaded model, with a conformer as the base encoder and an LSTM as the cascaded encoder. Concatenation is used to fuse the audiovisual features.

LSTM CAT: The encoder is an LSTM that uses concatenation to fuse the audiovisual features. For cascaded models, both the base and the cascaded encoder are LSTMs.

LSTM-CON CAT: This is only used in the context of the cascaded model, with an LSTM as the base encoder and a conformer as the cascaded encoder. Concatenation is used to fuse the audiovisual features.

The conformer encoders are configured with 17 layers, full context attention, model dimension 512, 8 attention heads, convolutional kernel size 32, no dropout (this refers to regular neural network dropout instead of video dropout), and group normalization with 32 groups in the place of layer normalization. The Cascade Utt, Cascade Frame, and Two-Pass models use 17 conformer layers for the base AM and 8 conformer layers for the cascaded AVM. The Vanilla (25L) model uses 25 conformer layers instead of 17.

The LSTM encoders are configured with 8 bidirectional layers, model dimension 512 (for each direction), and weight normalization. The Cascade Utt model uses 8 LSTM layers for the base AM and 4 LSTM layers for the cascaded AVM.

**Optimization**    All our models are trained in exactly the same way: Adam with $\beta_1 = 0.9, \beta_2 = 0.97$, batch size 4096, and learning rate 0.001 for a total of 500k steps with linear warmup in the first 10k steps and an exponential decay to the smaller learning rate of 0.0001 from steps 300k to 400k. Two-Pass uses the same optimization setup for both training passes.

# 6 Results

To begin, we walk the reader through examples in Table 2 to show how we establish the robustness of a model. Cascade Utt is robust because the WER increases monotonically as more frames are dropped (Test-Time Robustness), while always staying below its AO baseline of 33.54 (Train-Time Robustness). On the other hand, the vanilla model is not robust, because when tested on the setting where all the frames are dropped, its WER of 35.51 exceeds its AO baseline, thus violating Train-Time Robustness. Dropout Frame is not robust, because it obtains lower WERs of 27.11 and 26.51, compared to 27.58, by dropping more frames at test time, which violates Test-Time Robustness.

**Vanilla AV ASR Models are Not Robust** The vanilla conformer and LSTM models were found to not be robust under most settings tested (cf. Table 3). Figure 3 shows the WER of a vanilla model increasing as the number of missing video frames at test time increases, eventually doing worse than its AO baseline. This underscores why the study of robustness is necessary: we would like our AV models to always do at least as well as the AO models regardless of video availability at test time.

**Type of Missing Video Affects Performance** For the same quantity of missing video, the manner in which the frames are dropped can significantly affect how quickly the performance of a model degrades. In Figure 3, we observe a relationship between WER and amount of dropped frames that is linear for $\mathcal{T}_{berUtt}$, $\mathcal{T}_{start}$, $\mathcal{T}_{mid}$, $\mathcal{T}_{end}$ and convex for $\mathcal{T}_{berFrame}$, $\mathcal{T}_{rate}$. In other words, the model is fairly tolerant of frames being randomly dropped or dropped at a constant rate, but not when dropped in a contiguous segment. This accords with the intuition from our discussion in Section 4.4.

**Tuning Sampling Hyper-Parameters** For the utterance methods, we ran a hyper-parameter search over $p \in \{0.25, 0.5, 0.75\}$. In every setting tested, we found that $p = 0.25$ was optimal for Cascade Utt, and $p = 0.5$ was optimal for Dropout Utt, i.e. it was never the case that setting an alternative $p$ led to robustness if this did not. This finding supports the motivation for the asymmetry in our definition of robustness in Section 4.3: it is not pertinent that $p = 0.25$, $p = 0.5$ were trained on less video information than $p = 0.75$, but noteworthy that they were better able to withstand the loss of video information at test time. For the frame methods, we searched $p \in \{0.1, 0.2\}$, and found $p = 0.1$ to be optimal for both Cascade Frame and Dropout Frame. For AV Dropout Utt, we found $p = \frac{1}{2}, q = \frac{1}{4}, r = \frac{1}{4}$, which is used by Shi et al. (2022), to be optimal over $p = \frac{1}{3}, q = \frac{1}{3}, r = \frac{1}{3}$, which is used by Chung et al. (2017).

**Cascade Utt is Consistently Robust** Like Makino et al. (2019), we saw that Dropout Utt can degrade the performance of an LSTM CAT to be worse than its AO baseline. This degradation is especially pronounced for Conformer CM (cf. Figure 4). Across all the architectural configurations, four noise conditions, and six test suites, Cascade Utt was the only technique found to be consistently robust (cf. Table 3). This finding held even when the base and cascaded encoder are of different architectures, like Con-LSTM CAT and LSTM-Con Cat, which points to the architecture-agnostic nature of cascaded models.

**Dropout Utt Performs Well on Conformer CAT** The main caveat to Cascade Utt is that while always robust, it does not always produce the lowest absolute WER. In Figure 4, we see that when tested on full AV in the 0db setting, Cascade Utt actually performs worse than Dropout Utt and the vanilla model for the concatenation models. This finding emphasizes that the evaluation of robustness, which consists of relative comparisons for a fixed model, is independent from the evaluation of absolute WER. It is thus fortuitous that Dropout Utt is both robust and has lower WER than the vanilla conformer CAT, which is a state-of-the-art architecture for AV ASR (Ma et al., 2021a;b).

**Frame Methods Underperform Utterance Methods** In general, the frame methods are not as robust as the utterance methods. This can perhaps be expected because training augmentations often do not generalize to unseen test corruptions (Mintun et al., 2021). For example, in Table 2, Cascade Frame underperforms its AO baseline. This might be because even if the probability of dropping each of 512 frames is high (90%), the model is nevertheless rarely exposed to an empty video at training time ($0.9^{512} = 10^{-24}$).

**Cascade Utt Achieves its Model Capacity** Cascade Utt was found to have the same or strictly lower WER than Two-Pass for all test suites on all quantities of missing video tested. This means that independent of the capacity of the cascaded model to achieve robustness, it is also important to consider how we optimize

it. Not only is it not necessary to achieve robustness by training in two passes, doing it in only one pass (i.e. using half the resources) can actually result in better absolute WER performance.

We train the cascaded model with just AO and AV data respectively to measure the capacity of the AM and AVM sub-models. In Figure 5, we note that Cascade Utt is able to achieve the capacity of both sub-models, and segue between them as frames are dropped during test time. For conformer CAT, a 17 layer AM cascaded with an 8 layer AVM performs equivalently on full AV data at test time to a vanilla 25 layer AVM at all noise conditions except 0db. This implies that unlike adversarial robustness, which is gained at a necessary cost to model size or performance on clean data (Bubeck & Sellke, 2021), robustness to missing video can be achieved via cascades without using any extra resources or trading off test performance on AV.

Table 2: Robustness to $\mathcal{T}_{rate}$ for 0db. Columns three to seven represent the amount of dropped video frames. The WER results in bold cause the model to be considered non-robust.

| Architecture | Method | 0 | $\frac{1}{32}$ | $\frac{1}{8}$ | $\frac{1}{2}$ | 1 | Robust |
|---|---|---|---|---|---|---|---|
| Conformer CAT | Audio Baseline | $33.54 \pm 0.43$ | $33.54 \pm 0.43$ | $33.54 \pm 0.43$ | $33.54 \pm 0.43$ | $33.54 \pm 0.43$ | - |
| Conformer CAT | Vanilla | $24.96 \pm 0.35$ | $24.95 \pm 0.35$ | $25.08 \pm 0.35$ | $25.90 \pm 0.35$ | $\mathbf{35.51 \pm 0.43}$ | ✗ |
| Conformer CAT | Cascade Utt | $26.12 \pm 0.36$ | $26.25 \pm 0.36$ | $26.72 \pm 0.37$ | $28.73 \pm 0.39$ | $31.08 \pm 0.40$ | ✓ |
| Conformer CAT | Dropout Utt | $24.33 \pm 0.34$ | $24.36 \pm 0.34$ | $24.49 \pm 0.34$ | $26.66 \pm 0.36$ | $31.67 \pm 0.40$ | ✓ |
| Conformer CAT | Cascade Frame | $28.04 \pm 0.38$ | $28.08 \pm 0.38$ | $28.23 \pm 0.39$ | $30.14 \pm 0.42$ | $\mathbf{34.17 \pm 0.44}$ | ✗ |
| Conformer CAT | Dropout Frame | $27.58 \pm 0.37$ | $\mathbf{27.11 \pm 0.36}$ | $\mathbf{26.51 \pm 0.36}$ | $27.56 \pm 0.38$ | $32.54 \pm 0.41$ | ✗ |
| Conformer CAT | AV Dropout Utt | $23.64 \pm 0.33$ | $23.63 \pm 0.33$ | $23.80 \pm 0.33$ | $25.36 \pm 0.35$ | $33.55 \pm 0.41$ | ✓ |
| Conformer CAT | Two-Pass | $30.06 \pm 0.42$ | $30.11 \pm 0.42$ | $30.47 \pm 0.43$ | $31.91 \pm 0.43$ | $33.54 \pm 0.43$ | ✓ |

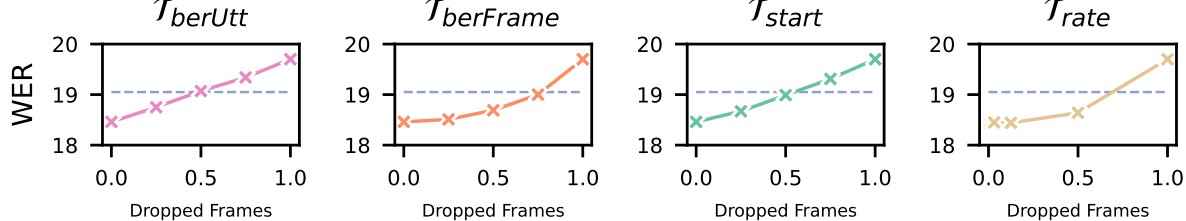

Figure 3: The X-marked lines show the WER of a vanilla 17 layer concatenation-based conformer model trained on AV, and tested on 10db with different test suites. The dashed line denote the performance of the same model trained on AO, i.e. its AO baseline. The vanilla AV model performs worse as more video frames are dropped at test time, eventually doing worse than its AO baseline.

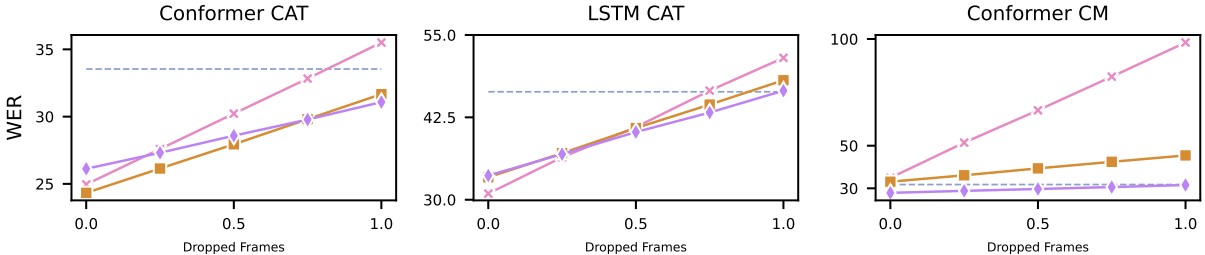

Figure 4: The X-marked, square-marked, diamond-marked, and dashed lines respectively show the WERs for the vanilla model, Dropout Utt, Cascade Utt, and the AO baseline tested on 0db using $\mathcal{T}_{berUtt}$ across three architectures.

Table 3: Robustness to $\mathcal{T}_{all}$ across different experimental settings. Cascade Utt is the only method found to be consistently robust across all settings tested.

| ARCHITECTURE, METHOD | CLEAN | 20DB | 10DB | 0DB |
|---|---|---|---|---|
| CONFORMER CAT, VANILLA | ✗ | ✗ | ✗ | ✗ |
| CONFORMER CAT, **Cascade Utt** | ✓ | ✓ | ✓ | ✓ |
| CONFORMER CAT, DROPOUT UTT | ✓ | ✓ | ✓ | ✓ |
| CONFORMER CAT, CASCADE FRAME | ✓ | ✓ | ✓ | ✗ |
| CONFORMER CAT, DROPOUT FRAME | ✓ | ✓ | ✓ | ✗ |
| CONFORMER CAT, AV DROPOUT UTT | ✓ | ✗ | ✓ | ✓ |
| CONFORMER CAT, TWO-PASS | ✓ | ✓ | ✓ | ✓ |
| CON-LSTM CAT, **Cascade Utt** | ✓ | ✓ | ✓ | ✓ |
| LSTM CAT, VANILLA | ✓ | ✓ | ✗ | ✗ |
| LSTM CAT, **Cascade Utt** | ✓ | ✓ | ✓ | ✓ |
| LSTM CAT, DROPOUT UTT | ✓ | ✓ | ✓ | ✗ |
| LSTM-CON CAT, **Cascade Utt** | ✓ | ✓ | ✓ | ✓ |
| CONFORMER CM, VANILLA | ✗ | ✗ | ✗ | ✗ |
| CONFORMER CM, **Cascade Utt** | ✓ | ✓ | ✓ | ✓ |
| CONFORMER CM, DROPOUT UTT | ✗ | ✗ | ✗ | ✗ |

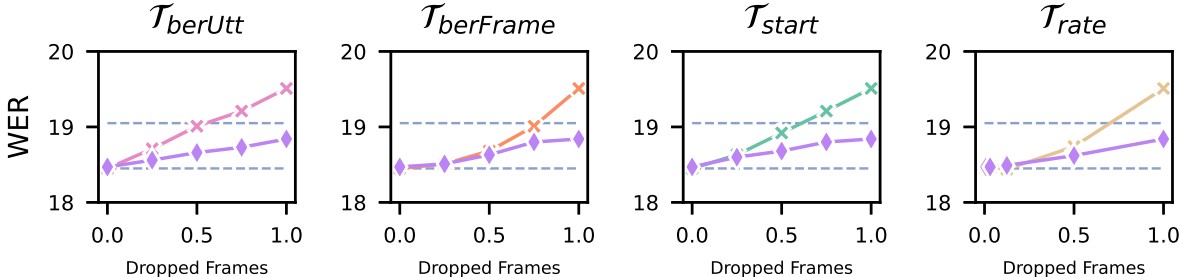

Figure 5: The X-marked and diamond-marked lines respectively show the WER of a vanilla 25 layer conformer AVM and a 25 layer cascaded conformer model (17 AM + 8 AVM layers) tested on 10db. Both models use concatenation for audiovisual fusion. The top and bottom dashed lines respectively denote the performance of the cascaded model when all its inputs are forcefully routed to the AO and the AV paths during both training and testing.

## 7 Limitations of Our Work

There are several limitations of our work that should be improved and addressed in future work. First, the proposed robustness framework only makes comparisons between the relative performance of a model under different test settings. This ignores the absolute WER performance of a given model, which is another important, although orthogonal, evaluation metric. Second, the framework makes binary evaluations: a model is either robust or not robust. A more general framework might be able to allow for more fine-grained evaluations that makes it easier to quantify and compare the robustness of a set of given models. A possible solution is to count the number of test suites under which a given model is robust. Third, the current work only studies robustness to missing video and not other common video corruptions like jitter and blur. Fourth, the cascade methods require knowledge that the video frame is missing in order to route the inputs, while the dropout methods do not. Because the mouth tracks are extracted with an independent face tracker (as explained in Section 5), this information is made known to the AV ASR model. A possible solution is to integrate the face tracker into an end-to-end system so that the model has to now also make decisions about whether the video frame is missing or not. Please see Appendix D for an extended discussion of this issue.

## 8 Conclusion

In this paper, we developed the first principled framework for evaluating robustness to missing video for audiovisual speech recognition. By providing a rigorous definition for robustness, our work fills a longstanding gap in the literature by making it easy for claims about robustness to be specified in an empirically testable way. Our extensive empirical experiments also show that cascaded models are a reliable method to achieve

robustness under a wide variety of architectural configurations and test suites, including in settings where existing techniques like dropout fall short. We hope that our work will spur the development of more robust models and techniques in the AV ASR and multi-modal learning communities.

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

# Appendix

## A  Overview

The appendix is organized as follows: Appendix B is a broader impact statement. Appendix C states the source of the numbers in Table 1. Appendix D provides an in-depth discussion regarding the assumption in our work that the AV ASR model possesses knowledge that the video frames are missing. Appendix E documents our results in the main paper in a more comprehensive manner. In particular, we plot the graphs of our main experiments shown in part in Figures 3 to 5 in their expanded form, and provided further analysis. We also document the absolute WER numbers and their confidence intervals across our extensive experimental settings, in a similar format to that of Table 2 in the main paper. Appendix F shows the results of additional experiments on the TED LRS3 dataset. In the Supplementary materials, we have attached an example script written in tensorflow code to show how our robustness test suites can be generated.

## B  Broader Impact Statement

By the nature of the dataset collection process, the performance of the model with under-represented groups is not being measured and that optimizing for the specific benchmark metrics proposed here may have unknown effects (either positive or negative) on the performance of models within such groups.

## C  About Table 1

The first row comes from Table 5 in Chung et al. (2017). The second row comes from Table 2 in Zhou et al. (2019). The third row comes from Table 4 in Shi et al. (2022). The fourth row comes from Tables 2 and 3 in Makino et al. (2019). The fifth row comes from Tables 1 and 2 in Zhang et al. (2019).

## D  Knowledge of Missing Video Frames

In our paper, we assume that the AV ASR model knows whether a given video frame is missing. This begs the question: since we know that there is missing video, why not just trivially swap to an audio-only model in that case? A naive ensemble containing an audio-only model and an audio-visual model would be robust to missing video by construction. However, there are several issues with this approach. First, the naive ensemble would be significantly more computationally demanding than having a cascaded model in terms of compute, memory, and disk. Second, the naive ensemble would not perform as well in the situation when there is partial video information available, since the audio model operates independently of the video model. It is not feasible to build an ensemble of $2^n$ models where $n$ is the number of frames to account for every missing video scenario. Third, a naive ensemble that does not share decoder state between the audio and audio-visual model would not be compatible with the streaming scenario. The proposed cascaded model can be used in the streaming scenario, because it shares decoder state via the stochastic routing of inputs.

# E Extended Results

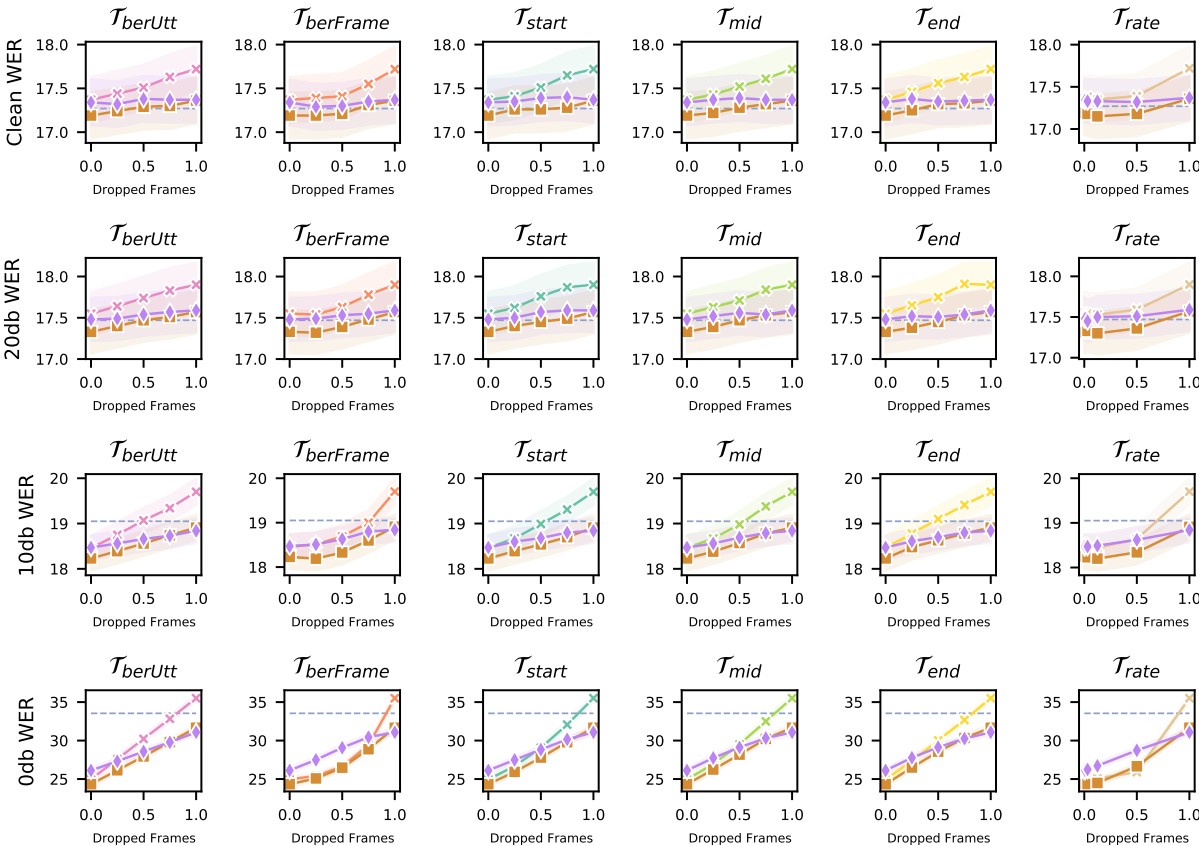

Figure 6: Conformer CAT shown across four acoustic noise conditions and six different test suites. The X-marked, square-marked, diamond-marked, and dashed lines respectively show the WER for the vanilla model, Dropout Utt, Cascade Utt, and the AO baseline. The shaded area around each line denotes the model's 95% confidence interval.

**Advantage of Vanilla Conformer CAT AV Models Varies Across Acoustic Noise Conditions** In Figure 6, we see that when all the frames are dropped, the vanilla model consistently underperforms its AO baseline. However, the point at which this happens varies depending on how noisy the audio is. For example, for $\mathcal{T}_{berUtt}$, we observe the overwhelming advantage of using the visual modality in the 0db and 10db setting, such that even with half the video frames available, the vanilla AV model would outperform its AO baseline. By contrast, in the clean or 20db setting, the visual modality brings such a small advantage, if any, that even with all the video frames available, the AV and AO models perform similarly within the bounds of the confidence interval.

As in the results from the main paper, we can see that Cascade Utt is consistently robust, but Dropout Utt is to be preferred for the Conformer CAT model since it has better WER performance.

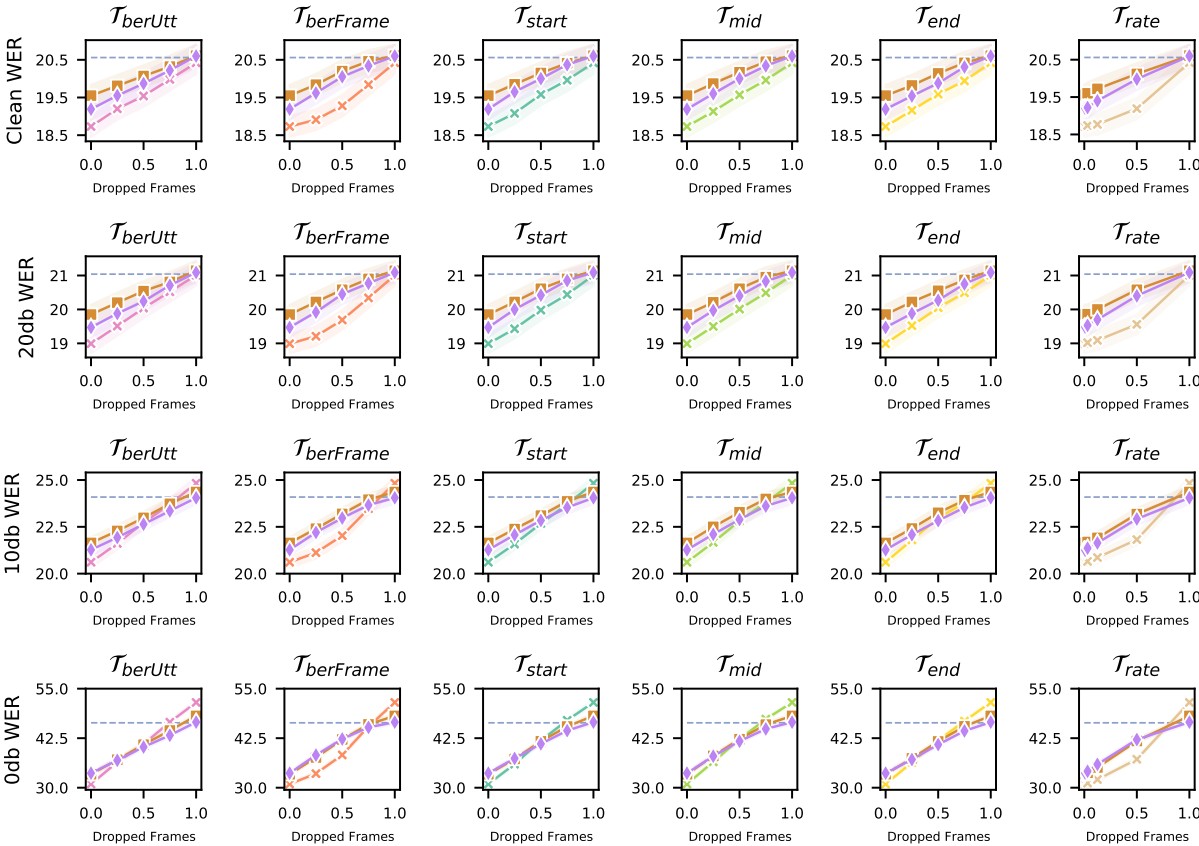

Figure 7: LSTM CAT shown across four acoustic noise conditions and six different test suites. The X-marked, square-marked, diamond-marked, and dashed lines respectively show the WER for the vanilla model, Dropout Utt, Cascade Utt, and the AO baseline. The shaded area around each line denotes the model's 95% confidence interval.

**LSTM CAT AV Models Have a Bigger Advantage over their AO Baseline**  In Figure 7, we see that unlike the Conformer CAT models, the vanilla LSTM AV models outperform their AO baseline at all noise conditions. This suggests that the advantage from the visual modality is larger when the base architecture is weaker. In other words, if the architecture is not as capable at learning good representations from the acoustics alone, it is more likely that the visual modality will bring bigger improvements.

We see that for the LSTM CAT architecture, Cascade Utt is still consistently robust, but now also outperforms Dropout Utt in terms of WER performance.

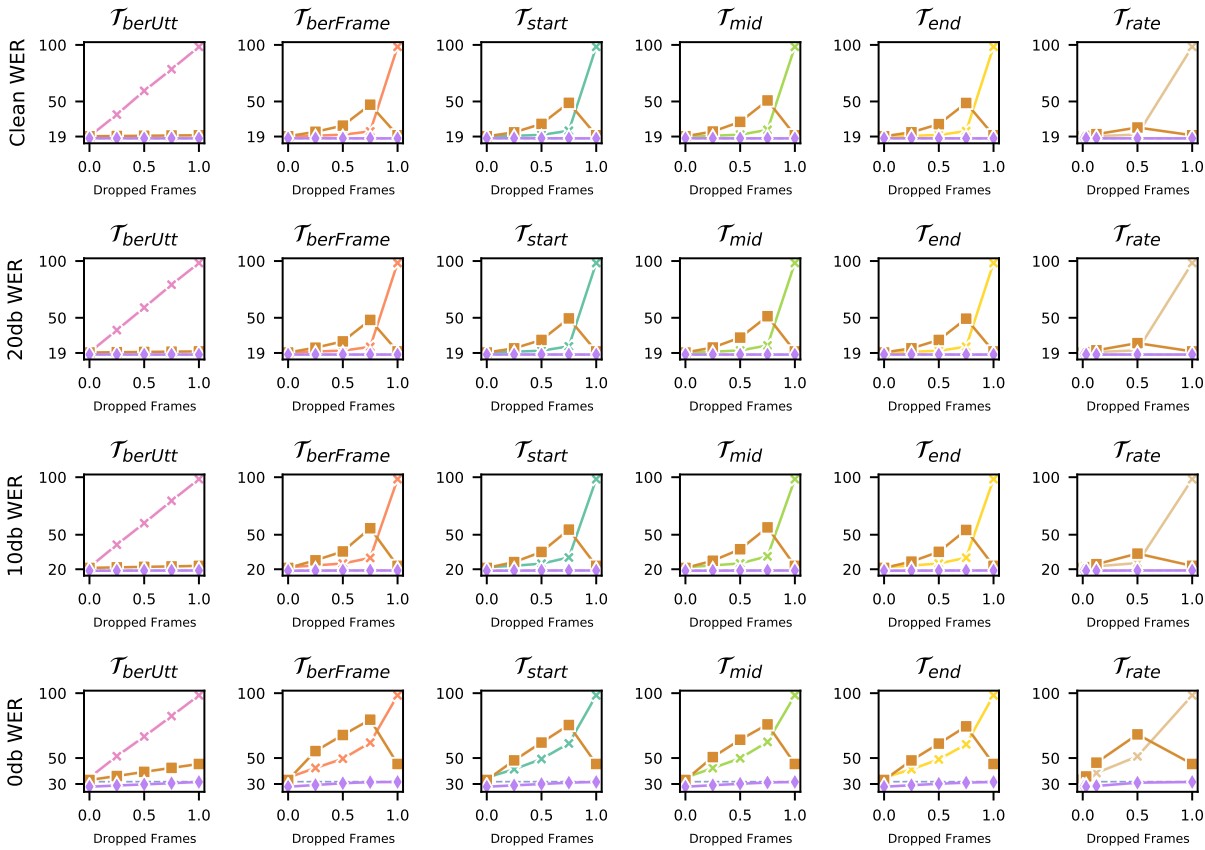

Figure 8: Conformer CM shown across four acoustic noise conditions and six different test suites. The X-marked, square-marked, diamond-marked, and dashed lines respectively show the WER for the vanilla model, Dropout Utt, Cascade Utt, and the AO baseline. The shaded area around each line denotes the model's 95% confidence interval. For $\mathcal{T}_{start}$, $\mathcal{T}_{mid}$, $\mathcal{T}_{end}$, $\mathcal{T}_{berFrame}$, $\mathcal{T}_{rate}$, Dropout Utt does not have its WER monotonically increase as more frames are dropped, indicating that it lacks Test-Time Robustness.

**Dropout Utt is Catastrophic on Conformer CM**    In Figure 8, we notice that Dropout Utt performs catastrophically in response to missing video. Perhaps, this is because the attention mechanism used does not play nicely with the anomalous behavior introduced by blank video frames. The WER degradation also does not scale monotonically with the number of dropped video frames, since we see that when the model actually does better with all the frames dropped compared to 50% or 75% of the frames dropped.

For the Conformer CM architecture, Cascade Utt remains consistently robust across the different test suites and acoustic noise conditions tested.

In the pages that follow, we present comprehensive tables of WER numbers across the various different experimental settings so that the reader can follow up on the details if interested.

Table 4: Robustness to $\mathcal{T}_{berUtt}$ for clean. Columns three to seven represent the amount of dropped video frames.

| ARCHITECTURE | METHOD | 0.0 | 0.25 | 0.5 | 0.75 | 1.0 | ROBUST |
|---|---|---|---|---|---|---|---|
| CONFORMER CAT | AUDIO BASELINE | $17.27 \pm 0.26$ | $17.27 \pm 0.26$ | $17.27 \pm 0.26$ | $17.27 \pm 0.26$ | $17.27 \pm 0.26$ | - |
| CONFORMER CAT | VANILLA | $17.37 \pm 0.26$ | $17.44 \pm 0.26$ | $17.51 \pm 0.26$ | $17.63 \pm 0.26$ | $17.72 \pm 0.26$ | ✗ |
| CONFORMER CAT | VANILLA (25L) | $17.33 \pm 0.26$ | $17.38 \pm 0.26$ | $17.42 \pm 0.26$ | $17.45 \pm 0.26$ | $17.51 \pm 0.26$ | - |
| CONFORMER CAT | CASCADE UTT | $17.34 \pm 0.26$ | $17.32 \pm 0.26$ | $17.38 \pm 0.26$ | $17.37 \pm 0.26$ | $17.37 \pm 0.26$ | ✓ |
| CONFORMER CAT | DROPOUT UTT | $17.19 \pm 0.26$ | $17.24 \pm 0.26$ | $17.29 \pm 0.26$ | $17.3 \pm 0.26$ | $17.36 \pm 0.26$ | ✓ |
| CONFORMER CAT | CASCADE FRAME | $17.18 \pm 0.26$ | $17.19 \pm 0.26$ | $17.22 \pm 0.26$ | $17.22 \pm 0.26$ | $17.25 \pm 0.26$ | ✓ |
| CONFORMER CAT | DROPOUT FRAME | $17.43 \pm 0.27$ | $17.45 \pm 0.27$ | $17.44 \pm 0.27$ | $17.47 \pm 0.27$ | $17.48 \pm 0.27$ | ✓ |
| CONFORMER CAT | AV DROPOUT UTT | $17.35 \pm 0.26$ | $17.41 \pm 0.26$ | $17.42 \pm 0.26$ | $17.5 \pm 0.26$ | $17.5 \pm 0.26$ | ✓ |
| CONFORMER CAT | TWO-PASS | $17.28 \pm 0.26$ | $17.27 \pm 0.26$ | $17.29 \pm 0.26$ | $17.28 \pm 0.26$ | $17.27 \pm 0.26$ | ✓ |
| CON-LSTM CAT | CASCADE UTT | $17.25 \pm 0.26$ | $17.28 \pm 0.26$ | $17.28 \pm 0.26$ | $17.26 \pm 0.26$ | $17.3 \pm 0.26$ | ✓ |
| LSTM CAT | AUDIO BASELINE | $20.56 \pm 0.29$ | $20.56 \pm 0.29$ | $20.56 \pm 0.29$ | $20.56 \pm 0.29$ | $20.56 \pm 0.29$ | - |
| LSTM CAT | VANILLA | $18.73 \pm 0.27$ | $19.2 \pm 0.28$ | $19.54 \pm 0.28$ | $19.98 \pm 0.29$ | $20.42 \pm 0.29$ | ✓ |
| LSTM CAT | CASCADE UTT | $19.19 \pm 0.27$ | $19.55 \pm 0.28$ | $19.87 \pm 0.28$ | $20.22 \pm 0.28$ | $20.6 \pm 0.29$ | ✓ |
| LSTM CAT | DROPOUT UTT | $19.55 \pm 0.28$ | $19.81 \pm 0.28$ | $20.07 \pm 0.28$ | $20.32 \pm 0.29$ | $20.61 \pm 0.29$ | ✓ |
| LSTM-CON CAT | CASCADE UTT | $18.22 \pm 0.27$ | $18.84 \pm 0.27$ | $19.48 \pm 0.28$ | $20.08 \pm 0.28$ | $20.64 \pm 0.29$ | ✓ |
| CONFORMER CM | AUDIO BASELINE | $17.34 \pm 0.26$ | $17.34 \pm 0.26$ | $17.34 \pm 0.26$ | $17.34 \pm 0.26$ | $17.34 \pm 0.26$ | - |
| CONFORMER CM | VANILLA | $19.15 \pm 0.28$ | $38.44 \pm 0.74$ | $59.35 \pm 0.86$ | $78.51 \pm 0.74$ | $98.33 \pm 0.05$ | ✗ |
| CONFORMER CM | CASCADE UTT | $17.45 \pm 0.26$ | $17.44 \pm 0.26$ | $17.41 \pm 0.26$ | $17.42 \pm 0.26$ | $17.37 \pm 0.26$ | ✓ |
| CONFORMER CM | DROPOUT UTT | $19.25 \pm 0.28$ | $19.47 \pm 0.28$ | $19.67 \pm 0.29$ | $19.94 \pm 0.29$ | $20.14 \pm 0.29$ | ✗ |

Table 5: Robustness to $\mathcal{T}_{berFrame}$ for clean. Columns three to seven represent the amount of dropped video frames.

| ARCHITECTURE | METHOD | 0.0 | 0.25 | 0.5 | 0.75 | 1.0 | ROBUST |
|---|---|---|---|---|---|---|---|
| CONFORMER CAT | AUDIO BASELINE | $17.27 \pm 0.26$ | $17.27 \pm 0.26$ | $17.27 \pm 0.26$ | $17.27 \pm 0.26$ | $17.27 \pm 0.26$ | - |
| CONFORMER CAT | VANILLA | $17.37 \pm 0.26$ | $17.39 \pm 0.26$ | $17.41 \pm 0.26$ | $17.55 \pm 0.26$ | $17.72 \pm 0.26$ | ✗ |
| CONFORMER CAT | VANILLA (25L) | $17.33 \pm 0.26$ | $17.29 \pm 0.26$ | $17.34 \pm 0.26$ | $17.41 \pm 0.26$ | $17.51 \pm 0.26$ | - |
| CONFORMER CAT | CASCADE UTT | $17.34 \pm 0.26$ | $17.29 \pm 0.26$ | $17.3 \pm 0.26$ | $17.35 \pm 0.26$ | $17.37 \pm 0.26$ | ✓ |
| CONFORMER CAT | DROPOUT UTT | $17.19 \pm 0.26$ | $17.19 \pm 0.26$ | $17.21 \pm 0.26$ | $17.31 \pm 0.26$ | $17.36 \pm 0.26$ | ✓ |
| CONFORMER CAT | CASCADE FRAME | $17.18 \pm 0.26$ | $17.15 \pm 0.26$ | $17.21 \pm 0.26$ | $17.22 \pm 0.26$ | $17.25 \pm 0.26$ | ✓ |
| CONFORMER CAT | DROPOUT FRAME | $17.43 \pm 0.27$ | $17.39 \pm 0.26$ | $17.39 \pm 0.27$ | $17.43 \pm 0.27$ | $17.48 \pm 0.27$ | ✓ |
| CONFORMER CAT | AV DROPOUT UTT | $17.35 \pm 0.26$ | $17.37 \pm 0.26$ | $17.42 \pm 0.26$ | $17.49 \pm 0.26$ | $17.5 \pm 0.26$ | ✓ |
| CONFORMER CAT | TWO-PASS | $17.28 \pm 0.26$ | $17.22 \pm 0.26$ | $17.25 \pm 0.26$ | $17.26 \pm 0.26$ | $17.27 \pm 0.26$ | ✓ |
| CON-LSTM CAT | CASCADE UTT | $17.25 \pm 0.26$ | $17.26 \pm 0.26$ | $17.29 \pm 0.26$ | $17.29 \pm 0.26$ | $17.3 \pm 0.26$ | ✓ |
| LSTM CAT | AUDIO BASELINE | $20.56 \pm 0.29$ | $20.56 \pm 0.29$ | $20.56 \pm 0.29$ | $20.56 \pm 0.29$ | $20.56 \pm 0.29$ | - |
| LSTM CAT | VANILLA | $18.73 \pm 0.27$ | $18.91 \pm 0.27$ | $19.28 \pm 0.28$ | $19.84 \pm 0.28$ | $20.42 \pm 0.29$ | ✓ |
| LSTM CAT | CASCADE UTT | $19.19 \pm 0.27$ | $19.62 \pm 0.28$ | $20.05 \pm 0.28$ | $20.34 \pm 0.29$ | $20.6 \pm 0.29$ | ✓ |
| LSTM CAT | DROPOUT UTT | $19.55 \pm 0.28$ | $19.84 \pm 0.28$ | $20.2 \pm 0.29$ | $20.46 \pm 0.29$ | $20.61 \pm 0.29$ | ✓ |
| LSTM-CON CAT | CASCADE UTT | $18.22 \pm 0.27$ | $18.58 \pm 0.27$ | $19.22 \pm 0.27$ | $19.9 \pm 0.28$ | $20.64 \pm 0.29$ | ✓ |
| CONFORMER CM | AUDIO BASELINE | $17.34 \pm 0.26$ | $17.34 \pm 0.26$ | $17.34 \pm 0.26$ | $17.34 \pm 0.26$ | $17.34 \pm 0.26$ | - |
| CONFORMER CM | VANILLA | $19.15 \pm 0.28$ | $19.77 \pm 0.28$ | $20.65 \pm 0.3$ | $23.56 \pm 0.33$ | $98.33 \pm 0.05$ | ✗ |
| CONFORMER CM | CASCADE UTT | $17.45 \pm 0.26$ | $17.44 \pm 0.26$ | $17.4 \pm 0.26$ | $17.39 \pm 0.26$ | $17.37 \pm 0.26$ | ✓ |
| CONFORMER CM | DROPOUT UTT | $19.25 \pm 0.28$ | $23.2 \pm 0.33$ | $28.66 \pm 0.39$ | $47.17 \pm 0.52$ | $20.14 \pm 0.29$ | ✗ |

Table 6: Robustness to $\mathcal{T}_{start}$ for clean. Columns three to seven represent the amount of dropped video frames.

| Architecture | Method | 0.0 | 0.25 | 0.5 | 0.75 | 1.0 | Robust |
|---|---|---|---|---|---|---|---|
| Conformer CAT | Audio Baseline | 17.27 ± 0.26 | 17.27 ± 0.26 | 17.27 ± 0.26 | 17.27 ± 0.26 | 17.27 ± 0.26 | - |
| Conformer CAT | Vanilla | 17.37 ± 0.26 | 17.41 ± 0.26 | 17.51 ± 0.26 | 17.65 ± 0.26 | 17.72 ± 0.26 | ✗ |
| Conformer CAT | Vanilla (25L) | 17.33 ± 0.26 | 17.32 ± 0.26 | 17.44 ± 0.26 | 17.43 ± 0.26 | 17.51 ± 0.26 | - |
| Conformer CAT | Cascade Utt | 17.34 ± 0.26 | 17.35 ± 0.26 | 17.39 ± 0.26 | 17.4 ± 0.26 | 17.37 ± 0.26 | ✓ |
| Conformer CAT | Dropout Utt | 17.19 ± 0.26 | 17.26 ± 0.26 | 17.26 ± 0.26 | 17.28 ± 0.26 | 17.36 ± 0.26 | ✓ |
| Conformer CAT | Cascade Frame | 17.18 ± 0.26 | 17.16 ± 0.26 | 17.2 ± 0.26 | 17.33 ± 0.26 | 17.25 ± 0.26 | ✓ |
| Conformer CAT | Dropout Frame | 17.43 ± 0.27 | 17.41 ± 0.27 | 17.43 ± 0.27 | 17.46 ± 0.27 | 17.48 ± 0.27 | ✓ |
| Conformer CAT | AV Dropout Utt | 17.35 ± 0.26 | 17.43 ± 0.26 | 17.46 ± 0.26 | 17.48 ± 0.26 | 17.5 ± 0.26 | ✓ |
| Conformer CAT | Two-Pass | 17.28 ± 0.26 | 17.32 ± 0.26 | 17.31 ± 0.26 | 17.3 ± 0.26 | 17.27 ± 0.26 | ✓ |
| Con-LSTM CAT | Cascade Utt | 17.25 ± 0.26 | 17.25 ± 0.26 | 17.25 ± 0.26 | 17.27 ± 0.26 | 17.3 ± 0.26 | ✓ |
| LSTM CAT | Audio Baseline | 20.56 ± 0.29 | 20.56 ± 0.29 | 20.56 ± 0.29 | 20.56 ± 0.29 | 20.56 ± 0.29 | - |
| LSTM CAT | Vanilla | 18.73 ± 0.27 | 19.08 ± 0.28 | 19.58 ± 0.28 | 19.96 ± 0.29 | 20.42 ± 0.29 | ✓ |
| LSTM CAT | Cascade Utt | 19.19 ± 0.27 | 19.65 ± 0.28 | 20.0 ± 0.28 | 20.37 ± 0.28 | 20.6 ± 0.29 | ✓ |
| LSTM CAT | Dropout Utt | 19.55 ± 0.28 | 19.85 ± 0.28 | 20.15 ± 0.28 | 20.41 ± 0.29 | 20.61 ± 0.29 | ✓ |
| LSTM-Con CAT | Cascade Utt | 18.22 ± 0.27 | 18.93 ± 0.27 | 19.58 ± 0.28 | 20.22 ± 0.29 | 20.64 ± 0.29 | ✓ |
| Conformer CM | Audio Baseline | 17.34 ± 0.26 | 17.34 ± 0.26 | 17.34 ± 0.26 | 17.34 ± 0.26 | 17.34 ± 0.26 | - |
| Conformer CM | Vanilla | 19.15 ± 0.28 | 19.64 ± 0.28 | 20.5 ± 0.29 | 24.1 ± 0.34 | 98.33 ± 0.05 | ✗ |
| Conformer CM | Cascade Utt | 17.45 ± 0.26 | 17.47 ± 0.26 | 17.45 ± 0.26 | 17.4 ± 0.26 | 17.37 ± 0.26 | ✓ |
| Conformer CM | Dropout Utt | 19.25 ± 0.28 | 22.7 ± 0.32 | 30.19 ± 0.41 | 48.8 ± 0.55 | 20.14 ± 0.29 | ✗ |

Table 7: Robustness to $\mathcal{T}_{mid}$ for clean. Columns three to seven represent the amount of dropped video frames.

| Architecture | Method | 0.0 | 0.25 | 0.5 | 0.75 | 1.0 | Robust |
|---|---|---|---|---|---|---|---|
| Conformer CAT | Audio Baseline | 17.27 ± 0.26 | 17.27 ± 0.26 | 17.27 ± 0.26 | 17.27 ± 0.26 | 17.27 ± 0.26 | - |
| Conformer CAT | Vanilla | 17.37 ± 0.26 | 17.43 ± 0.26 | 17.52 ± 0.26 | 17.61 ± 0.26 | 17.72 ± 0.26 | ✗ |
| Conformer CAT | Vanilla (25L) | 17.33 ± 0.26 | 17.35 ± 0.26 | 17.45 ± 0.26 | 17.48 ± 0.26 | 17.51 ± 0.26 | - |
| Conformer CAT | Cascade Utt | 17.34 ± 0.26 | 17.37 ± 0.26 | 17.39 ± 0.26 | 17.37 ± 0.26 | 17.37 ± 0.26 | ✓ |
| Conformer CAT | Dropout Utt | 17.19 ± 0.26 | 17.22 ± 0.26 | 17.28 ± 0.26 | 17.32 ± 0.26 | 17.36 ± 0.26 | ✓ |
| Conformer CAT | Cascade Frame | 17.18 ± 0.26 | 17.2 ± 0.26 | 17.24 ± 0.26 | 17.33 ± 0.26 | 17.25 ± 0.26 | ✓ |
| Conformer CAT | Dropout Frame | 17.43 ± 0.27 | 17.43 ± 0.27 | 17.45 ± 0.27 | 17.48 ± 0.27 | 17.48 ± 0.27 | ✓ |
| Conformer CAT | AV Dropout Utt | 17.35 ± 0.26 | 17.42 ± 0.26 | 17.52 ± 0.26 | 17.51 ± 0.26 | 17.5 ± 0.26 | ✓ |
| Conformer CAT | Two-Pass | 17.28 ± 0.26 | 17.24 ± 0.26 | 17.28 ± 0.26 | 17.27 ± 0.26 | 17.27 ± 0.26 | ✓ |
| Con-LSTM CAT | Cascade Utt | 17.25 ± 0.26 | 17.25 ± 0.26 | 17.27 ± 0.26 | 17.31 ± 0.26 | 17.3 ± 0.26 | ✓ |
| LSTM CAT | Audio Baseline | 20.56 ± 0.29 | 20.56 ± 0.29 | 20.56 ± 0.29 | 20.56 ± 0.29 | 20.56 ± 0.29 | - |
| LSTM CAT | Vanilla | 18.73 ± 0.27 | 19.13 ± 0.28 | 19.57 ± 0.28 | 19.96 ± 0.28 | 20.42 ± 0.29 | ✓ |
| LSTM CAT | Cascade Utt | 19.19 ± 0.27 | 19.6 ± 0.28 | 20.0 ± 0.28 | 20.34 ± 0.28 | 20.6 ± 0.29 | ✓ |
| LSTM CAT | Dropout Utt | 19.55 ± 0.28 | 19.87 ± 0.28 | 20.17 ± 0.28 | 20.45 ± 0.29 | 20.61 ± 0.29 | ✓ |
| LSTM-Con CAT | Cascade Utt | 18.22 ± 0.27 | 18.76 ± 0.27 | 19.45 ± 0.28 | 20.11 ± 0.28 | 20.64 ± 0.29 | ✓ |
| Conformer CM | Audio Baseline | 17.34 ± 0.26 | 17.34 ± 0.26 | 17.34 ± 0.26 | 17.34 ± 0.26 | 17.34 ± 0.26 | - |
| Conformer CM | Vanilla | 19.15 ± 0.28 | 19.7 ± 0.29 | 20.65 ± 0.3 | 24.9 ± 0.35 | 98.33 ± 0.05 | ✗ |
| Conformer CM | Cascade Utt | 17.45 ± 0.26 | 17.42 ± 0.26 | 17.4 ± 0.26 | 17.38 ± 0.26 | 17.37 ± 0.26 | ✓ |
| Conformer CM | Dropout Utt | 19.25 ± 0.28 | 23.45 ± 0.33 | 32.0 ± 0.41 | 50.91 ± 0.53 | 20.14 ± 0.29 | ✗ |

Table 8: Robustness to $\mathcal{T}_{end}$ for clean. Columns three to seven represent the amount of dropped video frames.

| Architecture | Method | 0.0 | 0.25 | 0.5 | 0.75 | 1.0 | Robust |
|---|---|---|---|---|---|---|---|
| Conformer CAT | Audio Baseline | 17.27 ± 0.26 | 17.27 ± 0.26 | 17.27 ± 0.26 | 17.27 ± 0.26 | 17.27 ± 0.26 | - |
| Conformer CAT | Vanilla | 17.37 ± 0.26 | 17.46 ± 0.26 | 17.56 ± 0.26 | 17.63 ± 0.26 | 17.72 ± 0.26 | ✗ |
| Conformer CAT | Vanilla (25L) | 17.33 ± 0.26 | 17.34 ± 0.26 | 17.36 ± 0.26 | 17.46 ± 0.26 | 17.51 ± 0.26 | - |
| Conformer CAT | Cascade Utt | 17.34 ± 0.26 | 17.38 ± 0.26 | 17.35 ± 0.26 | 17.36 ± 0.26 | 17.37 ± 0.26 | ✓ |
| Conformer CAT | Dropout Utt | 17.19 ± 0.26 | 17.25 ± 0.26 | 17.32 ± 0.26 | 17.33 ± 0.26 | 17.36 ± 0.26 | ✓ |
| Conformer CAT | Cascade Frame | 17.18 ± 0.26 | 17.24 ± 0.26 | 17.29 ± 0.26 | 17.35 ± 0.26 | 17.25 ± 0.26 | ✓ |
| Conformer CAT | Dropout Frame | 17.43 ± 0.27 | 17.4 ± 0.27 | 17.45 ± 0.27 | 17.49 ± 0.27 | 17.48 ± 0.27 | ✓ |
| Conformer CAT | AV Dropout Utt | 17.35 ± 0.26 | 17.42 ± 0.26 | 17.46 ± 0.26 | 17.53 ± 0.26 | 17.5 ± 0.26 | ✓ |
| Conformer CAT | Two-Pass | 17.28 ± 0.26 | 17.24 ± 0.26 | 17.25 ± 0.26 | 17.25 ± 0.26 | 17.27 ± 0.26 | ✓ |
| Con-LSTM CAT | Cascade Utt | 17.25 ± 0.26 | 17.29 ± 0.26 | 17.29 ± 0.26 | 17.31 ± 0.26 | 17.3 ± 0.26 | ✓ |
| LSTM CAT | Audio Baseline | 20.56 ± 0.29 | 20.56 ± 0.29 | 20.56 ± 0.29 | 20.56 ± 0.29 | 20.56 ± 0.29 | - |
| LSTM CAT | Vanilla | 18.73 ± 0.27 | 19.16 ± 0.28 | 19.59 ± 0.28 | 19.94 ± 0.28 | 20.42 ± 0.29 | ✓ |
| LSTM CAT | Cascade Utt | 19.19 ± 0.27 | 19.54 ± 0.28 | 19.88 ± 0.28 | 20.31 ± 0.28 | 20.6 ± 0.29 | ✓ |
| LSTM CAT | Dropout Utt | 19.55 ± 0.28 | 19.82 ± 0.28 | 20.14 ± 0.28 | 20.42 ± 0.29 | 20.61 ± 0.29 | ✓ |
| LSTM-Con CAT | Cascade Utt | 18.22 ± 0.27 | 18.66 ± 0.27 | 19.3 ± 0.27 | 19.96 ± 0.28 | 20.64 ± 0.29 | ✓ |
| Conformer CM | Audio Baseline | 17.34 ± 0.26 | 17.34 ± 0.26 | 17.34 ± 0.26 | 17.34 ± 0.26 | 17.34 ± 0.26 | - |
| Conformer CM | Vanilla | 19.15 ± 0.28 | 19.64 ± 0.28 | 20.38 ± 0.29 | 23.79 ± 0.34 | 98.33 ± 0.05 | ✗ |
| Conformer CM | Cascade Utt | 17.45 ± 0.26 | 17.44 ± 0.26 | 17.4 ± 0.26 | 17.36 ± 0.26 | 17.37 ± 0.26 | ✓ |
| Conformer CM | Dropout Utt | 19.25 ± 0.28 | 22.75 ± 0.32 | 29.96 ± 0.39 | 48.67 ± 0.51 | 20.14 ± 0.29 | ✗ |

Table 9: Robustness to $\mathcal{T}_{rate}$ for clean. Columns three to eight represent the amount of dropped video frames.

| Architecture | Method | 0 | $\frac{1}{128}$ | $\frac{1}{32}$ | $\frac{1}{8}$ | $\frac{1}{2}$ | 1 | Robust |
|---|---|---|---|---|---|---|---|---|
| Conformer CAT | Audio Baseline | $17.27 \pm 0.26$ | $17.27 \pm 0.26$ | $17.27 \pm 0.26$ | $17.27 \pm 0.26$ | $17.27 \pm 0.26$ | $17.27 \pm 0.26$ | - |
| Conformer CAT | Vanilla | $17.37 \pm 0.26$ | $17.37 \pm 0.26$ | $17.38 \pm 0.26$ | $17.35 \pm 0.26$ | $17.39 \pm 0.26$ | $17.72 \pm 0.26$ | ✗ |
| Conformer CAT | Vanilla (25L) | $17.33 \pm 0.26$ | $17.32 \pm 0.26$ | $17.3 \pm 0.26$ | $17.29 \pm 0.26$ | $17.35 \pm 0.26$ | $17.51 \pm 0.26$ | - |
| Conformer CAT | Cascade Utt | $17.34 \pm 0.26$ | $17.34 \pm 0.26$ | $17.33 \pm 0.26$ | $17.33 \pm 0.26$ | $17.32 \pm 0.26$ | $17.37 \pm 0.26$ | ✓ |
| Conformer CAT | Dropout Utt | $17.19 \pm 0.26$ | $17.19 \pm 0.26$ | $17.18 \pm 0.26$ | $17.15 \pm 0.26$ | $17.18 \pm 0.26$ | $17.36 \pm 0.26$ | ✓ |
| Conformer CAT | Cascade Frame | $17.18 \pm 0.26$ | $17.17 \pm 0.26$ | $17.2 \pm 0.26$ | $17.19 \pm 0.26$ | $17.23 \pm 0.26$ | $17.25 \pm 0.26$ | ✓ |
| Conformer CAT | Dropout Frame | $17.43 \pm 0.27$ | $17.43 \pm 0.27$ | $17.44 \pm 0.27$ | $17.37 \pm 0.26$ | $17.42 \pm 0.26$ | $17.48 \pm 0.27$ | ✓ |
| Conformer CAT | AV Dropout Utt | $17.35 \pm 0.26$ | $17.36 \pm 0.26$ | $17.34 \pm 0.26$ | $17.33 \pm 0.26$ | $17.43 \pm 0.26$ | $17.5 \pm 0.26$ | ✓ |
| Conformer CAT | Two-Pass | $17.28 \pm 0.26$ | $17.29 \pm 0.26$ | $17.27 \pm 0.26$ | $17.25 \pm 0.26$ | $17.23 \pm 0.26$ | $17.27 \pm 0.26$ | ✓ |
| Con-LSTM CAT | Cascade Utt | $17.25 \pm 0.26$ | $17.25 \pm 0.26$ | $17.26 \pm 0.26$ | $17.29 \pm 0.26$ | $17.31 \pm 0.26$ | $17.3 \pm 0.26$ | ✓ |
| LSTM CAT | Audio Baseline | $20.56 \pm 0.29$ | $20.56 \pm 0.29$ | $20.56 \pm 0.29$ | $20.56 \pm 0.29$ | $20.56 \pm 0.29$ | $20.56 \pm 0.29$ | - |
| LSTM CAT | Vanilla | $18.73 \pm 0.27$ | $18.71 \pm 0.27$ | $18.74 \pm 0.27$ | $18.77 \pm 0.27$ | $19.19 \pm 0.28$ | $20.42 \pm 0.29$ | ✓ |
| LSTM CAT | Cascade Utt | $19.19 \pm 0.27$ | $19.2 \pm 0.27$ | $19.22 \pm 0.27$ | $19.4 \pm 0.27$ | $19.98 \pm 0.28$ | $20.6 \pm 0.29$ | ✓ |
| LSTM CAT | Dropout Utt | $19.55 \pm 0.28$ | $19.56 \pm 0.28$ | $19.6 \pm 0.28$ | $19.72 \pm 0.28$ | $20.12 \pm 0.28$ | $20.61 \pm 0.29$ | ✓ |
| LSTM-Con CAT | Cascade Utt | $18.22 \pm 0.27$ | $18.22 \pm 0.27$ | $18.24 \pm 0.27$ | $18.29 \pm 0.27$ | $19.03 \pm 0.27$ | $20.64 \pm 0.29$ | ✓ |
| Conformer CM | Audio Baseline | $17.34 \pm 0.26$ | $17.34 \pm 0.26$ | $17.34 \pm 0.26$ | $17.34 \pm 0.26$ | $17.34 \pm 0.26$ | $17.34 \pm 0.26$ | - |
| Conformer CM | Vanilla | $19.15 \pm 0.28$ | $19.13 \pm 0.28$ | $19.2 \pm 0.28$ | $19.43 \pm 0.28$ | $20.72 \pm 0.3$ | $98.33 \pm 0.05$ | ✗ |
| Conformer CM | Cascade Utt | $17.45 \pm 0.26$ | $17.45 \pm 0.26$ | $17.45 \pm 0.26$ | $17.42 \pm 0.26$ | $17.42 \pm 0.26$ | $17.37 \pm 0.26$ | ✓ |
| Conformer CM | Dropout Utt | $19.25 \pm 0.28$ | $19.27 \pm 0.28$ | $19.55 \pm 0.28$ | $21.1 \pm 0.3$ | $27.0 \pm 0.37$ | $20.14 \pm 0.29$ | ✗ |

Table 10: Robustness to $\mathcal{T}_{berUtt}$ for 20db. Columns three to seven represent the amount of dropped video frames.

| Architecture | Method | 0.0 | 0.25 | 0.5 | 0.75 | 1.0 | Robust |
|---|---|---|---|---|---|---|---|
| Conformer CAT | Audio Baseline | $17.47 \pm 0.26$ | $17.47 \pm 0.26$ | $17.47 \pm 0.26$ | $17.47 \pm 0.26$ | $17.47 \pm 0.26$ | - |
| Conformer CAT | Vanilla | $17.55 \pm 0.26$ | $17.64 \pm 0.26$ | $17.74 \pm 0.26$ | $17.83 \pm 0.27$ | $17.9 \pm 0.27$ | ✗ |
| Conformer CAT | Vanilla (25L) | $17.48 \pm 0.26$ | $17.54 \pm 0.26$ | $17.63 \pm 0.26$ | $17.72 \pm 0.26$ | $17.79 \pm 0.26$ | - |
| Conformer CAT | Cascade Utt | $17.48 \pm 0.26$ | $17.49 \pm 0.26$ | $17.54 \pm 0.26$ | $17.57 \pm 0.26$ | $17.59 \pm 0.26$ | ✓ |
| Conformer CAT | Dropout Utt | $17.33 \pm 0.26$ | $17.4 \pm 0.26$ | $17.47 \pm 0.26$ | $17.51 \pm 0.26$ | $17.57 \pm 0.26$ | ✓ |
| Conformer CAT | Cascade Frame | $17.35 \pm 0.26$ | $17.4 \pm 0.26$ | $17.38 \pm 0.26$ | $17.45 \pm 0.26$ | $17.45 \pm 0.26$ | ✓ |
| Conformer CAT | Dropout Frame | $17.62 \pm 0.27$ | $17.63 \pm 0.27$ | $17.65 \pm 0.27$ | $17.67 \pm 0.27$ | $17.68 \pm 0.27$ | ✓ |
| Conformer CAT | AV Dropout Utt | $17.52 \pm 0.26$ | $17.57 \pm 0.26$ | $17.68 \pm 0.26$ | $17.71 \pm 0.26$ | $17.78 \pm 0.26$ | ✗ |
| Conformer CAT | Two-Pass | $17.5 \pm 0.27$ | $17.51 \pm 0.26$ | $17.51 \pm 0.27$ | $17.48 \pm 0.26$ | $17.47 \pm 0.26$ | ✓ |
| Con-LSTM CAT | Cascade Utt | $17.48 \pm 0.26$ | $17.51 \pm 0.26$ | $17.51 \pm 0.26$ | $17.53 \pm 0.26$ | $17.54 \pm 0.26$ | ✓ |
| LSTM CAT | Audio Baseline | $21.04 \pm 0.29$ | $21.04 \pm 0.29$ | $21.04 \pm 0.29$ | $21.04 \pm 0.29$ | $21.04 \pm 0.29$ | - |
| LSTM CAT | Vanilla | $18.99 \pm 0.27$ | $19.51 \pm 0.28$ | $20.04 \pm 0.29$ | $20.52 \pm 0.29$ | $21.01 \pm 0.3$ | ✓ |
| LSTM CAT | Cascade Utt | $19.47 \pm 0.28$ | $19.88 \pm 0.28$ | $20.24 \pm 0.28$ | $20.71 \pm 0.29$ | $21.09 \pm 0.29$ | ✓ |
| LSTM CAT | Dropout Utt | $19.85 \pm 0.28$ | $20.2 \pm 0.29$ | $20.54 \pm 0.29$ | $20.81 \pm 0.29$ | $21.14 \pm 0.29$ | ✓ |
| LSTM-Con CAT | Cascade Utt | $18.48 \pm 0.27$ | $19.13 \pm 0.28$ | $19.81 \pm 0.28$ | $20.48 \pm 0.29$ | $21.11 \pm 0.29$ | ✓ |
| Conformer CM | Audio Baseline | $17.6 \pm 0.26$ | $17.6 \pm 0.26$ | $17.6 \pm 0.26$ | $17.6 \pm 0.26$ | $17.6 \pm 0.26$ | - |
| Conformer CM | Vanilla | $19.48 \pm 0.28$ | $39.15 \pm 0.77$ | $58.97 \pm 0.86$ | $79.16 \pm 0.73$ | $98.3 \pm 0.05$ | ✗ |
| Conformer CM | Cascade Utt | $17.67 \pm 0.26$ | $17.64 \pm 0.26$ | $17.63 \pm 0.26$ | $17.63 \pm 0.26$ | $17.59 \pm 0.26$ | ✓ |
| Conformer CM | Dropout Utt | $19.43 \pm 0.28$ | $19.65 \pm 0.28$ | $19.9 \pm 0.29$ | $20.13 \pm 0.29$ | $20.39 \pm 0.3$ | ✗ |

Table 11: Robustness to $\mathcal{T}_{berFrame}$ for 20db. Columns three to seven represent the amount of dropped video frames.

| Architecture | Method | 0.0 | 0.25 | 0.5 | 0.75 | 1.0 | Robust |
|---|---|---|---|---|---|---|---|
| Conformer CAT | Audio Baseline | $17.47 \pm 0.26$ | $17.47 \pm 0.26$ | $17.47 \pm 0.26$ | $17.47 \pm 0.26$ | $17.47 \pm 0.26$ | - |
| Conformer CAT | Vanilla | $17.55 \pm 0.26$ | $17.54 \pm 0.26$ | $17.63 \pm 0.26$ | $17.78 \pm 0.26$ | $17.9 \pm 0.27$ | ✗ |
| Conformer CAT | Vanilla (25L) | $17.48 \pm 0.26$ | $17.5 \pm 0.26$ | $17.56 \pm 0.26$ | $17.64 \pm 0.26$ | $17.79 \pm 0.26$ | - |
| Conformer CAT | Cascade Utt | $17.48 \pm 0.26$ | $17.49 \pm 0.26$ | $17.53 \pm 0.26$ | $17.55 \pm 0.26$ | $17.59 \pm 0.26$ | ✓ |
| Conformer CAT | Dropout Utt | $17.33 \pm 0.26$ | $17.32 \pm 0.26$ | $17.39 \pm 0.26$ | $17.48 \pm 0.26$ | $17.57 \pm 0.26$ | ✓ |
| Conformer CAT | Cascade Frame | $17.35 \pm 0.26$ | $17.33 \pm 0.26$ | $17.36 \pm 0.26$ | $17.38 \pm 0.26$ | $17.45 \pm 0.26$ | ✓ |
| Conformer CAT | Dropout Frame | $17.62 \pm 0.27$ | $17.59 \pm 0.27$ | $17.62 \pm 0.27$ | $17.66 \pm 0.27$ | $17.68 \pm 0.27$ | ✓ |
| Conformer CAT | AV Dropout Utt | $17.52 \pm 0.26$ | $17.53 \pm 0.26$ | $17.58 \pm 0.26$ | $17.72 \pm 0.26$ | $17.78 \pm 0.26$ | ✗ |
| Conformer CAT | Two-Pass | $17.5 \pm 0.27$ | $17.47 \pm 0.26$ | $17.49 \pm 0.26$ | $17.5 \pm 0.26$ | $17.47 \pm 0.26$ | ✓ |
| Con-LSTM CAT | Cascade Utt | $17.48 \pm 0.26$ | $17.53 \pm 0.26$ | $17.54 \pm 0.26$ | $17.56 \pm 0.26$ | $17.54 \pm 0.26$ | ✓ |
| LSTM CAT | Audio Baseline | $21.04 \pm 0.29$ | $21.04 \pm 0.29$ | $21.04 \pm 0.29$ | $21.04 \pm 0.29$ | $21.04 \pm 0.29$ | - |
| LSTM CAT | Vanilla | $18.99 \pm 0.27$ | $19.21 \pm 0.28$ | $19.69 \pm 0.28$ | $20.34 \pm 0.29$ | $21.01 \pm 0.3$ | ✓ |
| LSTM CAT | Cascade Utt | $19.47 \pm 0.28$ | $19.92 \pm 0.28$ | $20.45 \pm 0.29$ | $20.77 \pm 0.29$ | $21.09 \pm 0.29$ | ✓ |
| LSTM CAT | Dropout Utt | $19.85 \pm 0.28$ | $20.22 \pm 0.28$ | $20.58 \pm 0.29$ | $20.9 \pm 0.29$ | $21.14 \pm 0.29$ | ✓ |
| LSTM-Con CAT | Cascade Utt | $18.48 \pm 0.27$ | $18.93 \pm 0.27$ | $19.52 \pm 0.28$ | $20.32 \pm 0.29$ | $21.11 \pm 0.29$ | ✓ |
| Conformer CM | Audio Baseline | $17.6 \pm 0.26$ | $17.6 \pm 0.26$ | $17.6 \pm 0.26$ | $17.6 \pm 0.26$ | $17.6 \pm 0.26$ | - |
| Conformer CM | Vanilla | $19.48 \pm 0.28$ | $20.27 \pm 0.29$ | $21.06 \pm 0.3$ | $24.36 \pm 0.34$ | $98.3 \pm 0.05$ | ✗ |
| Conformer CM | Cascade Utt | $17.67 \pm 0.26$ | $17.62 \pm 0.26$ | $17.62 \pm 0.26$ | $17.6 \pm 0.26$ | $17.59 \pm 0.26$ | ✓ |
| Conformer CM | Dropout Utt | $19.43 \pm 0.28$ | $23.65 \pm 0.33$ | $29.21 \pm 0.39$ | $48.13 \pm 0.53$ | $20.39 \pm 0.3$ | ✗ |

Table 12: Robustness to $\mathcal{T}_{start}$ for 20db. Columns three to seven represent the amount of dropped video frames.

| Architecture | Method | 0.0 | 0.25 | 0.5 | 0.75 | 1.0 | Robust |
|---|---|---|---|---|---|---|---|
| Conformer CAT | Audio Baseline | $17.47 \pm 0.26$ | $17.47 \pm 0.26$ | $17.47 \pm 0.26$ | $17.47 \pm 0.26$ | $17.47 \pm 0.26$ | - |
| Conformer CAT | Vanilla | $17.55 \pm 0.26$ | $17.62 \pm 0.26$ | $17.76 \pm 0.26$ | $17.87 \pm 0.26$ | $17.9 \pm 0.27$ | ✗ |
| Conformer CAT | Vanilla (25L) | $17.48 \pm 0.26$ | $17.5 \pm 0.26$ | $17.64 \pm 0.26$ | $17.69 \pm 0.26$ | $17.79 \pm 0.26$ | - |
| Conformer CAT | Cascade Utt | $17.48 \pm 0.26$ | $17.5 \pm 0.26$ | $17.57 \pm 0.26$ | $17.59 \pm 0.26$ | $17.59 \pm 0.26$ | ✓ |
| Conformer CAT | Dropout Utt | $17.33 \pm 0.26$ | $17.4 \pm 0.26$ | $17.45 \pm 0.26$ | $17.49 \pm 0.26$ | $17.57 \pm 0.26$ | ✓ |
| Conformer CAT | Cascade Frame | $17.35 \pm 0.26$ | $17.3 \pm 0.26$ | $17.37 \pm 0.26$ | $17.52 \pm 0.26$ | $17.45 \pm 0.26$ | ✓ |
| Conformer CAT | Dropout Frame | $17.62 \pm 0.27$ | $17.61 \pm 0.27$ | $17.62 \pm 0.27$ | $17.68 \pm 0.27$ | $17.68 \pm 0.27$ | ✓ |
| Conformer CAT | AV Dropout Utt | $17.52 \pm 0.26$ | $17.58 \pm 0.26$ | $17.67 \pm 0.26$ | $17.7 \pm 0.26$ | $17.78 \pm 0.26$ | ✗ |
| Conformer CAT | Two-Pass | $17.5 \pm 0.27$ | $17.5 \pm 0.26$ | $17.52 \pm 0.26$ | $17.52 \pm 0.26$ | $17.47 \pm 0.26$ | ✓ |
| Con-LSTM CAT | Cascade Utt | $17.48 \pm 0.26$ | $17.48 \pm 0.26$ | $17.51 \pm 0.26$ | $17.52 \pm 0.26$ | $17.54 \pm 0.26$ | ✓ |
| LSTM CAT | Audio Baseline | $21.04 \pm 0.29$ | $21.04 \pm 0.29$ | $21.04 \pm 0.29$ | $21.04 \pm 0.29$ | $21.04 \pm 0.29$ | - |
| LSTM CAT | Vanilla | $18.99 \pm 0.27$ | $19.43 \pm 0.28$ | $19.98 \pm 0.29$ | $20.44 \pm 0.29$ | $21.01 \pm 0.3$ | ✓ |
| LSTM CAT | Cascade Utt | $19.47 \pm 0.28$ | $20.0 \pm 0.28$ | $20.42 \pm 0.29$ | $20.85 \pm 0.29$ | $21.09 \pm 0.29$ | ✓ |
| LSTM CAT | Dropout Utt | $19.85 \pm 0.28$ | $20.22 \pm 0.29$ | $20.61 \pm 0.29$ | $20.9 \pm 0.29$ | $21.14 \pm 0.29$ | ✓ |
| LSTM-Con CAT | Cascade Utt | $18.48 \pm 0.27$ | $19.24 \pm 0.28$ | $19.93 \pm 0.28$ | $20.62 \pm 0.29$ | $21.11 \pm 0.29$ | ✓ |
| Conformer CM | Audio Baseline | $17.6 \pm 0.26$ | $17.6 \pm 0.26$ | $17.6 \pm 0.26$ | $17.6 \pm 0.26$ | $17.6 \pm 0.26$ | - |
| Conformer CM | Vanilla | $19.48 \pm 0.28$ | $20.1 \pm 0.29$ | $21.0 \pm 0.3$ | $24.85 \pm 0.35$ | $98.3 \pm 0.05$ | ✗ |
| Conformer CM | Cascade Utt | $17.67 \pm 0.26$ | $17.68 \pm 0.26$ | $17.64 \pm 0.26$ | $17.63 \pm 0.26$ | $17.59 \pm 0.26$ | ✓ |
| Conformer CM | Dropout Utt | $19.43 \pm 0.28$ | $23.07 \pm 0.32$ | $30.48 \pm 0.41$ | $49.47 \pm 0.55$ | $20.39 \pm 0.3$ | ✗ |

Table 13: Robustness to $\mathcal{T}_{mid}$ for 20db. Columns three to seven represent the amount of dropped video frames.

| Architecture | Method | 0.0 | 0.25 | 0.5 | 0.75 | 1.0 | Robust |
|---|---|---|---|---|---|---|---|
| Conformer CAT | Audio Baseline | $17.47 \pm 0.26$ | $17.47 \pm 0.26$ | $17.47 \pm 0.26$ | $17.47 \pm 0.26$ | $17.47 \pm 0.26$ | - |
| Conformer CAT | Vanilla | $17.55 \pm 0.26$ | $17.63 \pm 0.26$ | $17.71 \pm 0.26$ | $17.84 \pm 0.26$ | $17.9 \pm 0.27$ | ✗ |
| Conformer CAT | Vanilla (25L) | $17.48 \pm 0.26$ | $17.53 \pm 0.26$ | $17.62 \pm 0.26$ | $17.71 \pm 0.26$ | $17.79 \pm 0.26$ | - |
| Conformer CAT | Cascade Utt | $17.48 \pm 0.26$ | $17.52 \pm 0.26$ | $17.56 \pm 0.26$ | $17.54 \pm 0.26$ | $17.59 \pm 0.26$ | ✓ |
| Conformer CAT | Dropout Utt | $17.33 \pm 0.26$ | $17.39 \pm 0.26$ | $17.47 \pm 0.26$ | $17.53 \pm 0.26$ | $17.57 \pm 0.26$ | ✓ |
| Conformer CAT | Cascade Frame | $17.35 \pm 0.26$ | $17.36 \pm 0.26$ | $17.41 \pm 0.26$ | $17.53 \pm 0.26$ | $17.45 \pm 0.26$ | ✓ |
| Conformer CAT | Dropout Frame | $17.62 \pm 0.27$ | $17.62 \pm 0.27$ | $17.63 \pm 0.27$ | $17.69 \pm 0.27$ | $17.68 \pm 0.27$ | ✓ |
| Conformer CAT | AV Dropout Utt | $17.52 \pm 0.26$ | $17.59 \pm 0.26$ | $17.7 \pm 0.26$ | $17.76 \pm 0.26$ | $17.78 \pm 0.26$ | ✗ |
| Conformer CAT | Two-Pass | $17.5 \pm 0.27$ | $17.48 \pm 0.26$ | $17.51 \pm 0.26$ | $17.51 \pm 0.26$ | $17.47 \pm 0.26$ | ✓ |
| Con-LSTM CAT | Cascade Utt | $17.48 \pm 0.26$ | $17.49 \pm 0.26$ | $17.49 \pm 0.26$ | $17.52 \pm 0.26$ | $17.54 \pm 0.26$ | ✓ |
| LSTM CAT | Audio Baseline | $21.04 \pm 0.29$ | $21.04 \pm 0.29$ | $21.04 \pm 0.29$ | $21.04 \pm 0.29$ | $21.04 \pm 0.29$ | - |
| LSTM CAT | Vanilla | $18.99 \pm 0.27$ | $19.5 \pm 0.28$ | $20.01 \pm 0.29$ | $20.49 \pm 0.29$ | $21.01 \pm 0.3$ | ✓ |
| LSTM CAT | Cascade Utt | $19.47 \pm 0.28$ | $19.97 \pm 0.28$ | $20.4 \pm 0.29$ | $20.83 \pm 0.29$ | $21.09 \pm 0.29$ | ✓ |
| LSTM CAT | Dropout Utt | $19.85 \pm 0.28$ | $20.21 \pm 0.28$ | $20.61 \pm 0.29$ | $20.95 \pm 0.29$ | $21.14 \pm 0.29$ | ✓ |
| LSTM-Con CAT | Cascade Utt | $18.48 \pm 0.27$ | $19.11 \pm 0.27$ | $19.83 \pm 0.28$ | $20.53 \pm 0.28$ | $21.11 \pm 0.29$ | ✓ |
| Conformer CM | Audio Baseline | $17.6 \pm 0.26$ | $17.6 \pm 0.26$ | $17.6 \pm 0.26$ | $17.6 \pm 0.26$ | $17.6 \pm 0.26$ | - |
| Conformer CM | Vanilla | $19.48 \pm 0.28$ | $20.11 \pm 0.29$ | $21.22 \pm 0.3$ | $25.62 \pm 0.36$ | $98.3 \pm 0.05$ | ✗ |
| Conformer CM | Cascade Utt | $17.67 \pm 0.26$ | $17.62 \pm 0.26$ | $17.61 \pm 0.26$ | $17.59 \pm 0.26$ | $17.59 \pm 0.26$ | ✓ |
| Conformer CM | Dropout Utt | $19.43 \pm 0.28$ | $23.81 \pm 0.33$ | $32.54 \pm 0.41$ | $51.38 \pm 0.53$ | $20.39 \pm 0.3$ | ✗ |

Table 14: Robustness to $\mathcal{T}_{end}$ for 20db. Columns three to seven represent the amount of dropped video frames.

| Architecture | Method | 0.0 | 0.25 | 0.5 | 0.75 | 1.0 | Robust |
|---|---|---|---|---|---|---|---|
| Conformer CAT | Audio Baseline | $17.47 \pm 0.26$ | $17.47 \pm 0.26$ | $17.47 \pm 0.26$ | $17.47 \pm 0.26$ | $17.47 \pm 0.26$ | - |
| Conformer CAT | Vanilla | $17.55 \pm 0.26$ | $17.65 \pm 0.26$ | $17.75 \pm 0.26$ | $17.91 \pm 0.26$ | $17.9 \pm 0.27$ | ✗ |
| Conformer CAT | Vanilla (25L) | $17.48 \pm 0.26$ | $17.53 \pm 0.26$ | $17.65 \pm 0.26$ | $17.68 \pm 0.26$ | $17.79 \pm 0.26$ | - |
| Conformer CAT | Cascade Utt | $17.48 \pm 0.26$ | $17.52 \pm 0.26$ | $17.51 \pm 0.26$ | $17.54 \pm 0.26$ | $17.59 \pm 0.26$ | ✓ |
| Conformer CAT | Dropout Utt | $17.33 \pm 0.26$ | $17.38 \pm 0.26$ | $17.45 \pm 0.26$ | $17.53 \pm 0.26$ | $17.57 \pm 0.26$ | ✓ |
| Conformer CAT | Cascade Frame | $17.35 \pm 0.26$ | $17.38 \pm 0.26$ | $17.47 \pm 0.26$ | $17.56 \pm 0.26$ | $17.45 \pm 0.26$ | ✓ |
| Conformer CAT | Dropout Frame | $17.62 \pm 0.27$ | $17.6 \pm 0.27$ | $17.68 \pm 0.27$ | $17.72 \pm 0.27$ | $17.68 \pm 0.27$ | ✓ |
| Conformer CAT | AV Dropout Utt | $17.52 \pm 0.26$ | $17.61 \pm 0.26$ | $17.69 \pm 0.26$ | $17.76 \pm 0.26$ | $17.78 \pm 0.26$ | ✗ |
| Conformer CAT | Two-Pass | $17.5 \pm 0.27$ | $17.45 \pm 0.26$ | $17.45 \pm 0.26$ | $17.51 \pm 0.26$ | $17.47 \pm 0.26$ | ✓ |
| Con-LSTM CAT | Cascade Utt | $17.48 \pm 0.26$ | $17.52 \pm 0.26$ | $17.53 \pm 0.26$ | $17.55 \pm 0.26$ | $17.54 \pm 0.26$ | ✓ |
| LSTM CAT | Audio Baseline | $21.04 \pm 0.29$ | $21.04 \pm 0.29$ | $21.04 \pm 0.29$ | $21.04 \pm 0.29$ | $21.04 \pm 0.29$ | - |
| LSTM CAT | Vanilla | $18.99 \pm 0.27$ | $19.52 \pm 0.28$ | $20.05 \pm 0.28$ | $20.49 \pm 0.29$ | $21.01 \pm 0.3$ | ✓ |
| LSTM CAT | Cascade Utt | $19.47 \pm 0.28$ | $19.88 \pm 0.28$ | $20.27 \pm 0.28$ | $20.75 \pm 0.29$ | $21.09 \pm 0.29$ | ✓ |
| LSTM CAT | Dropout Utt | $19.85 \pm 0.28$ | $20.22 \pm 0.28$ | $20.55 \pm 0.29$ | $20.87 \pm 0.29$ | $21.14 \pm 0.29$ | ✓ |
| LSTM-Con CAT | Cascade Utt | $18.48 \pm 0.27$ | $19.02 \pm 0.27$ | $19.66 \pm 0.28$ | $20.44 \pm 0.28$ | $21.11 \pm 0.29$ | ✓ |
| Conformer CM | Audio Baseline | $17.6 \pm 0.26$ | $17.6 \pm 0.26$ | $17.6 \pm 0.26$ | $17.6 \pm 0.26$ | $17.6 \pm 0.26$ | - |
| Conformer CM | Vanilla | $19.48 \pm 0.28$ | $20.05 \pm 0.29$ | $20.93 \pm 0.3$ | $24.49 \pm 0.34$ | $98.3 \pm 0.05$ | ✗ |
| Conformer CM | Cascade Utt | $17.67 \pm 0.26$ | $17.67 \pm 0.26$ | $17.62 \pm 0.26$ | $17.6 \pm 0.26$ | $17.59 \pm 0.26$ | ✓ |
| Conformer CM | Dropout Utt | $19.43 \pm 0.28$ | $23.1 \pm 0.32$ | $30.47 \pm 0.39$ | $49.24 \pm 0.51$ | $20.39 \pm 0.3$ | ✗ |

Table 15: Robustness to $\mathcal{T}_{rate}$ for 20db. Columns three to eight represent the amount of dropped video frames.

| Architecture | Method | 0 | $\frac{1}{128}$ | $\frac{1}{32}$ | $\frac{1}{8}$ | $\frac{1}{2}$ | 1 | Robust |
|---|---|---|---|---|---|---|---|---|
| Conformer CAT | Audio Baseline | $17.47 \pm 0.26$ | $17.47 \pm 0.26$ | $17.47 \pm 0.26$ | $17.47 \pm 0.26$ | $17.47 \pm 0.26$ | $17.47 \pm 0.26$ | - |
| Conformer CAT | Vanilla | $17.55 \pm 0.26$ | $17.55 \pm 0.26$ | $17.54 \pm 0.26$ | $17.53 \pm 0.26$ | $17.59 \pm 0.26$ | $17.9 \pm 0.27$ | ✗ |
| Conformer CAT | Vanilla (25L) | $17.48 \pm 0.26$ | $17.47 \pm 0.26$ | $17.49 \pm 0.26$ | $17.45 \pm 0.26$ | $17.53 \pm 0.26$ | $17.79 \pm 0.26$ | - |
| Conformer CAT | Cascade Utt | $17.48 \pm 0.26$ | $17.48 \pm 0.26$ | $17.45 \pm 0.26$ | $17.5 \pm 0.26$ | $17.51 \pm 0.26$ | $17.59 \pm 0.26$ | ✓ |
| Conformer CAT | Dropout Utt | $17.33 \pm 0.26$ | $17.31 \pm 0.26$ | $17.33 \pm 0.26$ | $17.3 \pm 0.26$ | $17.36 \pm 0.26$ | $17.57 \pm 0.26$ | ✓ |
| Conformer CAT | Cascade Frame | $17.35 \pm 0.26$ | $17.35 \pm 0.26$ | $17.33 \pm 0.26$ | $17.36 \pm 0.26$ | $17.37 \pm 0.26$ | $17.45 \pm 0.26$ | ✓ |
| Conformer CAT | Dropout Frame | $17.62 \pm 0.27$ | $17.62 \pm 0.27$ | $17.6 \pm 0.27$ | $17.57 \pm 0.27$ | $17.62 \pm 0.27$ | $17.68 \pm 0.27$ | ✓ |
| Conformer CAT | AV Dropout Utt | $17.52 \pm 0.26$ | $17.5 \pm 0.26$ | $17.53 \pm 0.26$ | $17.51 \pm 0.26$ | $17.64 \pm 0.26$ | $17.78 \pm 0.26$ | ✗ |
| Conformer CAT | Two-Pass | $17.5 \pm 0.27$ | $17.49 \pm 0.27$ | $17.49 \pm 0.27$ | $17.45 \pm 0.26$ | $17.44 \pm 0.26$ | $17.47 \pm 0.26$ | ✓ |
| Con-LSTM CAT | Cascade Utt | $17.48 \pm 0.26$ | $17.48 \pm 0.26$ | $17.48 \pm 0.26$ | $17.5 \pm 0.26$ | $17.54 \pm 0.26$ | $17.54 \pm 0.26$ | ✓ |
| LSTM CAT | Audio Baseline | $21.04 \pm 0.29$ | $21.04 \pm 0.29$ | $21.04 \pm 0.29$ | $21.04 \pm 0.29$ | $21.04 \pm 0.29$ | $21.04 \pm 0.29$ | - |
| LSTM CAT | Vanilla | $18.99 \pm 0.27$ | $18.99 \pm 0.27$ | $19.02 \pm 0.28$ | $19.09 \pm 0.28$ | $19.56 \pm 0.28$ | $21.01 \pm 0.3$ | ✓ |
| LSTM CAT | Cascade Utt | $19.47 \pm 0.28$ | $19.48 \pm 0.28$ | $19.53 \pm 0.28$ | $19.7 \pm 0.28$ | $20.4 \pm 0.29$ | $21.09 \pm 0.29$ | ✓ |
| LSTM CAT | Dropout Utt | $19.85 \pm 0.28$ | $19.86 \pm 0.28$ | $19.86 \pm 0.28$ | $20.0 \pm 0.28$ | $20.58 \pm 0.29$ | $21.14 \pm 0.29$ | ✓ |
| LSTM-Con CAT | Cascade Utt | $18.48 \pm 0.27$ | $18.46 \pm 0.27$ | $18.49 \pm 0.27$ | $18.61 \pm 0.27$ | $19.4 \pm 0.27$ | $21.11 \pm 0.29$ | ✓ |
| Conformer CM | Audio Baseline | $17.6 \pm 0.26$ | $17.6 \pm 0.26$ | $17.6 \pm 0.26$ | $17.6 \pm 0.26$ | $17.6 \pm 0.26$ | $17.6 \pm 0.26$ | - |
| Conformer CM | Vanilla | $19.48 \pm 0.28$ | $19.48 \pm 0.28$ | $19.57 \pm 0.28$ | $19.85 \pm 0.29$ | $21.27 \pm 0.3$ | $98.3 \pm 0.05$ | ✗ |
| Conformer CM | Cascade Utt | $17.67 \pm 0.26$ | $17.67 \pm 0.26$ | $17.64 \pm 0.26$ | $17.59 \pm 0.26$ | $17.62 \pm 0.26$ | $17.59 \pm 0.26$ | ✓ |
| Conformer CM | Dropout Utt | $19.43 \pm 0.28$ | $19.46 \pm 0.28$ | $19.78 \pm 0.29$ | $21.34 \pm 0.3$ | $27.68 \pm 0.37$ | $20.39 \pm 0.3$ | ✗ |

Table 16: Robustness to $\mathcal{T}_{berUtt}$ for 10db. Columns three to seven represent the amount of dropped video frames.

| Architecture | Method | 0.0 | 0.25 | 0.5 | 0.75 | 1.0 | Robust |
|---|---|---|---|---|---|---|---|
| Conformer CAT | Audio Baseline | $19.05 \pm 0.28$ | $19.05 \pm 0.28$ | $19.05 \pm 0.28$ | $19.05 \pm 0.28$ | $19.05 \pm 0.28$ | - |
| Conformer CAT | Vanilla | $18.46 \pm 0.27$ | $18.75 \pm 0.28$ | $19.07 \pm 0.28$ | $19.34 \pm 0.28$ | $19.7 \pm 0.29$ | ✗ |
| Conformer CAT | Vanilla (25L) | $18.45 \pm 0.27$ | $18.71 \pm 0.28$ | $19.01 \pm 0.28$ | $19.21 \pm 0.28$ | $19.51 \pm 0.28$ | - |
| Conformer CAT | Cascade Utt | $18.47 \pm 0.27$ | $18.56 \pm 0.27$ | $18.66 \pm 0.27$ | $18.73 \pm 0.27$ | $18.84 \pm 0.27$ | ✓ |
| Conformer CAT | Dropout Utt | $18.23 \pm 0.27$ | $18.39 \pm 0.27$ | $18.56 \pm 0.27$ | $18.74 \pm 0.27$ | $18.91 \pm 0.28$ | ✓ |
| Conformer CAT | Cascade Frame | $18.41 \pm 0.27$ | $18.55 \pm 0.27$ | $18.67 \pm 0.27$ | $18.75 \pm 0.27$ | $18.91 \pm 0.28$ | ✓ |
| Conformer CAT | Dropout Frame | $18.74 \pm 0.28$ | $18.86 \pm 0.28$ | $18.99 \pm 0.28$ | $19.06 \pm 0.28$ | $19.19 \pm 0.28$ | ✓ |
| Conformer CAT | AV Dropout Utt | $18.29 \pm 0.27$ | $18.51 \pm 0.27$ | $18.8 \pm 0.27$ | $19.0 \pm 0.28$ | $19.25 \pm 0.28$ | ✓ |
| Conformer CAT | Two-Pass | $18.83 \pm 0.28$ | $18.9 \pm 0.28$ | $18.89 \pm 0.28$ | $19.0 \pm 0.28$ | $19.05 \pm 0.28$ | ✓ |
| Con-LSTM CAT | Cascade Utt | $18.54 \pm 0.27$ | $18.64 \pm 0.28$ | $18.74 \pm 0.28$ | $18.78 \pm 0.28$ | $18.86 \pm 0.28$ | ✓ |
| LSTM CAT | Audio Baseline | $24.09 \pm 0.32$ | $24.09 \pm 0.32$ | $24.09 \pm 0.32$ | $24.09 \pm 0.32$ | $24.09 \pm 0.32$ | - |
| LSTM CAT | Vanilla | $20.61 \pm 0.29$ | $21.61 \pm 0.3$ | $22.73 \pm 0.32$ | $23.7 \pm 0.32$ | $24.83 \pm 0.33$ | ✗ |
| LSTM CAT | Cascade Utt | $21.27 \pm 0.29$ | $21.93 \pm 0.3$ | $22.64 \pm 0.31$ | $23.35 \pm 0.31$ | $24.05 \pm 0.32$ | ✓ |
| LSTM CAT | Dropout Utt | $21.64 \pm 0.3$ | $22.28 \pm 0.3$ | $22.98 \pm 0.31$ | $23.73 \pm 0.32$ | $24.36 \pm 0.32$ | ✓ |
| LSTM-Con CAT | Cascade Utt | $19.86 \pm 0.29$ | $20.88 \pm 0.3$ | $21.98 \pm 0.3$ | $23.03 \pm 0.31$ | $24.08 \pm 0.32$ | ✓ |
| Conformer CM | Audio Baseline | $18.92 \pm 0.28$ | $18.92 \pm 0.28$ | $18.92 \pm 0.28$ | $18.92 \pm 0.28$ | $18.92 \pm 0.28$ | - |
| Conformer CM | Vanilla | $21.79 \pm 0.31$ | $41.3 \pm 0.77$ | $60.07 \pm 0.85$ | $79.49 \pm 0.71$ | $98.31 \pm 0.05$ | ✗ |
| Conformer CM | Cascade Utt | $18.71 \pm 0.27$ | $18.76 \pm 0.27$ | $18.83 \pm 0.27$ | $18.87 \pm 0.27$ | $18.9 \pm 0.27$ | ✓ |
| Conformer CM | Dropout Utt | $21.15 \pm 0.3$ | $21.65 \pm 0.31$ | $22.07 \pm 0.32$ | $22.47 \pm 0.32$ | $22.95 \pm 0.33$ | ✗ |

Table 17: Robustness to $\mathcal{T}_{berFrame}$ for 10db. Columns three to seven represent the amount of dropped video frames.

| Architecture | Method | 0.0 | 0.25 | 0.5 | 0.75 | 1.0 | Robust |
|---|---|---|---|---|---|---|---|
| Conformer CAT | Audio Baseline | $19.05 \pm 0.28$ | $19.05 \pm 0.28$ | $19.05 \pm 0.28$ | $19.05 \pm 0.28$ | $19.05 \pm 0.28$ | - |
| Conformer CAT | Vanilla | $18.46 \pm 0.27$ | $18.51 \pm 0.27$ | $18.69 \pm 0.27$ | $19.0 \pm 0.28$ | $19.7 \pm 0.29$ | ✗ |
| Conformer CAT | Vanilla (25L) | $18.45 \pm 0.27$ | $18.51 \pm 0.27$ | $18.68 \pm 0.28$ | $19.01 \pm 0.28$ | $19.51 \pm 0.28$ | - |
| Conformer CAT | Cascade Utt | $18.47 \pm 0.27$ | $18.51 \pm 0.27$ | $18.63 \pm 0.27$ | $18.8 \pm 0.27$ | $18.84 \pm 0.27$ | ✓ |
| Conformer CAT | Dropout Utt | $18.23 \pm 0.27$ | $18.19 \pm 0.27$ | $18.33 \pm 0.27$ | $18.6 \pm 0.27$ | $18.91 \pm 0.28$ | ✓ |
| Conformer CAT | Cascade Frame | $18.41 \pm 0.27$ | $18.39 \pm 0.27$ | $18.55 \pm 0.27$ | $18.66 \pm 0.28$ | $18.91 \pm 0.28$ | ✓ |
| Conformer CAT | Dropout Frame | $18.74 \pm 0.28$ | $18.65 \pm 0.28$ | $18.82 \pm 0.28$ | $19.11 \pm 0.28$ | $19.19 \pm 0.28$ | ✓ |
| Conformer CAT | AV Dropout Utt | $18.29 \pm 0.27$ | $18.37 \pm 0.27$ | $18.53 \pm 0.27$ | $18.88 \pm 0.28$ | $19.25 \pm 0.28$ | ✓ |
| Conformer CAT | Two-Pass | $18.83 \pm 0.28$ | $18.84 \pm 0.28$ | $18.92 \pm 0.28$ | $19.07 \pm 0.28$ | $19.05 \pm 0.28$ | ✓ |
| Con-LSTM CAT | Cascade Utt | $18.54 \pm 0.27$ | $18.66 \pm 0.28$ | $18.76 \pm 0.28$ | $18.83 \pm 0.28$ | $18.86 \pm 0.28$ | ✓ |
| LSTM CAT | Audio Baseline | $24.09 \pm 0.32$ | $24.09 \pm 0.32$ | $24.09 \pm 0.32$ | $24.09 \pm 0.32$ | $24.09 \pm 0.32$ | - |
| LSTM CAT | Vanilla | $20.61 \pm 0.29$ | $21.12 \pm 0.3$ | $22.03 \pm 0.31$ | $23.47 \pm 0.32$ | $24.83 \pm 0.33$ | ✗ |
| LSTM CAT | Cascade Utt | $21.27 \pm 0.29$ | $22.2 \pm 0.3$ | $22.97 \pm 0.31$ | $23.66 \pm 0.32$ | $24.05 \pm 0.32$ | ✓ |
| LSTM CAT | Dropout Utt | $21.64 \pm 0.3$ | $22.4 \pm 0.31$ | $23.19 \pm 0.31$ | $23.95 \pm 0.32$ | $24.36 \pm 0.32$ | ✓ |
| LSTM-Con CAT | Cascade Utt | $19.86 \pm 0.29$ | $20.66 \pm 0.29$ | $21.75 \pm 0.3$ | $23.0 \pm 0.31$ | $24.08 \pm 0.32$ | ✓ |
| Conformer CM | Audio Baseline | $18.92 \pm 0.28$ | $18.92 \pm 0.28$ | $18.92 \pm 0.28$ | $18.92 \pm 0.28$ | $18.92 \pm 0.28$ | - |
| Conformer CM | Vanilla | $21.79 \pm 0.31$ | $23.32 \pm 0.33$ | $25.0 \pm 0.34$ | $29.84 \pm 0.39$ | $98.31 \pm 0.05$ | ✗ |
| Conformer CM | Cascade Utt | $18.71 \pm 0.27$ | $18.77 \pm 0.27$ | $18.89 \pm 0.28$ | $18.93 \pm 0.28$ | $18.9 \pm 0.27$ | ✓ |
| Conformer CM | Dropout Utt | $21.15 \pm 0.3$ | $27.76 \pm 0.37$ | $35.48 \pm 0.44$ | $55.64 \pm 0.53$ | $22.95 \pm 0.33$ | ✗ |

Table 18: Robustness to $\mathcal{T}_{start}$ for 10db. Columns three to seven represent the amount of dropped video frames.

| Architecture | Method | 0.0 | 0.25 | 0.5 | 0.75 | 1.0 | Robust |
|---|---|---|---|---|---|---|---|
| Conformer CAT | Audio Baseline | $19.05 \pm 0.28$ | $19.05 \pm 0.28$ | $19.05 \pm 0.28$ | $19.05 \pm 0.28$ | $19.05 \pm 0.28$ | - |
| Conformer CAT | Vanilla | $18.46 \pm 0.27$ | $18.67 \pm 0.27$ | $18.99 \pm 0.28$ | $19.31 \pm 0.28$ | $19.7 \pm 0.29$ | ✗ |
| Conformer CAT | Vanilla (25L) | $18.45 \pm 0.27$ | $18.63 \pm 0.28$ | $18.92 \pm 0.28$ | $19.21 \pm 0.28$ | $19.51 \pm 0.28$ | - |
| Conformer CAT | Cascade Utt | $18.47 \pm 0.27$ | $18.6 \pm 0.27$ | $18.68 \pm 0.27$ | $18.8 \pm 0.27$ | $18.84 \pm 0.27$ | ✓ |
| Conformer CAT | Dropout Utt | $18.23 \pm 0.27$ | $18.39 \pm 0.27$ | $18.54 \pm 0.27$ | $18.7 \pm 0.27$ | $18.91 \pm 0.28$ | ✓ |
| Conformer CAT | Cascade Frame | $18.41 \pm 0.27$ | $18.45 \pm 0.27$ | $18.58 \pm 0.27$ | $18.82 \pm 0.28$ | $18.91 \pm 0.28$ | ✓ |
| Conformer CAT | Dropout Frame | $18.74 \pm 0.28$ | $18.82 \pm 0.28$ | $18.93 \pm 0.28$ | $19.16 \pm 0.28$ | $19.19 \pm 0.28$ | ✓ |
| Conformer CAT | AV Dropout Utt | $18.29 \pm 0.27$ | $18.49 \pm 0.27$ | $18.74 \pm 0.27$ | $19.03 \pm 0.28$ | $19.25 \pm 0.28$ | ✓ |
| Conformer CAT | Two-Pass | $18.83 \pm 0.28$ | $18.94 \pm 0.28$ | $19.01 \pm 0.28$ | $19.07 \pm 0.28$ | $19.05 \pm 0.28$ | ✓ |
| Con-LSTM CAT | Cascade Utt | $18.54 \pm 0.27$ | $18.63 \pm 0.27$ | $18.68 \pm 0.28$ | $18.77 \pm 0.28$ | $18.86 \pm 0.28$ | ✓ |
| LSTM CAT | Audio Baseline | $24.09 \pm 0.32$ | $24.09 \pm 0.32$ | $24.09 \pm 0.32$ | $24.09 \pm 0.32$ | $24.09 \pm 0.32$ | - |
| LSTM CAT | Vanilla | $20.61 \pm 0.29$ | $21.58 \pm 0.3$ | $22.68 \pm 0.31$ | $23.7 \pm 0.32$ | $24.83 \pm 0.33$ | ✗ |
| LSTM CAT | Cascade Utt | $21.27 \pm 0.29$ | $22.07 \pm 0.3$ | $22.84 \pm 0.31$ | $23.53 \pm 0.32$ | $24.05 \pm 0.32$ | ✓ |
| LSTM CAT | Dropout Utt | $21.64 \pm 0.3$ | $22.38 \pm 0.31$ | $23.1 \pm 0.31$ | $23.86 \pm 0.32$ | $24.36 \pm 0.32$ | ✓ |
| LSTM-Con CAT | Cascade Utt | $19.86 \pm 0.29$ | $21.04 \pm 0.29$ | $22.13 \pm 0.3$ | $23.25 \pm 0.31$ | $24.08 \pm 0.32$ | ✓ |
| Conformer CM | Audio Baseline | $18.92 \pm 0.28$ | $18.92 \pm 0.28$ | $18.92 \pm 0.28$ | $18.92 \pm 0.28$ | $18.92 \pm 0.28$ | - |
| Conformer CM | Vanilla | $21.79 \pm 0.31$ | $23.02 \pm 0.32$ | $24.87 \pm 0.34$ | $30.24 \pm 0.4$ | $98.31 \pm 0.05$ | ✗ |
| Conformer CM | Cascade Utt | $18.71 \pm 0.27$ | $18.78 \pm 0.27$ | $18.89 \pm 0.28$ | $18.91 \pm 0.28$ | $18.9 \pm 0.27$ | ✓ |
| Conformer CM | Dropout Utt | $21.15 \pm 0.3$ | $26.22 \pm 0.36$ | $35.04 \pm 0.44$ | $54.45 \pm 0.54$ | $22.95 \pm 0.33$ | ✗ |

Table 19: Robustness to $\mathcal{T}_{mid}$ for 10db. Columns three to seven represent the amount of dropped video frames.

| Architecture | Method | 0.0 | 0.25 | 0.5 | 0.75 | 1.0 | Robust |
|---|---|---|---|---|---|---|---|
| Conformer CAT | Audio Baseline | $19.05 \pm 0.28$ | $19.05 \pm 0.28$ | $19.05 \pm 0.28$ | $19.05 \pm 0.28$ | $19.05 \pm 0.28$ | - |
| Conformer CAT | Vanilla | $18.46 \pm 0.27$ | $18.67 \pm 0.27$ | $18.98 \pm 0.28$ | $19.38 \pm 0.28$ | $19.7 \pm 0.29$ | ✗ |
| Conformer CAT | Vanilla (25L) | $18.45 \pm 0.27$ | $18.61 \pm 0.27$ | $18.93 \pm 0.28$ | $19.2 \pm 0.28$ | $19.51 \pm 0.28$ | - |
| Conformer CAT | Cascade Utt | $18.47 \pm 0.27$ | $18.56 \pm 0.27$ | $18.69 \pm 0.27$ | $18.79 \pm 0.27$ | $18.84 \pm 0.27$ | ✓ |
| Conformer CAT | Dropout Utt | $18.23 \pm 0.27$ | $18.38 \pm 0.27$ | $18.57 \pm 0.27$ | $18.78 \pm 0.28$ | $18.91 \pm 0.28$ | ✓ |
| Conformer CAT | Cascade Frame | $18.41 \pm 0.27$ | $18.45 \pm 0.27$ | $18.58 \pm 0.27$ | $18.87 \pm 0.28$ | $18.91 \pm 0.28$ | ✓ |
| Conformer CAT | Dropout Frame | $18.74 \pm 0.28$ | $18.78 \pm 0.28$ | $18.97 \pm 0.28$ | $19.17 \pm 0.28$ | $19.19 \pm 0.28$ | ✓ |
| Conformer CAT | AV Dropout Utt | $18.29 \pm 0.27$ | $18.49 \pm 0.27$ | $18.81 \pm 0.28$ | $19.05 \pm 0.28$ | $19.25 \pm 0.28$ | ✓ |
| Conformer CAT | Two-Pass | $18.83 \pm 0.28$ | $18.88 \pm 0.28$ | $18.99 \pm 0.28$ | $19.07 \pm 0.28$ | $19.05 \pm 0.28$ | ✓ |
| Con-LSTM CAT | Cascade Utt | $18.54 \pm 0.27$ | $18.63 \pm 0.27$ | $18.71 \pm 0.28$ | $18.8 \pm 0.28$ | $18.86 \pm 0.28$ | ✓ |
| LSTM CAT | Audio Baseline | $24.09 \pm 0.32$ | $24.09 \pm 0.32$ | $24.09 \pm 0.32$ | $24.09 \pm 0.32$ | $24.09 \pm 0.32$ | - |
| LSTM CAT | Vanilla | $20.61 \pm 0.29$ | $21.67 \pm 0.3$ | $22.82 \pm 0.31$ | $23.81 \pm 0.32$ | $24.83 \pm 0.33$ | ✗ |
| LSTM CAT | Cascade Utt | $21.27 \pm 0.29$ | $22.1 \pm 0.3$ | $22.89 \pm 0.31$ | $23.61 \pm 0.31$ | $24.05 \pm 0.32$ | ✓ |
| LSTM CAT | Dropout Utt | $21.64 \pm 0.3$ | $22.5 \pm 0.31$ | $23.28 \pm 0.31$ | $23.99 \pm 0.32$ | $24.36 \pm 0.32$ | ✓ |
| LSTM-Con CAT | Cascade Utt | $19.86 \pm 0.29$ | $20.99 \pm 0.29$ | $22.09 \pm 0.3$ | $23.19 \pm 0.31$ | $24.08 \pm 0.32$ | ✓ |
| Conformer CM | Audio Baseline | $18.92 \pm 0.28$ | $18.92 \pm 0.28$ | $18.92 \pm 0.28$ | $18.92 \pm 0.28$ | $18.92 \pm 0.28$ | - |
| Conformer CM | Vanilla | $21.79 \pm 0.31$ | $23.29 \pm 0.32$ | $25.1 \pm 0.34$ | $31.25 \pm 0.42$ | $98.31 \pm 0.05$ | ✗ |
| Conformer CM | Cascade Utt | $18.71 \pm 0.27$ | $18.76 \pm 0.27$ | $18.83 \pm 0.28$ | $18.89 \pm 0.28$ | $18.9 \pm 0.27$ | ✓ |
| Conformer CM | Dropout Utt | $21.15 \pm 0.3$ | $27.38 \pm 0.37$ | $37.39 \pm 0.45$ | $56.38 \pm 0.53$ | $22.95 \pm 0.33$ | ✗ |

Table 20: Robustness to $\mathcal{T}_{end}$ for 10db. Columns three to seven represent the amount of dropped video frames.

| Architecture | Method | 0.0 | 0.25 | 0.5 | 0.75 | 1.0 | Robust |
|---|---|---|---|---|---|---|---|
| Conformer CAT | Audio Baseline | $19.05 \pm 0.28$ | $19.05 \pm 0.28$ | $19.05 \pm 0.28$ | $19.05 \pm 0.28$ | $19.05 \pm 0.28$ | - |
| Conformer CAT | Vanilla | $18.46 \pm 0.27$ | $18.78 \pm 0.27$ | $19.1 \pm 0.28$ | $19.41 \pm 0.28$ | $19.7 \pm 0.29$ | ✗ |
| Conformer CAT | Vanilla (25L) | $18.45 \pm 0.27$ | $18.74 \pm 0.27$ | $18.98 \pm 0.28$ | $19.24 \pm 0.28$ | $19.51 \pm 0.28$ | - |
| Conformer CAT | Cascade Utt | $18.47 \pm 0.27$ | $18.61 \pm 0.27$ | $18.7 \pm 0.27$ | $18.8 \pm 0.27$ | $18.84 \pm 0.27$ | ✓ |
| Conformer CAT | Dropout Utt | $18.23 \pm 0.27$ | $18.48 \pm 0.27$ | $18.63 \pm 0.27$ | $18.78 \pm 0.28$ | $18.91 \pm 0.28$ | ✓ |
| Conformer CAT | Cascade Frame | $18.41 \pm 0.27$ | $18.55 \pm 0.27$ | $18.72 \pm 0.27$ | $18.97 \pm 0.28$ | $18.91 \pm 0.28$ | ✓ |
| Conformer CAT | Dropout Frame | $18.74 \pm 0.28$ | $18.79 \pm 0.28$ | $18.99 \pm 0.28$ | $19.16 \pm 0.28$ | $19.19 \pm 0.28$ | ✓ |
| Conformer CAT | AV Dropout Utt | $18.29 \pm 0.27$ | $18.62 \pm 0.27$ | $18.87 \pm 0.27$ | $19.09 \pm 0.28$ | $19.25 \pm 0.28$ | ✓ |
| Conformer CAT | Two-Pass | $18.83 \pm 0.28$ | $18.84 \pm 0.28$ | $18.92 \pm 0.28$ | $19.03 \pm 0.28$ | $19.05 \pm 0.28$ | ✓ |
| Con-LSTM CAT | Cascade Utt | $18.54 \pm 0.27$ | $18.64 \pm 0.28$ | $18.73 \pm 0.28$ | $18.79 \pm 0.28$ | $18.86 \pm 0.28$ | ✓ |
| LSTM CAT | Audio Baseline | $24.09 \pm 0.32$ | $24.09 \pm 0.32$ | $24.09 \pm 0.32$ | $24.09 \pm 0.32$ | $24.09 \pm 0.32$ | - |
| LSTM CAT | Vanilla | $20.61 \pm 0.29$ | $21.8 \pm 0.3$ | $22.96 \pm 0.31$ | $23.8 \pm 0.32$ | $24.83 \pm 0.33$ | ✗ |
| LSTM CAT | Cascade Utt | $21.27 \pm 0.29$ | $22.08 \pm 0.3$ | $22.83 \pm 0.31$ | $23.53 \pm 0.31$ | $24.05 \pm 0.32$ | ✓ |
| LSTM CAT | Dropout Utt | $21.64 \pm 0.3$ | $22.41 \pm 0.3$ | $23.25 \pm 0.31$ | $23.92 \pm 0.32$ | $24.36 \pm 0.32$ | ✓ |
| LSTM-Con CAT | Cascade Utt | $19.86 \pm 0.29$ | $20.8 \pm 0.29$ | $21.97 \pm 0.3$ | $23.03 \pm 0.31$ | $24.08 \pm 0.32$ | ✓ |
| Conformer CM | Audio Baseline | $18.92 \pm 0.28$ | $18.92 \pm 0.28$ | $18.92 \pm 0.28$ | $18.92 \pm 0.28$ | $18.92 \pm 0.28$ | - |
| Conformer CM | Vanilla | $21.79 \pm 0.31$ | $23.21 \pm 0.32$ | $24.99 \pm 0.34$ | $29.99 \pm 0.4$ | $98.31 \pm 0.05$ | ✗ |
| Conformer CM | Cascade Utt | $18.71 \pm 0.27$ | $18.76 \pm 0.27$ | $18.84 \pm 0.28$ | $18.9 \pm 0.28$ | $18.9 \pm 0.27$ | ✓ |
| Conformer CM | Dropout Utt | $21.15 \pm 0.3$ | $26.62 \pm 0.35$ | $35.14 \pm 0.42$ | $54.16 \pm 0.51$ | $22.95 \pm 0.33$ | ✗ |

Table 21: Robustness to $\mathcal{T}_{rate}$ for 10db. Columns three to eight represent the amount of dropped video frames.

| Architecture | Method | 0 | $\frac{1}{128}$ | $\frac{1}{32}$ | $\frac{1}{8}$ | $\frac{1}{2}$ | 1 | Robust |
|---|---|---|---|---|---|---|---|---|
| Conformer CAT | Audio Baseline | $19.05 \pm 0.28$ | $19.05 \pm 0.28$ | $19.05 \pm 0.28$ | $19.05 \pm 0.28$ | $19.05 \pm 0.28$ | $19.05 \pm 0.28$ | - |
| Conformer CAT | Vanilla | $18.46 \pm 0.27$ | $18.45 \pm 0.27$ | $18.45 \pm 0.27$ | $18.44 \pm 0.27$ | $18.64 \pm 0.27$ | $19.7 \pm 0.29$ | ✗ |
| Conformer CAT | Vanilla (25L) | $18.45 \pm 0.27$ | $18.44 \pm 0.27$ | $18.46 \pm 0.27$ | $18.44 \pm 0.27$ | $18.74 \pm 0.28$ | $19.51 \pm 0.28$ | - |
| Conformer CAT | Cascade Utt | $18.47 \pm 0.27$ | $18.47 \pm 0.27$ | $18.47 \pm 0.27$ | $18.49 \pm 0.27$ | $18.62 \pm 0.27$ | $18.84 \pm 0.27$ | ✓ |
| Conformer CAT | Dropout Utt | $18.23 \pm 0.27$ | $18.25 \pm 0.27$ | $18.23 \pm 0.27$ | $18.2 \pm 0.27$ | $18.34 \pm 0.27$ | $18.91 \pm 0.28$ | ✓ |
| Conformer CAT | Cascade Frame | $18.41 \pm 0.27$ | $18.41 \pm 0.27$ | $18.39 \pm 0.27$ | $18.34 \pm 0.27$ | $18.5 \pm 0.27$ | $18.91 \pm 0.28$ | ✓ |
| Conformer CAT | Dropout Frame | $18.74 \pm 0.28$ | $18.75 \pm 0.28$ | $18.72 \pm 0.28$ | $18.59 \pm 0.28$ | $18.76 \pm 0.28$ | $19.19 \pm 0.28$ | ✓ |
| Conformer CAT | AV Dropout Utt | $18.29 \pm 0.27$ | $18.28 \pm 0.27$ | $18.3 \pm 0.27$ | $18.31 \pm 0.27$ | $18.56 \pm 0.27$ | $19.25 \pm 0.28$ | ✓ |
| Conformer CAT | Two-Pass | $18.83 \pm 0.28$ | $18.83 \pm 0.28$ | $18.84 \pm 0.28$ | $18.81 \pm 0.28$ | $18.93 \pm 0.28$ | $19.05 \pm 0.28$ | ✓ |
| Con-LSTM CAT | Cascade Utt | $18.54 \pm 0.27$ | $18.53 \pm 0.27$ | $18.55 \pm 0.27$ | $18.61 \pm 0.28$ | $18.78 \pm 0.28$ | $18.86 \pm 0.28$ | ✓ |
| LSTM CAT | Audio Baseline | $24.09 \pm 0.32$ | $24.09 \pm 0.32$ | $24.09 \pm 0.32$ | $24.09 \pm 0.32$ | $24.09 \pm 0.32$ | $24.09 \pm 0.32$ | - |
| LSTM CAT | Vanilla | $20.61 \pm 0.29$ | $20.63 \pm 0.29$ | $20.65 \pm 0.29$ | $20.86 \pm 0.29$ | $21.82 \pm 0.3$ | $24.83 \pm 0.33$ | ✗ |
| LSTM CAT | Cascade Utt | $21.27 \pm 0.29$ | $21.25 \pm 0.29$ | $21.36 \pm 0.29$ | $21.64 \pm 0.3$ | $22.92 \pm 0.31$ | $24.05 \pm 0.32$ | ✓ |
| LSTM CAT | Dropout Utt | $21.64 \pm 0.3$ | $21.64 \pm 0.3$ | $21.69 \pm 0.3$ | $21.91 \pm 0.3$ | $23.18 \pm 0.31$ | $24.36 \pm 0.32$ | ✓ |
| LSTM-Con CAT | Cascade Utt | $19.86 \pm 0.29$ | $19.86 \pm 0.28$ | $19.97 \pm 0.29$ | $20.17 \pm 0.29$ | $21.54 \pm 0.3$ | $24.08 \pm 0.32$ | ✓ |
| Conformer CM | Audio Baseline | $18.92 \pm 0.28$ | $18.92 \pm 0.28$ | $18.92 \pm 0.28$ | $18.92 \pm 0.28$ | $18.92 \pm 0.28$ | $18.92 \pm 0.28$ | - |
| Conformer CM | Vanilla | $21.79 \pm 0.31$ | $21.79 \pm 0.31$ | $21.92 \pm 0.31$ | $22.51 \pm 0.32$ | $25.35 \pm 0.35$ | $98.31 \pm 0.05$ | ✗ |
| Conformer CM | Cascade Utt | $18.71 \pm 0.27$ | $18.68 \pm 0.27$ | $18.7 \pm 0.27$ | $18.71 \pm 0.27$ | $18.89 \pm 0.28$ | $18.9 \pm 0.27$ | ✓ |
| Conformer CM | Dropout Utt | $21.15 \pm 0.3$ | $21.24 \pm 0.3$ | $21.81 \pm 0.31$ | $24.37 \pm 0.34$ | $33.56 \pm 0.42$ | $22.95 \pm 0.33$ | ✗ |

Table 22: Robustness to $\mathcal{T}_{berUtt}$ for 0db. Columns three to seven represent the amount of dropped video frames.

| Architecture | Method | 0.0 | 0.25 | 0.5 | 0.75 | 1.0 | Robust |
|---|---|---|---|---|---|---|---|
| Conformer CAT | Audio Baseline | $33.54 \pm 0.43$ | $33.54 \pm 0.43$ | $33.54 \pm 0.43$ | $33.54 \pm 0.43$ | $33.54 \pm 0.43$ | - |
| Conformer CAT | Vanilla | $24.96 \pm 0.35$ | $27.59 \pm 0.38$ | $30.22 \pm 0.4$ | $32.83 \pm 0.41$ | $35.51 \pm 0.43$ | ✗ |
| Conformer CAT | Vanilla (25L) | $24.75 \pm 0.35$ | $27.23 \pm 0.38$ | $29.8 \pm 0.41$ | $32.35 \pm 0.42$ | $34.88 \pm 0.43$ | - |
| Conformer CAT | Cascade Utt | $26.12 \pm 0.36$ | $27.31 \pm 0.38$ | $28.58 \pm 0.38$ | $29.78 \pm 0.39$ | $31.08 \pm 0.4$ | ✓ |
| Conformer CAT | Dropout Utt | $24.33 \pm 0.34$ | $26.14 \pm 0.36$ | $27.94 \pm 0.38$ | $29.81 \pm 0.39$ | $31.67 \pm 0.4$ | ✓ |
| Conformer CAT | Cascade Frame | $28.04 \pm 0.38$ | $29.5 \pm 0.4$ | $31.08 \pm 0.41$ | $32.71 \pm 0.43$ | $34.17 \pm 0.44$ | ✗ |
| Conformer CAT | Dropout Frame | $27.58 \pm 0.37$ | $28.83 \pm 0.38$ | $30.11 \pm 0.39$ | $31.23 \pm 0.41$ | $32.54 \pm 0.41$ | ✓ |
| Conformer CAT | AV Dropout Utt | $23.64 \pm 0.33$ | $26.13 \pm 0.36$ | $28.58 \pm 0.38$ | $31.05 \pm 0.4$ | $33.55 \pm 0.41$ | ✓ |
| Conformer CAT | Two-Pass | $30.06 \pm 0.42$ | $30.83 \pm 0.42$ | $31.78 \pm 0.43$ | $32.68 \pm 0.43$ | $33.54 \pm 0.43$ | ✓ |
| Con-LSTM CAT | Cascade Utt | $27.9 \pm 0.39$ | $28.81 \pm 0.4$ | $29.68 \pm 0.4$ | $30.53 \pm 0.4$ | $31.4 \pm 0.41$ | ✓ |
| LSTM CAT | Audio Baseline | $46.38 \pm 0.44$ | $46.38 \pm 0.44$ | $46.38 \pm 0.44$ | $46.38 \pm 0.44$ | $46.38 \pm 0.44$ | - |
| LSTM CAT | Vanilla | $30.93 \pm 0.39$ | $36.45 \pm 0.45$ | $41.02 \pm 0.48$ | $46.55 \pm 0.47$ | $51.53 \pm 0.45$ | ✗ |
| LSTM CAT | Cascade Utt | $33.67 \pm 0.4$ | $36.92 \pm 0.42$ | $40.27 \pm 0.45$ | $43.23 \pm 0.45$ | $46.55 \pm 0.44$ | ✓ |
| LSTM CAT | Dropout Utt | $33.41 \pm 0.4$ | $37.06 \pm 0.43$ | $40.87 \pm 0.45$ | $44.44 \pm 0.46$ | $48.13 \pm 0.44$ | ✗ |
| LSTM-Con CAT | Cascade Utt | $29.37 \pm 0.37$ | $33.74 \pm 0.42$ | $37.79 \pm 0.45$ | $42.22 \pm 0.45$ | $46.55 \pm 0.44$ | ✓ |
| Conformer CM | Audio Baseline | $31.79 \pm 0.42$ | $31.79 \pm 0.42$ | $31.79 \pm 0.42$ | $31.79 \pm 0.42$ | $31.79 \pm 0.42$ | - |
| Conformer CM | Vanilla | $34.91 \pm 0.43$ | $51.42 \pm 0.74$ | $66.58 \pm 0.78$ | $82.34 \pm 0.68$ | $98.34 \pm 0.05$ | ✗ |
| Conformer CM | Cascade Utt | $27.89 \pm 0.37$ | $28.83 \pm 0.38$ | $29.7 \pm 0.39$ | $30.6 \pm 0.4$ | $31.56 \pm 0.4$ | ✓ |
| Conformer CM | Dropout Utt | $33.11 \pm 0.43$ | $36.12 \pm 0.45$ | $39.36 \pm 0.47$ | $42.43 \pm 0.48$ | $45.43 \pm 0.47$ | ✗ |

Table 23: Robustness to $\mathcal{T}_{berFrame}$ for 0db. Columns three to seven represent the amount of dropped video frames.

| Architecture | Method | 0.0 | 0.25 | 0.5 | 0.75 | 1.0 | Robust |
|---|---|---|---|---|---|---|---|
| Conformer CAT | Audio Baseline | $33.54 \pm 0.43$ | $33.54 \pm 0.43$ | $33.54 \pm 0.43$ | $33.54 \pm 0.43$ | $33.54 \pm 0.43$ | - |
| Conformer CAT | Vanilla | $24.96 \pm 0.35$ | $25.41 \pm 0.35$ | $26.61 \pm 0.36$ | $29.44 \pm 0.38$ | $35.51 \pm 0.43$ | ✗ |
| Conformer CAT | Vanilla (25L) | $24.75 \pm 0.35$ | $25.39 \pm 0.36$ | $26.8 \pm 0.37$ | $29.6 \pm 0.39$ | $34.88 \pm 0.43$ | - |
| Conformer CAT | Cascade Utt | $26.12 \pm 0.36$ | $27.5 \pm 0.37$ | $29.07 \pm 0.38$ | $30.44 \pm 0.39$ | $31.08 \pm 0.4$ | ✓ |
| Conformer CAT | Dropout Utt | $24.33 \pm 0.34$ | $25.08 \pm 0.35$ | $26.48 \pm 0.36$ | $28.88 \pm 0.38$ | $31.67 \pm 0.4$ | ✓ |
| Conformer CAT | Cascade Frame | $28.04 \pm 0.38$ | $28.84 \pm 0.4$ | $30.17 \pm 0.41$ | $32.19 \pm 0.43$ | $34.17 \pm 0.44$ | ✗ |
| Conformer CAT | Dropout Frame | $27.58 \pm 0.37$ | $26.74 \pm 0.36$ | $28.0 \pm 0.38$ | $30.23 \pm 0.4$ | $32.54 \pm 0.41$ | ✗ |
| Conformer CAT | AV Dropout Utt | $23.64 \pm 0.33$ | $24.28 \pm 0.33$ | $25.83 \pm 0.35$ | $29.2 \pm 0.38$ | $33.55 \pm 0.41$ | ✓ |
| Conformer CAT | Two-Pass | $30.06 \pm 0.42$ | $31.01 \pm 0.43$ | $32.1 \pm 0.43$ | $33.14 \pm 0.43$ | $33.54 \pm 0.43$ | ✓ |
| Con-LSTM CAT | Cascade Utt | $27.9 \pm 0.39$ | $29.59 \pm 0.4$ | $30.52 \pm 0.4$ | $31.15 \pm 0.41$ | $31.4 \pm 0.41$ | ✓ |
| LSTM CAT | Audio Baseline | $46.38 \pm 0.44$ | $46.38 \pm 0.44$ | $46.38 \pm 0.44$ | $46.38 \pm 0.44$ | $46.38 \pm 0.44$ | - |
| LSTM CAT | Vanilla | $30.93 \pm 0.39$ | $33.55 \pm 0.41$ | $38.28 \pm 0.43$ | $45.45 \pm 0.46$ | $51.53 \pm 0.45$ | ✗ |
| LSTM CAT | Cascade Utt | $33.67 \pm 0.4$ | $38.23 \pm 0.42$ | $42.3 \pm 0.43$ | $45.23 \pm 0.44$ | $46.55 \pm 0.44$ | ✓ |
| LSTM CAT | Dropout Utt | $33.41 \pm 0.4$ | $37.63 \pm 0.42$ | $42.18 \pm 0.44$ | $45.98 \pm 0.45$ | $48.13 \pm 0.44$ | ✗ |
| LSTM-Con CAT | Cascade Utt | $29.37 \pm 0.37$ | $33.23 \pm 0.39$ | $38.42 \pm 0.42$ | $43.39 \pm 0.44$ | $46.55 \pm 0.44$ | ✓ |
| Conformer CM | Audio Baseline | $31.79 \pm 0.42$ | $31.79 \pm 0.42$ | $31.79 \pm 0.42$ | $31.79 \pm 0.42$ | $31.79 \pm 0.42$ | - |
| Conformer CM | Vanilla | $34.91 \pm 0.43$ | $42.36 \pm 0.47$ | $49.48 \pm 0.5$ | $61.87 \pm 0.51$ | $98.34 \pm 0.05$ | ✗ |
| Conformer CM | Cascade Utt | $27.89 \pm 0.37$ | $29.23 \pm 0.39$ | $30.6 \pm 0.4$ | $31.41 \pm 0.4$ | $31.56 \pm 0.4$ | ✓ |
| Conformer CM | Dropout Utt | $33.11 \pm 0.43$ | $55.38 \pm 0.52$ | $67.77 \pm 0.5$ | $79.58 \pm 0.46$ | $45.43 \pm 0.47$ | ✗ |

Table 24: Robustness to $\mathcal{T}_{start}$ for 0db. Columns three to seven represent the amount of dropped video frames.

| ARCHITECTURE | METHOD | 0.0 | 0.25 | 0.5 | 0.75 | 1.0 | ROBUST |
|---|---|---|---|---|---|---|---|
| CONFORMER CAT | AUDIO BASELINE | $33.54 \pm 0.43$ | $33.54 \pm 0.43$ | $33.54 \pm 0.43$ | $33.54 \pm 0.43$ | $33.54 \pm 0.43$ | - |
| CONFORMER CAT | VANILLA | $24.96 \pm 0.35$ | $26.68 \pm 0.36$ | $29.06 \pm 0.37$ | $32.07 \pm 0.4$ | $35.51 \pm 0.43$ | ✗ |
| CONFORMER CAT | VANILLA (25L) | $24.75 \pm 0.35$ | $26.58 \pm 0.36$ | $29.13 \pm 0.38$ | $31.97 \pm 0.41$ | $34.88 \pm 0.43$ | - |
| CONFORMER CAT | CASCADE UTT | $26.12 \pm 0.36$ | $27.49 \pm 0.38$ | $28.83 \pm 0.38$ | $30.16 \pm 0.39$ | $31.08 \pm 0.4$ | ✓ |
| CONFORMER CAT | DROPOUT UTT | $24.33 \pm 0.34$ | $25.93 \pm 0.35$ | $27.79 \pm 0.37$ | $29.79 \pm 0.39$ | $31.67 \pm 0.4$ | ✓ |
| CONFORMER CAT | CASCADE FRAME | $28.04 \pm 0.38$ | $29.24 \pm 0.4$ | $30.96 \pm 0.42$ | $33.01 \pm 0.43$ | $34.17 \pm 0.44$ | ✗ |
| CONFORMER CAT | DROPOUT FRAME | $27.58 \pm 0.37$ | $28.11 \pm 0.37$ | $29.55 \pm 0.39$ | $31.36 \pm 0.41$ | $32.54 \pm 0.41$ | ✓ |
| CONFORMER CAT | AV DROPOUT UTT | $23.64 \pm 0.33$ | $25.67 \pm 0.34$ | $28.27 \pm 0.37$ | $31.04 \pm 0.39$ | $33.55 \pm 0.41$ | ✓ |
| CONFORMER CAT | TWO-PASS | $30.06 \pm 0.42$ | $30.99 \pm 0.43$ | $32.0 \pm 0.43$ | $33.01 \pm 0.43$ | $33.54 \pm 0.43$ | ✓ |
| CON-LSTM CAT | CASCADE UTT | $27.9 \pm 0.39$ | $28.76 \pm 0.39$ | $29.64 \pm 0.4$ | $30.66 \pm 0.41$ | $31.4 \pm 0.41$ | ✓ |
| LSTM CAT | AUDIO BASELINE | $46.38 \pm 0.44$ | $46.38 \pm 0.44$ | $46.38 \pm 0.44$ | $46.38 \pm 0.44$ | $46.38 \pm 0.44$ | - |
| LSTM CAT | VANILLA | $30.93 \pm 0.39$ | $35.95 \pm 0.42$ | $41.9 \pm 0.45$ | $47.03 \pm 0.45$ | $51.53 \pm 0.45$ | ✗ |
| LSTM CAT | CASCADE UTT | $33.67 \pm 0.4$ | $37.38 \pm 0.42$ | $41.07 \pm 0.43$ | $44.41 \pm 0.44$ | $46.55 \pm 0.44$ | ✓ |
| LSTM CAT | DROPOUT UTT | $33.41 \pm 0.4$ | $37.26 \pm 0.41$ | $41.71 \pm 0.43$ | $45.45 \pm 0.44$ | $48.13 \pm 0.44$ | ✗ |
| LSTM-CON CAT | CASCADE UTT | $29.37 \pm 0.37$ | $34.07 \pm 0.4$ | $38.71 \pm 0.42$ | $43.41 \pm 0.44$ | $46.55 \pm 0.44$ | ✓ |
| CONFORMER CM | AUDIO BASELINE | $31.79 \pm 0.42$ | $31.79 \pm 0.42$ | $31.79 \pm 0.42$ | $31.79 \pm 0.42$ | $31.79 \pm 0.42$ | - |
| CONFORMER CM | VANILLA | $34.91 \pm 0.43$ | $41.37 \pm 0.45$ | $49.26 \pm 0.48$ | $61.27 \pm 0.52$ | $98.34 \pm 0.05$ | ✗ |
| CONFORMER CM | CASCADE UTT | $27.89 \pm 0.37$ | $28.96 \pm 0.39$ | $29.97 \pm 0.39$ | $31.02 \pm 0.4$ | $31.56 \pm 0.4$ | ✓ |
| CONFORMER CM | DROPOUT UTT | $33.11 \pm 0.43$ | $48.2 \pm 0.51$ | $61.94 \pm 0.52$ | $75.46 \pm 0.49$ | $45.43 \pm 0.47$ | ✗ |

Table 25: Robustness to $\mathcal{T}_{mid}$ for 0db. Columns three to seven represent the amount of dropped video frames.

| ARCHITECTURE | METHOD | 0.0 | 0.25 | 0.5 | 0.75 | 1.0 | ROBUST |
|---|---|---|---|---|---|---|---|
| CONFORMER CAT | AUDIO BASELINE | $33.54 \pm 0.43$ | $33.54 \pm 0.43$ | $33.54 \pm 0.43$ | $33.54 \pm 0.43$ | $33.54 \pm 0.43$ | - |
| CONFORMER CAT | VANILLA | $24.96 \pm 0.35$ | $26.95 \pm 0.36$ | $29.51 \pm 0.38$ | $32.5 \pm 0.4$ | $35.51 \pm 0.43$ | ✗ |
| CONFORMER CAT | VANILLA (25L) | $24.75 \pm 0.35$ | $26.84 \pm 0.36$ | $29.55 \pm 0.38$ | $32.27 \pm 0.41$ | $34.88 \pm 0.43$ | - |
| CONFORMER CAT | CASCADE UTT | $26.12 \pm 0.36$ | $27.74 \pm 0.38$ | $29.11 \pm 0.38$ | $30.32 \pm 0.39$ | $31.08 \pm 0.4$ | ✓ |
| CONFORMER CAT | DROPOUT UTT | $24.33 \pm 0.34$ | $26.24 \pm 0.36$ | $28.18 \pm 0.37$ | $30.22 \pm 0.39$ | $31.67 \pm 0.4$ | ✓ |
| CONFORMER CAT | CASCADE FRAME | $28.04 \pm 0.38$ | $29.26 \pm 0.39$ | $30.99 \pm 0.41$ | $33.03 \pm 0.43$ | $34.17 \pm 0.44$ | ✗ |
| CONFORMER CAT | DROPOUT FRAME | $27.58 \pm 0.37$ | $28.13 \pm 0.37$ | $29.82 \pm 0.39$ | $31.55 \pm 0.4$ | $32.54 \pm 0.41$ | ✓ |
| CONFORMER CAT | AV DROPOUT UTT | $23.64 \pm 0.33$ | $25.9 \pm 0.34$ | $28.63 \pm 0.37$ | $31.42 \pm 0.39$ | $33.55 \pm 0.41$ | ✓ |
| CONFORMER CAT | TWO-PASS | $30.06 \pm 0.42$ | $31.17 \pm 0.42$ | $32.16 \pm 0.43$ | $33.13 \pm 0.43$ | $33.54 \pm 0.43$ | ✓ |
| CON-LSTM CAT | CASCADE UTT | $27.9 \pm 0.39$ | $29.02 \pm 0.39$ | $29.99 \pm 0.4$ | $30.85 \pm 0.41$ | $31.4 \pm 0.41$ | ✓ |
| LSTM CAT | AUDIO BASELINE | $46.38 \pm 0.44$ | $46.38 \pm 0.44$ | $46.38 \pm 0.44$ | $46.38 \pm 0.44$ | $46.38 \pm 0.44$ | - |
| LSTM CAT | VANILLA | $30.93 \pm 0.39$ | $36.53 \pm 0.41$ | $42.25 \pm 0.43$ | $47.31 \pm 0.44$ | $51.53 \pm 0.45$ | ✗ |
| LSTM CAT | CASCADE UTT | $33.67 \pm 0.4$ | $37.95 \pm 0.41$ | $41.73 \pm 0.43$ | $44.84 \pm 0.44$ | $46.55 \pm 0.44$ | ✓ |
| LSTM CAT | DROPOUT UTT | $33.41 \pm 0.4$ | $38.01 \pm 0.41$ | $42.25 \pm 0.43$ | $45.93 \pm 0.44$ | $48.13 \pm 0.44$ | ✗ |
| LSTM-CON CAT | CASCADE UTT | $29.37 \pm 0.37$ | $34.44 \pm 0.39$ | $39.16 \pm 0.42$ | $43.59 \pm 0.43$ | $46.55 \pm 0.44$ | ✓ |
| CONFORMER CM | AUDIO BASELINE | $31.79 \pm 0.42$ | $31.79 \pm 0.42$ | $31.79 \pm 0.42$ | $31.79 \pm 0.42$ | $31.79 \pm 0.42$ | - |
| CONFORMER CM | VANILLA | $34.91 \pm 0.43$ | $42.26 \pm 0.45$ | $49.84 \pm 0.48$ | $62.43 \pm 0.52$ | $98.34 \pm 0.05$ | ✗ |
| CONFORMER CM | CASCADE UTT | $27.89 \pm 0.37$ | $29.07 \pm 0.39$ | $30.17 \pm 0.39$ | $31.14 \pm 0.4$ | $31.56 \pm 0.4$ | ✓ |
| CONFORMER CM | DROPOUT UTT | $33.11 \pm 0.43$ | $50.73 \pm 0.51$ | $64.11 \pm 0.51$ | $75.94 \pm 0.47$ | $45.43 \pm 0.47$ | ✗ |

Table 26: Robustness to $\mathcal{T}_{end}$ for 0db. Columns three to seven represent the amount of dropped video frames.

| ARCHITECTURE | METHOD | 0.0 | 0.25 | 0.5 | 0.75 | 1.0 | ROBUST |
|---|---|---|---|---|---|---|---|
| CONFORMER CAT | AUDIO BASELINE | $33.54 \pm 0.43$ | $33.54 \pm 0.43$ | $33.54 \pm 0.43$ | $33.54 \pm 0.43$ | $33.54 \pm 0.43$ | - |
| CONFORMER CAT | VANILLA | $24.96 \pm 0.35$ | $27.36 \pm 0.36$ | $30.03 \pm 0.38$ | $32.67 \pm 0.4$ | $35.51 \pm 0.43$ | ✗ |
| CONFORMER CAT | VANILLA (25L) | $24.75 \pm 0.35$ | $27.27 \pm 0.36$ | $29.97 \pm 0.38$ | $32.51 \pm 0.4$ | $34.88 \pm 0.43$ | - |
| CONFORMER CAT | CASCADE UTT | $26.12 \pm 0.36$ | $27.77 \pm 0.37$ | $29.13 \pm 0.38$ | $30.26 \pm 0.39$ | $31.08 \pm 0.4$ | ✓ |
| CONFORMER CAT | DROPOUT UTT | $24.33 \pm 0.34$ | $26.5 \pm 0.35$ | $28.56 \pm 0.37$ | $30.33 \pm 0.39$ | $31.67 \pm 0.4$ | ✓ |
| CONFORMER CAT | CASCADE FRAME | $28.04 \pm 0.38$ | $29.46 \pm 0.38$ | $31.18 \pm 0.4$ | $33.04 \pm 0.42$ | $34.17 \pm 0.44$ | ✗ |
| CONFORMER CAT | DROPOUT FRAME | $27.58 \pm 0.37$ | $28.26 \pm 0.37$ | $29.96 \pm 0.39$ | $31.64 \pm 0.4$ | $32.54 \pm 0.41$ | ✓ |
| CONFORMER CAT | AV DROPOUT UTT | $23.64 \pm 0.33$ | $26.3 \pm 0.34$ | $29.04 \pm 0.36$ | $31.57 \pm 0.39$ | $33.55 \pm 0.41$ | ✓ |
| CONFORMER CAT | TWO-PASS | $30.06 \pm 0.42$ | $30.94 \pm 0.42$ | $31.93 \pm 0.43$ | $32.87 \pm 0.43$ | $33.54 \pm 0.43$ | ✓ |
| CON-LSTM CAT | CASCADE UTT | $27.9 \pm 0.39$ | $29.01 \pm 0.39$ | $30.06 \pm 0.4$ | $30.88 \pm 0.41$ | $31.4 \pm 0.41$ | ✓ |
| LSTM CAT | AUDIO BASELINE | $46.38 \pm 0.44$ | $46.38 \pm 0.44$ | $46.38 \pm 0.44$ | $46.38 \pm 0.44$ | $46.38 \pm 0.44$ | - |
| LSTM CAT | VANILLA | $30.93 \pm 0.39$ | $36.39 \pm 0.39$ | $41.94 \pm 0.41$ | $46.81 \pm 0.43$ | $51.53 \pm 0.45$ | ✗ |
| LSTM CAT | CASCADE UTT | $33.67 \pm 0.4$ | $37.11 \pm 0.4$ | $40.88 \pm 0.42$ | $44.33 \pm 0.43$ | $46.55 \pm 0.44$ | ✓ |
| LSTM CAT | DROPOUT UTT | $33.41 \pm 0.4$ | $37.4 \pm 0.4$ | $41.68 \pm 0.42$ | $45.49 \pm 0.44$ | $48.13 \pm 0.44$ | ✗ |
| LSTM-CON CAT | CASCADE UTT | $29.37 \pm 0.37$ | $33.62 \pm 0.38$ | $38.41 \pm 0.4$ | $43.07 \pm 0.43$ | $46.55 \pm 0.44$ | ✓ |
| CONFORMER CM | AUDIO BASELINE | $31.79 \pm 0.42$ | $31.79 \pm 0.42$ | $31.79 \pm 0.42$ | $31.79 \pm 0.42$ | $31.79 \pm 0.42$ | - |
| CONFORMER CM | VANILLA | $34.91 \pm 0.43$ | $41.29 \pm 0.43$ | $48.96 \pm 0.45$ | $60.59 \pm 0.5$ | $98.34 \pm 0.05$ | ✗ |
| CONFORMER CM | CASCADE UTT | $27.89 \pm 0.37$ | $29.06 \pm 0.38$ | $30.2 \pm 0.39$ | $31.05 \pm 0.4$ | $31.56 \pm 0.4$ | ✓ |
| CONFORMER CM | DROPOUT UTT | $33.11 \pm 0.43$ | $48.17 \pm 0.47$ | $61.14 \pm 0.48$ | $74.43 \pm 0.46$ | $45.43 \pm 0.47$ | ✗ |

Table 27: Robustness to $\mathcal{T}_{rate}$ for 0db. Columns three to eight represent the amount of dropped video frames.

| Architecture | Method | 0 | $\frac{1}{128}$ | $\frac{1}{32}$ | $\frac{1}{8}$ | $\frac{1}{2}$ | 1 | Robust |
|---|---|---|---|---|---|---|---|---|
| Conformer CAT | Audio Baseline | $33.54 \pm 0.43$ | $33.54 \pm 0.43$ | $33.54 \pm 0.43$ | $33.54 \pm 0.43$ | $33.54 \pm 0.43$ | $33.54 \pm 0.43$ | - |
| Conformer CAT | Vanilla | $24.96 \pm 0.35$ | $24.94 \pm 0.35$ | $24.95 \pm 0.35$ | $25.08 \pm 0.35$ | $25.9 \pm 0.35$ | $35.51 \pm 0.43$ | ✗ |
| Conformer CAT | Vanilla (25L) | $24.75 \pm 0.35$ | $24.76 \pm 0.35$ | $24.79 \pm 0.35$ | $24.91 \pm 0.35$ | $26.78 \pm 0.37$ | $34.88 \pm 0.43$ | - |
| Conformer CAT | Cascade Utt | $26.12 \pm 0.36$ | $26.13 \pm 0.36$ | $26.25 \pm 0.36$ | $26.72 \pm 0.37$ | $28.73 \pm 0.39$ | $31.08 \pm 0.4$ | ✓ |
| Conformer CAT | Dropout Utt | $24.33 \pm 0.34$ | $24.35 \pm 0.34$ | $24.36 \pm 0.34$ | $24.49 \pm 0.34$ | $26.66 \pm 0.36$ | $31.67 \pm 0.4$ | ✓ |
| Conformer CAT | Cascade Frame | $28.04 \pm 0.38$ | $28.05 \pm 0.38$ | $28.08 \pm 0.38$ | $28.23 \pm 0.39$ | $30.14 \pm 0.42$ | $34.17 \pm 0.44$ | ✗ |
| Conformer CAT | Dropout Frame | $27.58 \pm 0.37$ | $27.47 \pm 0.37$ | $27.11 \pm 0.36$ | $26.51 \pm 0.36$ | $27.56 \pm 0.38$ | $32.54 \pm 0.41$ | ✗ |
| Conformer CAT | AV Dropout Utt | $23.64 \pm 0.33$ | $23.62 \pm 0.33$ | $23.63 \pm 0.33$ | $23.8 \pm 0.33$ | $25.36 \pm 0.35$ | $33.55 \pm 0.41$ | ✓ |
| Conformer CAT | Two-Pass | $30.06 \pm 0.42$ | $30.05 \pm 0.42$ | $30.11 \pm 0.42$ | $30.47 \pm 0.43$ | $31.91 \pm 0.43$ | $33.54 \pm 0.43$ | ✓ |
| Con-LSTM CAT | Cascade Utt | $27.9 \pm 0.39$ | $27.96 \pm 0.39$ | $28.11 \pm 0.39$ | $28.84 \pm 0.39$ | $30.7 \pm 0.41$ | $31.4 \pm 0.41$ | ✓ |
| LSTM CAT | Audio Baseline | $46.38 \pm 0.44$ | $46.38 \pm 0.44$ | $46.38 \pm 0.44$ | $46.38 \pm 0.44$ | $46.38 \pm 0.44$ | $46.38 \pm 0.44$ | - |
| LSTM CAT | Vanilla | $30.93 \pm 0.39$ | $30.94 \pm 0.39$ | $31.16 \pm 0.39$ | $32.06 \pm 0.4$ | $37.19 \pm 0.44$ | $51.53 \pm 0.45$ | ✗ |
| LSTM CAT | Cascade Utt | $33.67 \pm 0.4$ | $33.71 \pm 0.4$ | $34.15 \pm 0.4$ | $35.9 \pm 0.41$ | $42.15 \pm 0.44$ | $46.55 \pm 0.44$ | ✓ |
| LSTM CAT | Dropout Utt | $33.41 \pm 0.4$ | $33.47 \pm 0.4$ | $33.78 \pm 0.4$ | $35.11 \pm 0.41$ | $41.79 \pm 0.44$ | $48.13 \pm 0.44$ | ✗ |
| LSTM-Con CAT | Cascade Utt | $29.37 \pm 0.37$ | $29.42 \pm 0.37$ | $29.72 \pm 0.38$ | $30.96 \pm 0.38$ | $37.42 \pm 0.42$ | $46.55 \pm 0.44$ | ✓ |
| Conformer CM | Audio Baseline | $31.79 \pm 0.42$ | $31.79 \pm 0.42$ | $31.79 \pm 0.42$ | $31.79 \pm 0.42$ | $31.79 \pm 0.42$ | $31.79 \pm 0.42$ | - |
| Conformer CM | Vanilla | $34.91 \pm 0.43$ | $35.03 \pm 0.43$ | $35.68 \pm 0.44$ | $38.48 \pm 0.45$ | $51.29 \pm 0.51$ | $98.34 \pm 0.05$ | ✗ |
| Conformer CM | Cascade Utt | $27.89 \pm 0.37$ | $27.9 \pm 0.37$ | $27.98 \pm 0.38$ | $28.48 \pm 0.38$ | $30.99 \pm 0.4$ | $31.56 \pm 0.4$ | ✓ |
| Conformer CM | Dropout Utt | $33.11 \pm 0.43$ | $33.46 \pm 0.42$ | $35.82 \pm 0.44$ | $46.46 \pm 0.51$ | $68.24 \pm 0.5$ | $45.43 \pm 0.47$ | ✗ |

## F Experiments on TED LRS3

We present additional results on the TED LRS3 dataset for just the Conformer CAT and LSTM CAT Cascade Utt models. The architectural configurations are the same as the ones detailed in Section 5, with the exception of differences in optimization. The models are trained with a constant learning rate of 0.0001 at batch size 512 for 35k steps. Like Afouras et al. (2018) and Ma et al. (2021b), we initialize our models from pre-trained weights. This was done using the training data described in Section 5. Like in the main paper, we augment the TED LRS3 test set with artificially added babble noise from the NoiseX corpus (Varga & Steeneken, 1993) at different SNR levels: clean, 20db, 10db, 0db. Like in the main paper, all the Cascade Utt models were found to be robust on all the test suites. To our knowledge, the 0.92 WER we are reporting on the clean TED LRS3 test set by the Conformer CAT Cascade Utt model is also the best number that has been reported on this dataset.

Table 28: Robustness to $\mathcal{T}_{berUtt}$ for Conformer and LSTM Cascade Utt models. Columns three to seven represent the amount of dropped video frames.

| ARCHITECTURE | NOISE | 0.0 | 0.25 | 0.5 | 0.75 | 1.0 | AO BASELINE | ROBUST |
|---|---|---|---|---|---|---|---|---|
| CONFORMER CAT | CLEAN | $0.92 \pm 0.26$ | $0.9 \pm 0.25$ | $0.99 \pm 0.27$ | $0.95 \pm 0.26$ | $1.01 \pm 0.27$ | $0.99 \pm 0.26$ | ✓ |
| LSTM CAT | CLEAN | $1.8 \pm 0.35$ | $2.69 \pm 0.47$ | $3.32 \pm 0.49$ | $4.16 \pm 0.58$ | $4.9 \pm 0.61$ | $4.82 \pm 0.61$ | ✓ |
| CONFORMER CAT | 20DB | $1.02 \pm 0.26$ | $1.0 \pm 0.26$ | $1.04 \pm 0.26$ | $1.03 \pm 0.26$ | $0.99 \pm 0.26$ | $1.11 \pm 0.29$ | ✓ |
| LSTM CAT | 20DB | $1.8 \pm 0.35$ | $2.64 \pm 0.45$ | $3.26 \pm 0.51$ | $4.16 \pm 0.57$ | $5.25 \pm 0.65$ | $5.16 \pm 0.64$ | ✓ |
| CONFORMER CAT | 10DB | $1.1 \pm 0.27$ | $1.13 \pm 0.27$ | $1.16 \pm 0.28$ | $1.11 \pm 0.27$ | $1.19 \pm 0.28$ | $1.41 \pm 0.31$ | ✓ |
| LSTM CAT | 10DB | $2.11 \pm 0.38$ | $3.31 \pm 0.52$ | $4.62 \pm 0.6$ | $5.57 \pm 0.68$ | $6.79 \pm 0.74$ | $6.6 \pm 0.73$ | ✓ |
| CONFORMER CAT | 0DB | $1.9 \pm 0.36$ | $2.53 \pm 0.43$ | $3.22 \pm 0.5$ | $3.91 \pm 0.55$ | $4.41 \pm 0.58$ | $4.41 \pm 0.57$ | ✓ |
| LSTM CAT | 0DB | $4.32 \pm 0.55$ | $8.95 \pm 0.99$ | $14.41 \pm 1.31$ | $20.79 \pm 1.44$ | $25.04 \pm 1.5$ | $24.54 \pm 1.45$ | ✓ |

Table 29: Robustness to $\mathcal{T}_{berFrame}$ for Conformer and LSTM Cascade Utt models. Columns three to seven represent the amount of dropped video frames.

| ARCHITECTURE | NOISE | 0.0 | 0.25 | 0.5 | 0.75 | 1.0 | AO BASELINE | ROBUST |
|---|---|---|---|---|---|---|---|---|
| CONFORMER CAT | CLEAN | $0.92 \pm 0.26$ | $1.0 \pm 0.26$ | $1.0 \pm 0.27$ | $0.99 \pm 0.27$ | $1.01 \pm 0.27$ | $0.99 \pm 0.26$ | ✓ |
| LSTM CAT | CLEAN | $1.8 \pm 0.35$ | $2.07 \pm 0.36$ | $2.85 \pm 0.44$ | $3.69 \pm 0.49$ | $4.9 \pm 0.61$ | $4.82 \pm 0.61$ | ✓ |
| CONFORMER CAT | 20DB | $1.02 \pm 0.26$ | $0.95 \pm 0.25$ | $0.97 \pm 0.26$ | $1.03 \pm 0.27$ | $0.99 \pm 0.26$ | $1.11 \pm 0.29$ | ✓ |
| LSTM CAT | 20DB | $1.8 \pm 0.35$ | $2.25 \pm 0.39$ | $2.92 \pm 0.44$ | $3.95 \pm 0.52$ | $5.25 \pm 0.65$ | $5.16 \pm 0.64$ | ✓ |
| CONFORMER CAT | 10DB | $1.1 \pm 0.27$ | $1.12 \pm 0.28$ | $1.13 \pm 0.27$ | $1.27 \pm 0.29$ | $1.19 \pm 0.28$ | $1.41 \pm 0.31$ | ✓ |
| LSTM CAT | 10DB | $2.11 \pm 0.38$ | $2.93 \pm 0.45$ | $3.66 \pm 0.5$ | $5.02 \pm 0.64$ | $6.79 \pm 0.74$ | $6.6 \pm 0.73$ | ✓ |
| CONFORMER CAT | 0DB | $1.9 \pm 0.36$ | $2.32 \pm 0.4$ | $3.38 \pm 0.51$ | $4.0 \pm 0.55$ | $4.41 \pm 0.58$ | $4.41 \pm 0.57$ | ✓ |
| LSTM CAT | 0DB | $4.32 \pm 0.55$ | $8.26 \pm 0.82$ | $12.93 \pm 1.03$ | $19.69 \pm 1.3$ | $25.04 \pm 1.5$ | $24.54 \pm 1.45$ | ✓ |

Table 30: Robustness to $\mathcal{T}_{start}$ for Conformer and LSTM Cascade Utt models. Columns three to seven represent the amount of dropped video frames.

| ARCHITECTURE | NOISE | 0.0 | 0.25 | 0.5 | 0.75 | 1.0 | AO BASELINE | ROBUST |
|---|---|---|---|---|---|---|---|---|
| CONFORMER CAT | CLEAN | $0.92 \pm 0.26$ | $0.93 \pm 0.25$ | $1.05 \pm 0.27$ | $1.06 \pm 0.27$ | $1.01 \pm 0.27$ | $0.99 \pm 0.26$ | ✓ |
| LSTM CAT | CLEAN | $1.8 \pm 0.35$ | $3.04 \pm 0.46$ | $4.03 \pm 0.53$ | $4.6 \pm 0.58$ | $4.9 \pm 0.61$ | $4.82 \pm 0.61$ | ✓ |
| CONFORMER CAT | 20DB | $1.02 \pm 0.26$ | $0.99 \pm 0.26$ | $1.04 \pm 0.26$ | $1.09 \pm 0.27$ | $0.99 \pm 0.26$ | $1.11 \pm 0.29$ | ✓ |
| LSTM CAT | 20DB | $1.8 \pm 0.35$ | $3.1 \pm 0.48$ | $4.13 \pm 0.56$ | $5.01 \pm 0.62$ | $5.25 \pm 0.65$ | $5.16 \pm 0.64$ | ✓ |
| CONFORMER CAT | 10DB | $1.1 \pm 0.27$ | $1.08 \pm 0.27$ | $1.15 \pm 0.28$ | $1.3 \pm 0.29$ | $1.19 \pm 0.28$ | $1.41 \pm 0.31$ | ✓ |
| LSTM CAT | 10DB | $2.11 \pm 0.38$ | $3.61 \pm 0.5$ | $4.87 \pm 0.6$ | $6.31 \pm 0.7$ | $6.79 \pm 0.74$ | $6.6 \pm 0.73$ | ✓ |
| CONFORMER CAT | 0DB | $1.9 \pm 0.36$ | $2.62 \pm 0.43$ | $3.32 \pm 0.5$ | $3.88 \pm 0.56$ | $4.41 \pm 0.58$ | $4.41 \pm 0.57$ | ✓ |
| LSTM CAT | 0DB | $4.32 \pm 0.55$ | $9.76 \pm 0.88$ | $16.05 \pm 1.19$ | $22.26 \pm 1.44$ | $25.04 \pm 1.5$ | $24.54 \pm 1.45$ | ✓ |

Table 31: Robustness to $\mathcal{T}_{mid}$ for Conformer and LSTM Cascade Utt models. Columns three to seven represent the amount of dropped video frames.

| Architecture | Noise | 0.0 | 0.25 | 0.5 | 0.75 | 1.0 | AO Baseline | Robust |
|---|---|---|---|---|---|---|---|---|
| Conformer CAT | Clean | $0.92 \pm 0.26$ | $1.04 \pm 0.27$ | $1.09 \pm 0.28$ | $1.04 \pm 0.27$ | $1.01 \pm 0.27$ | $0.99 \pm 0.26$ | ✓ |
| LSTM CAT | Clean | $1.8 \pm 0.35$ | $2.63 \pm 0.42$ | $3.46 \pm 0.49$ | $4.37 \pm 0.57$ | $4.9 \pm 0.61$ | $4.82 \pm 0.61$ | ✓ |
| Conformer CAT | 20db | $1.02 \pm 0.26$ | $1.02 \pm 0.26$ | $1.08 \pm 0.27$ | $1.01 \pm 0.26$ | $0.99 \pm 0.26$ | $1.11 \pm 0.29$ | ✓ |
| LSTM CAT | 20db | $1.8 \pm 0.35$ | $2.52 \pm 0.42$ | $3.59 \pm 0.5$ | $4.58 \pm 0.59$ | $5.25 \pm 0.65$ | $5.16 \pm 0.64$ | ✓ |
| Conformer CAT | 10db | $1.1 \pm 0.27$ | $1.09 \pm 0.27$ | $1.28 \pm 0.29$ | $1.26 \pm 0.28$ | $1.19 \pm 0.28$ | $1.41 \pm 0.31$ | ✓ |
| LSTM CAT | 10db | $2.11 \pm 0.38$ | $3.28 \pm 0.48$ | $4.8 \pm 0.6$ | $6.0 \pm 0.7$ | $6.79 \pm 0.74$ | $6.6 \pm 0.73$ | ✓ |
| Conformer CAT | 0db | $1.9 \pm 0.36$ | $2.76 \pm 0.45$ | $3.26 \pm 0.49$ | $4.2 \pm 0.58$ | $4.41 \pm 0.58$ | $4.41 \pm 0.57$ | ✓ |
| LSTM CAT | 0db | $4.32 \pm 0.55$ | $10.57 \pm 0.94$ | $16.98 \pm 1.23$ | $22.22 \pm 1.39$ | $25.04 \pm 1.5$ | $24.54 \pm 1.45$ | ✓ |

Table 32: Robustness to $\mathcal{T}_{end}$ for Conformer and LSTM Cascade Utt models. Columns three to seven represent the amount of dropped video frames.

| Architecture | Noise | 0.0 | 0.25 | 0.5 | 0.75 | 1.0 | AO Baseline | Robust |
|---|---|---|---|---|---|---|---|---|
| Conformer CAT | Clean | $0.92 \pm 0.26$ | $1.02 \pm 0.27$ | $1.04 \pm 0.28$ | $1.06 \pm 0.28$ | $1.01 \pm 0.27$ | $0.99 \pm 0.26$ | ✓ |
| LSTM CAT | Clean | $1.8 \pm 0.35$ | $1.92 \pm 0.36$ | $2.78 \pm 0.43$ | $3.74 \pm 0.52$ | $4.9 \pm 0.61$ | $4.82 \pm 0.61$ | ✓ |
| Conformer CAT | 20db | $1.02 \pm 0.26$ | $0.95 \pm 0.26$ | $1.01 \pm 0.27$ | $1.04 \pm 0.27$ | $0.99 \pm 0.26$ | $1.11 \pm 0.29$ | ✓ |
| LSTM CAT | 20db | $1.8 \pm 0.35$ | $2.0 \pm 0.37$ | $2.79 \pm 0.45$ | $3.96 \pm 0.54$ | $5.25 \pm 0.65$ | $5.16 \pm 0.64$ | ✓ |
| Conformer CAT | 10db | $1.1 \pm 0.27$ | $1.09 \pm 0.27$ | $1.11 \pm 0.27$ | $1.2 \pm 0.28$ | $1.19 \pm 0.28$ | $1.41 \pm 0.31$ | ✓ |
| LSTM CAT | 10db | $2.11 \pm 0.38$ | $2.33 \pm 0.38$ | $3.65 \pm 0.52$ | $5.31 \pm 0.66$ | $6.79 \pm 0.74$ | $6.6 \pm 0.73$ | ✓ |
| Conformer CAT | 0db | $1.9 \pm 0.36$ | $2.59 \pm 0.42$ | $3.39 \pm 0.49$ | $3.95 \pm 0.54$ | $4.41 \pm 0.58$ | $4.41 \pm 0.57$ | ✓ |
| LSTM CAT | 0db | $4.32 \pm 0.55$ | $7.76 \pm 0.72$ | $13.45 \pm 1.01$ | $20.08 \pm 1.31$ | $25.04 \pm 1.5$ | $24.54 \pm 1.45$ | ✓ |

Table 33: Robustness to $\mathcal{T}_{rate}$ for Conformer and LSTM Cascade Utt models. Columns three to eight represent the amount of dropped video frames.

| Architecture | Noise | 0 | $\frac{1}{128}$ | $\frac{1}{32}$ | $\frac{1}{8}$ | $\frac{1}{2}$ | 1 | AO Baseline | Robust |
|---|---|---|---|---|---|---|---|---|---|
| Conformer CAT | Clean | $0.92 \pm 0.26$ | $0.92 \pm 0.26$ | $0.94 \pm 0.26$ | $0.94 \pm 0.26$ | $0.98 \pm 0.26$ | $1.01 \pm 0.27$ | $0.99 \pm 0.26$ | ✓ |
| LSTM CAT | Clean | $1.8 \pm 0.35$ | $1.77 \pm 0.35$ | $1.76 \pm 0.35$ | $1.94 \pm 0.36$ | $2.47 \pm 0.39$ | $4.9 \pm 0.61$ | $4.82 \pm 0.61$ | ✓ |
| Conformer CAT | 20db | $1.02 \pm 0.26$ | $1.02 \pm 0.26$ | $1.0 \pm 0.26$ | $1.02 \pm 0.26$ | $1.01 \pm 0.27$ | $0.99 \pm 0.26$ | $1.11 \pm 0.29$ | ✓ |
| LSTM CAT | 20db | $1.8 \pm 0.35$ | $1.78 \pm 0.35$ | $1.76 \pm 0.35$ | $1.94 \pm 0.36$ | $2.59 \pm 0.41$ | $5.25 \pm 0.65$ | $5.16 \pm 0.64$ | ✓ |
| Conformer CAT | 10db | $1.1 \pm 0.27$ | $1.1 \pm 0.27$ | $1.11 \pm 0.27$ | $1.11 \pm 0.27$ | $1.07 \pm 0.27$ | $1.08 \pm 0.26$ | $1.41 \pm 0.31$ | ✓ |
| LSTM CAT | 10db | $2.11 \pm 0.38$ | $2.12 \pm 0.38$ | $2.15 \pm 0.4$ | $2.33 \pm 0.4$ | $3.1 \pm 0.45$ | $6.79 \pm 0.74$ | $6.6 \pm 0.73$ | ✓ |
| Conformer CAT | 0db | $1.9 \pm 0.36$ | $1.9 \pm 0.36$ | $1.95 \pm 0.37$ | $2.35 \pm 0.41$ | $2.91 \pm 0.46$ | $4.41 \pm 0.58$ | $4.41 \pm 0.57$ | ✓ |
| LSTM CAT | 0db | $4.32 \pm 0.55$ | $4.33 \pm 0.55$ | $4.77 \pm 0.61$ | $5.95 \pm 0.67$ | $11.93 \pm 1.02$ | $25.04 \pm 1.5$ | $24.54 \pm 1.45$ | ✓ |

