# OpenReview forum: "On Robustness to Missing Video for Audiovisual Speech Recognition"
_TMLR — Accepted by TMLR_

### Review · Reviewer_6k5k · 2022-06-14

**Summary Of Contributions:**

The paper deals with audiovisual speech recognition. It shows that current techniques while capable of leveraging visual data to improve speech recognition, they are often not robust to missing video data, ie when models trained on audio-visual data are tested on data with missing video information, they can underperform audio-only (unimodal) models. The paper seeks to formalize how to evaluate robustness to missing video information, and makes three main contributions.

1) A sensible robustness evaluation framework, spanning a variety of realistic conditions under which missing video could arise. Under this framework, a model is robust if its performance improves as it is given more video information (ie less missing frames).

2) An evaluation of previously proposed robustness techniques (mostly focused on modality dropout). This evaluation shows that robustness to missing video information is not always guaranteed with modality dropout. This is an interesting observation.

3) A cascaded audio-visual architecture that fuses audio-only predictions with audio-visual predictions. At inference time, the cascaded model can be only partially used, if only audio inputs are available, or fully used if both audio-visual information is available. By training the model with both unimodal and multimodal inputs, the cascade is shown to be robust to missing video in a variety of noise conditions and using a variety of backbone components.

**Broader Impact Concerns:**

A broader impact statement should be added, acknowledging that by the nature of the dataset collection process, the performance of the model with under-represented groups is not being measured and that optimizing for the specific benchmark metrics proposed here may have unknown effects (either positive or negative) on the performance of models within such groups.

**Requested Changes:**

Critical changes are described in MW1, W1 and W3 (especially MW1). This however entails significant changes to the whole manuscript, and I’d like to review it again before publication. If these changes are done properly, I believe this could be a strong paper. Properly evaluating robustness is a contribution that would be of interest to a subset of TMLR readers.

**Strengths And Weaknesses:**

Strengths:

S1) The proposed evaluation framework is intuitive and sensible. However, please see MW1.

S2) Although the cascaded architecture is not novel, its application to audiovisual speech recognition is sound and intuitive.

S3) The paper provides an extensive comparison of several methods under several conditions of missing video and noise levels. The cascaded architecture does produce more robust results (as initially claimed). While I find this to be a strength, I think the definition of robustness needs to be expanded to warrant publication.

S4) The paper is well structured and easy to read. The mathematical framework is clear and cogent.

Major Weakness:

MW1) In my view, the major weakness of this framework is that it only addresses robustness to known missing video information, ie, when frames are dropped due to low bandwidth or when a user turns off their camera. However, in all these cases, we know that video frames are missing, which means that we could trivially swap to an audio-only model, instead of relying on an audio-visual model with zeroed-out video input. This trivial approach would be robust under this paper’s definition of robustness.

Perhaps a more realistic and challenging definition of robustness would be robustness to unknown missing video. This would occur when for example the crop of the speaker's lips fails, or if the speaker looks away or their mouth becomes occluded. In these cases, we might have video information, but it is not relevant to the task at hand. Since we may not know which frames are relevant, we couldn’t just swap to an audio-only model. Another potential source of corruption that models should also be robust would be robustness to common corruptions like jitter and blur (as acknowledged by the authors in the limitation section).

Given that the source of corruption studied in this work (known missing frames) could be easily addressed with an audio-only model, it seems to me that expanding the definition of robustness is necessary to warrant publication.

Weaknesses:

W1) Lacking information about used backbone models. I understand the paper does not propose a new backbone architecture. Instead, it uses LSTMs and Conformers proposed in previous work. However, more information is necessary to fully describe the models used in the experiments. For example, the language model and joiner are not specified.

W2) According to Section 5 “Training data”, the training dataset is collected by selecting videos in which user uploaded transcripts match a production quality ASR system. It seems to me that the (already existing) ASR system is the upper bound to any model trained on this data. Are these systems simply relearning the predictions of the “production quality” model? In which case, what’s the point? Having said that, I understand that this may be common practice in this area, and this paper does not seek to address this problem.

W3) Although acknowledge as a limitation in the paper, I believe that the binary “robust/not robust” metric does not provide enough information about the robustness of a model. After reading the paper, I began to think of robustness as a constraint, which can be violated with different levels of severity. However, the binary “robust/not robust” metric does not capture this. For example, we may be willing to forego a bit of robustness in order to obtain higher accuracy. Since establishing such a metric could be accomplished without any extra experiments or evaluation settings, I believe it should have been included in the manuscript.

Other comments:

O1) While I do agree with the requirement for robustness to missing video information, it is worth noting to point out that, one of the most reliable speech recognition systems (the human brain) is not fully robust. Take for example the widely known McGurk effect.

O2) Section 4.3 corollary 4.7. While clearly true, it is unclear to me why this corollary is mentioned. I didn’t see any applications or important conclusions drawn from this corollary.

O3) It would be worth pointing out in related work that robustness in multi-modal learning has been studied beyond audio-visual speech recognition, eg [A] Morgado et al. Robust audio-visual instance discrimination. CVPR. 2021. or [B] Han, et al. Noise-Tolerant Learning for Audio-Visual Action Recognition. arXiv:2205.07611, 2022.

---

> ### Author Response · Authors · 2022-06-15
> **Response**
>
> Thank you so much for the careful reading of our manuscript and the thoughtful feedback provided. We will do our best to address your concerns below and have revised the manuscript accordingly.
>
> > MW1
>
> 1) In the AV ASR literature ([1], [2]), the visual information is usually extracted with a face tracker that draws regions of interest in the video around the lips, which then passes that information to the AV ASR model. Thus if the video frame is missing, this would generally be known to the model. We have stated this in Section 5: Audiovisual Features. To emphasize this, we have re-arranged this section so it appears before Section 5: Training Data and elaborated on it. Of course, the assumption in our paper and in the AV ASR literature is that the AV ASR model is only focused on transcribing the speech, and the extraction of the mouth track can be outsourced. In practice, we agree with the reviewer that problems like occluded lips are robustness challenges, since the face tracker is imperfect and can pass occluded lips instead of a blank frame.
> 2) We agree with the reviewer that it is important to make it clear whether the model actually needs to make use of this knowledge that the video frame is missing in order to perform inference. The cascade methods require this knowledge, but the dropout methods do not. We have updated Section 7  to make this clear. It is conceivable that a fully end-to-end system where the face tracker is integrated and trained as part of the AV ASR model might run into robustness challenges like the reviewer mentioned: the speaker lips become occluded but the model sees the occlusion instead of a blank frame.
> 3) Even though we know that the video frames are missing, this does not mean that it is trivial to perform the switching to an audio-only model. When all the frames are missing, the naive switching method would be to have an ensemble of two models: an audio model and a video model. This is more computationally demanding than having a cascaded model in terms of compute, memory, and disk. When only some of the frames are missing, the naive ensemble would also not be able to fully take advantage of both modalities, since the audio model is operating independently of the video model. Furthermore, the cascaded model is fully compatible with the streaming scenario but the naive ensemble may not be.
>
> > W1
>
> We specified the language model and the joiner in Section 5: Architectural Configurations. We have updated it with additional details including the embedding dimension and the beam width. If there is any other specific detail that the reviewer would like us to include, please let us know.
>
> > W2
>
> We agree with the reviewer that this is common practice and outside of the primary focus of this paper. However, we wish to point out that it is not true that the pre-existing ASR system forms the upper bound.
> 1) The user uploaded transcript is still used as the ground truth. By only keeping labeled data where they match a pre-existing ASR system, we get high confidence in our labels, but nonetheless these are still human labels. We do not keep the machine predicted labels that do not agree with the human labels.
> 2) State-of-the-art methods in ASR like [3] essentially train an ASR system on noisy labels, use the trained model to re-label the data, and rinse-and-repeat. The final model obtained is far superior to the initial model, so the initial model cannot be considered an upper bound.
>
> > W3
>
> The robustness metric is binary with respect to a given test suite. In the Appendix, we have extensively documented which model is robust under which test suite. If the reviewer would like to see a numerical score, perhaps we can sum up the number of test suites under which a model is robust.
>
> > O2
>
> The corollary shows that train-time robustness and test-time robustness together spans the entire space of training and testing conditions. For example, if we know that an AV model outperforms an AO model at 50% video frames, we can infer from the corollary that it also outperforms an AO model at 25% video frames.
>
> > O3
>
> Thanks. We have cited numerous examples of robustness in multi-modal learning that are not AV ASR in Section 1. We have included these two additional references in Section 1.
>
> > Broader impact statement.
>
> We have added it in the Appendix.
>
> Thank you again for the reviewer's feedback. We believe that while narrowly scoped, our contribution is of great interest to TMLR readers both in the robustness framework proposed and the empirical results comparing different robustness methods in AV ASR.
>
> [1]: Deep Audio-Visual Speech Recognition. Afouras et al.
>
> [2]: Recurrent Neural Network Transducer for Audio-Visual Speech Recognition. Makino et al.
>
> [3]: Improved Noisy Student Training for Automatic Speech Recognition. Park et al.

---

### Review · Reviewer_EdkC · 2022-06-19

**Summary Of Contributions:**

The paper focuses on robustness of speech recognition systems, in particular audiovisual speech recognition systems when visual signals may or may not be available.

The paper argues that the current approaches for robustness evaluations do not provide a complete picture and a more formal approach is necessary. The paper presents a formal definition of robustness to missing video through order theory and then defines common case test conditions (of missing video) under which different models are evaluated.

To enhance the robustness itself, the paper argues for a Cascade framework in which an audio-only (AO) model is cascaded by audiovisual model and the framework ensures that in case of missing video - the system is at least as good as an audio-only model.


**Broader Impact Concerns:**

I do not have broader impact concerns.


**Requested Changes:**

I would propose that authors address the weaknesses pointed out in the previous section.

For 1 & 2. It would be best if these are rectified through additional experiments. I think adding them would be important in arguing for acceptance of the paper.

For 3 to 9. I would urge the authors to respond to these comments and make appropriate changes in the text to address them when necessary.


**Strengths And Weaknesses:**

Strengths
1. Multimodal learning for a variety of problems, including speech recognition, are gaining significance, and understanding robustness in terms of availability of different modalities is an important problem. The lack of consistent evaluation strategies, as pointed out in the paper, does make it difficult to understand the robustness of different methods. The paper is a step in the right direction - the robustness and evaluation strategies should be well defined. The paper is also well-written and presented.

2. The robustness framework makes sense and does rectify the problem of lack of robustness definition and test conditions. The test conditions presented cover most of the common cases. This should make the paper a good read on the topic of AV speech recognition.

3. The empirical results and analyses are also good.  I do have some concerns around them but they can be rectified.


Weaknesses

1. One primary concern is regarding the dataset used in experiments. The paper uses only one dataset and that too a private training and test dataset sourced from YouTube which will not be available for the research community in general. Other AV speech recognition dataset are available and analyses of results on those would have helped paint a better picture and improve reproducibility. Given that the paper is aimed to study “robustness” - studying multiple datasets will go a long way in establishing the claims.

2. The robustness framework mentions T_all test suite but this is not really empirically studied. In general any composition of the 6 different test conditions are realistically possible. From the definition of the robustness framework also, it is not clear to me what would it mean to think of robustness when 2 or more of these test conditions are simultaneously present. It would also be good to see some empirical results on this.

3. In some cases, the CASCADE UTT ends up performing better than AO, even when video is completely missing. For example, in Table 2, it is better than Audio baseline by almost 2%. Given the way Cascade is defined in  Eq 5, why should this be possible ? In section 3.2 authors mention that Cascade “explicitly disentangles” AV from AO representations.

4. For the Cascade framework, why is CASCADE FRAME not robust ? Some discussion on it would be helpful .

5. Not sure if its only me but the terms ``Train-Time Robustness” and “Test-time robustness” are a bit confusing.

6. Section 4.4 presents different test conditions. Discussing the connection between these and the robustness definition would be really helpful for readers. What does a given test condition with different rates mean in terms of robustness for a given model ?

7. In Section “Cascade Utt achieves model capacity” authors mention that one pass uses half the resources of two pass training. I am not clear about this. In the two passes optimization,  AM is trained first and then frozen, and then AVM is trained. WHy will it take twice the resources ?

8. Are the results in Tables for T_end miss-labeled. The columns should ready  1.0, 0.75, 0.5, 0.25, 0 ??

9. Several of the references in section 1, the paragraph about prior works of missing modality, are not really about missing modality. They seem to be just multimodal learning frameworks, examples Nagrani 2021, Gogate 2021 etc.

---

> ### Author Response · Authors · 2022-06-28
> **Response**
>
> Thank you so much for your constructive feedback and careful reading of our paper.
>
> We have conducted additional experiments on the publicly available TED LRS3 dataset and uploaded an updated version of the manuscript. Below, we will address the various questions raised in the Weaknesses section of your review.
>
> 1) As requested, we have documented additional experiments in Appendix Section E. These experiments were done on the TED LRS3 dataset, using artificially added noise as in the main paper. We verified that our analysis in the main paper holds for TED as well: the Conformer and LSTM Cascade Utt models were found to be robust under all test suites. Incidentally, the numbers we report for the Conformer model under the clean condition when all the video frames are present are also state-of-the-art on this dataset: 0.9 WER versus 2.3 WER in [1].
>
> 2) The benefits of a framework that uses partial order is that not every test condition needs to be comparable. Excepting the boundary conditions where all the video frames are present or none of them are, none of the conditions in our six test suites are comparable across test suites. So T_all is really a trivial composition that sums up robustness across the six test suites, rather than an empirical analysis of the phenomenon composed from two or more test suites. It would be interesting to have non-trivial compositions of different test suites, but it is worth noting that this will easily lead to an exponential number of test suites, which is intractable.
>
> 3) In general, there can be benefits from training on AV even when we are only interested in testing on AO. For example, this was done in [2], where the authors argued that this led to better cluster assignments. While the cascade model explicitly disentangles the AV representations from the AO ones, there could still be optimization reasons why we observe that training on AV helps testing on AO. For example, we found that the Cascade Utt model outperformed the Two Pass model, even though they have the same architecture.
>
> 4) Thanks for the comment, please see the discussion in Section 6: Frame Methods Underperform Utterance Methods. While this observation appears counter-intuitive, consider that even if the probability of dropping each of 512 frames is high (90%), the model is nevertheless rarely exposed to an empty video at training time (0.9512 = 10−24). This will thus cause it to perform badly in test conditions when the video is completely absent. Indeed, we see that Cascade Frame underperforms Cascade Utt, and Dropout Frame underperforms Dropout Utt in terms of robustness, primarily from bad performance in the condition when the video is completely absent.
>
> 5) Thanks for the feedback. Do you have a suggestion for alternative terms?
>
> 6) The number of different rates specifies the granularity of the test suite. The more rates there are, the more granular the test suite is. At the least granular level, we might only be interested in whether the video is present or absent, rather than the percentage of frames dropped.
>
> 7) It takes twice the resources because we train the Two-Pass model twice with the exact same optimization setup in both instances. It's  of course possible to spend half the training time for each pass to equalize the amount of resources used.
>
> 8) Thanks for the feedback. Can you point out specifically which table needs fixing?
>
> 9) We included those references because they tested their models under differing conditions of the presence of various modalities. In many cases, the implicit or explicit argument that is made is that the multi-modal models perform better than the single modality models, which dovetails into multi-modal robustness. We can remove the references, or perhaps re-phrase the paragraph to improve its clarity.
>
> Thanks again for your extensive feedback. Please let us know if this addresses your concerns, and if any other changes are needed.
>
> [1] End-to-End Audio-Visual Speech Recognition with Conformers. Ma et al.
>
> [2] Learning Audio-Visual Speech Representation by Masked Multimodal Cluster Prediction. Shi et al.

---

### Review · Reviewer_gReF · 2022-07-05

**Summary Of Contributions:**

This paper presents a novel methodology to evaluate robustness of audio-visual speech recognition models to missing videos frames. A framework is introduced, where a model is considered robust when performance without video is not worse than performance of an audio-only model. The paper provides empirical results for well known architectures and approaches, and shows that the Cascade approach, which has an audio-only path used when the video is missing, is the most robust approach.

**Broader Impact Concerns:**

The Broader Impact Statement is present and sufficiently addresses the ethical implications.

**Requested Changes:**

- Section 2:
    - Change title.
    - Define the terms used in Equation (1).
    - Please provide the definition of the RNN-T loss.
- Clarify the parameters of the test suites.
- Review the structure of the related work.
- Clarify the methods used in Section 6.
- Expand the conclusion.


**Strengths And Weaknesses:**

Strengths:
- The proposed framework is novel as far as I can tell, and could be very useful for the community.
- The cascade approach seems also novel and is a significant finding, as it is the most robust approach.
- Overall it's a very good paper, it is a significant contribution to the field of audio-visual speech recognition.

Weaknesses:
- My main concern is that the results and the framework are not reproducible: (1) the training data seems to be a private dataset as no reference or link are provided and (2) the parameters of the test suits are not clearly documented. Section 4.4 gives examples of parameters, but it's not clear that the examples are the actual values used in the experiments. It's a big issue for me, as one of the goals of such a benchmark is to provide a way for the entire community to evaluate and compare their model, which is not possible currently. Can the authors clarify the experimental setup and comments on their plan to share their data?
- The paper is generally well written it has some issues in terms of structure and clarity:
    - in Section 1.1, after the end of the c) paragraph, it's not clear if the following two paragraphs are still part of c) or part of 1.1.
     - Section 2 is a bit confusing, the title suggests that the section is going to be about the task, thus will describe the related work, but it's really the description of the proposed model. Why not call it something like "Methodology" or "proposed approach" ? It is also really short and could benefit from more information (see "Requested Changes").
     - The related work is scattered in different parts of the paper: in Section 1, in 3.1 and 4.2. I think the paper could benefit from having a clearer separation between related work and novel contributions by having a dedicated "Related Work" Section.
     - The methods evaluated in Section 6 (vanilla, cascade utt, etc) are hard to find, as they are briefly described at the end of Section 3.2. I think it would be better to describe them more clearly, for instance the same way the architectures are described in Section 5.
     - The conclusion is very short, a summary of the findings in Section 6 would be beneficial.

---

> ### Author Response · Authors · 2022-07-06
> **Response**
>
> Thank you for your helpful comments. We have made the following changes to improve the clarity of our manuscript as requested:
>
> In Section 1.1, after the end of the c) paragraph, it's not clear if the following two paragraphs are still part of c) or part of 1.1.
> > We added a new sub-header Section 1.2 to distinguish it from Section 1.1.
>
> Section 2 is a bit confusing
> > We have updated the title of Section 2, and provided a more extensive description of the RNN-T loss. In general, losses used for ASR like RNN-T and CTC can be pretty tricky, which makes it difficult to provide a self-contained explanation. Thus, we refer the reader to the relevant literature so that they can familiarize themselves with it.
>
> Clarify the parameters of the test suites.
> > We have clarified the parameters of the test suites in Section 4.4 by changing "For example" to "In our experiments" to make it clear that those parameters are what we are using in our experiments.
>
> Review the structure of the related work.
> > We cover prior work in two separate places 3.1 and 4.2 because they refer to conceptually distinct prior work. 3.1 talks about prior work for techniques that make models more robust, while 4.2 talks about prior work for evaluating the robustness of a model.
>
> Clarify the methods used in Section 6.
> > As you have suggested, we have described them in the same way the architectures are described.
>
> Expand the conclusion.
> > Done.
>
>
> Thank you for your numerous suggestions. They have certainly increased the clarity and effectiveness of our writing.
>
> Regarding the concern about the experimental setup, please allow us the opportunity to comment on the state of the datasets used in the AV ASR literature.
>
> > There are three public AV ASR datasets commonly used in the literature made available by researchers from Oxford (https://www.robots.ox.ac.uk/~vgg/data/lip_reading/): LRW, LRS2, and LRS3.
>
> > LRW and LRS2 are based on BBC videos, and are problematic for two reasons. First, the licensing agreement prohibits their use by commercial organizations. Researchers from Google have explicitly cited licensing as their reason for avoiding the use of these datasets [1]. This is what researchers from Meta have said about these datasets (https://openreview.net/forum?id=Z1Qlm11uOM):
> "We would love to evaluate AV-HuBERT on LRS2. However, the license (link) does not permit usage by all researchers, e.g., doesn’t permit commercial organizations, and hence the LRS dataset is in fact not available to everyone in the research community. Note that such a restriction had also prevented many prior studies [1,2] from evaluating their large-scale experiments on the LRS2 dataset. On the other hand, LRS3 is the most widely used benchmark for large-scale lip reading models. Using LRS3 allows us to directly compare our model to existing approaches from other academic and industrial organizations. In addition, LRS3 is the largest publicly available lip reading dataset. Its total duration of videos is twice the size of LRS2, which makes it easy to simulate both low-resource and high-resource settings. Compared to other lip reading datasets, it is also more challenging as all the videos are collected from online resources with more diverse acoustic and linguistic variations. It covers a wide range of speakers (>5K) and has a speaker-independent setup, which is harder yet more realistic". Second, given the sensitive nature of biometric data, it is unclear if the videos used by LRW and LRS2 are actually compliant with European Data Protection laws. For these reasons, we avoid the use of these datasets, and encourage others in the community to avoid them as well.
>
> > We do report on experiments by training and testing on LRS3 in Appendix Section E, and incidentally show state-of-the-art numbers on them. We also follow common practice in the literature by employing the widely used NoiseX corpus to simulate noisy test conditions. There are multiple open-source scripts for doing this (for example, https://github.com/jtkim-kaist/Speech-enhancement). Thus, it is in fact possible for others in the literature to compare their work with us. We do not have a table comparing our numbers on LRS3 with prior work in the literature, because absolute WER numbers are not the focus of our work. However, we can include them if you'd like us to.
>
> > We understand the reviewer's concern that results based on internal datasets presented in the main paper are not reproducible. However, we have nonetheless decided to use them in our main paper and leave LRS3 for the Appendix, because our test data is 30x bigger than LRS3, which increases the statistical confidence of our results. While our organization forbids us from sharing internal user data, we have extensively documented our results in the Appendix to allow other researchers as much insight as possible from such large-scale AV ASR studies.
>
> [1] Audio-visual Speech Recognition is Worth 32x32x8 Voxels. Serdyuk et al.

---

> > ### Author Response · Authors · 2022-07-06
> > **Response 2**
> >
> > The LRS3 dataset is based on a Creative Commons license, which means it might be possible for us to release an actual benchmark dataset based on LRS3 where we make available the various test suites we proposed in our paper. We seek your patience as we determine whether we are allowed to do so.

---

> > > ### Author Response · Authors · 2022-07-06
> > > **Response 3**
> > >
> > > Unfortunately, it seems that maintaining GDPR compliance is difficult even when it comes to releasing simulated datasets from already open-source datasets like LRS3. If our paper is accepted, we will release an example script to show how one can easily create missing video test suites from LRS3 or any other dataset.

---

### Decision · Action_Editors · 2022-08-05

**Recommendation:** Accept with minor revision

**Comment:**

Two reviewers are leaning to accept, but one reviewer has raised some concerns on the fact that the proposed cascaded method relies on knowing exactly which frames are missing during evaluation. In this case, simply swapping between an audio-only model and audio-visual model can be a trivial solution. I agree with this reviewer to some degree. However, considering the follow reasons, I recommend acception but with some revisions.

Reasons:
1. The contribution of this paper include both a systematic evaluation framework for robustness and a cascaded model to achieve robustness. The overall study and analysis on robustness is of some value to the community, even if there are some concerns on the proposed cascaded framework.
2. When knowing exactly which frames are missing during evaluation, I think simply swapping between an audio-only and audio-visual models is non-trivial. Assume that there are some frames with video missing, I am not sure whether we can simply concatenate the outputs of two models to form the final output, since the alignment between text output and audio/video input is not accurate enough. How to decide when to switch to another model?

Revision:
1. The authors should revise the paper carefully according to each reviewer's comments.
2. The authors should comprehensively discuss this issue (the proposed cascaded method relies on knowing exactly which frames are missing during evaluation) and propose some possible solutions.

---

> ### Author Response · Authors · 2022-08-08
> **Response**
>
> Thank you so much for the acceptance of our paper.
>
> We have uploaded a camera ready version taking into accounting the extensive suggestions made by the reviewers. In particular, we acknowledge the limitation that the cascade method relies on knowledge that the video frame is missing, and have included a discussion of this issue along with possible solutions.
>
> We have also uploaded an example test script in tensorflow in the Supplementary materials that shows how one can generate these robustness test suites. We hope this will be useful for others in the community to do their own robustness to missing video analysis on their own models and datasets.
>
> We sincerely appreciate all the constructive feedback and guidance from the reviewers and action editors. Thank you.